# Ultra-stable low-coordinated $Pt_{SA}$/CeZrO$_2$ ordered macroporous structure integrated industrial-scale monolithic catalysts for high-temperature oxidation

Baojian Zhang[1,2,3], Rui Liu[1,2,3], Liangwei Li[1,2,3], Weihong Guo[1,2,3], Biluan Zhang[1,2], Bosheng Chen[1,2], Weidong Yuan[1,2], Pan Li[1,2,3], Shaowen Zhang[1,2], Jinlong Wang[1,2,3], Ji Yang[1,2,3], Zhu Luo [1,2,3] ✉ & Yanbing Guo [1,2,3] ✉

Platinum-group metals (Pt) commonly used in thermal catalytic processes often suffer from catalyst deactivation, such as Pt sintering, Pt overoxidation, and Pt loss under high-temperature conditions. To address these, we present a novel $Pt_{SA}$/CeZrO$_2$ catalyst, featuring isolated Pt single atoms ($Pt_{SA}$) on a Ce$_{0.8}$Zr$_{0.2}$O$_2$ support with an ordered macroporous (OM) structure. Firstly, Zr-stabilized dynamic low-coordinated $Pt_{SA}$ releases more free $d$-electrons by reducing Pt-O bond occupation, thereby preserving peroxide activity at high temperatures and enhancing propane C−H activation. Additionally, the OM structure prevents Pt loss and reduces Pt loading to 0.4 $g_{Pt}$/L, compared with 0.9 $g_{Pt}$/L in commercial diesel oxidation catalysts. As a result, the $Pt_{SA}$/CeZrO$_2$ maintains 92% conversion at 450 °C even after 50 h aging at 800 °C with 10 vol.% H$_2$O. Finally, the catalyst is integrated into a 3.4-liter commercial cordierite monolith for developing and scaling robust catalytic converters.

Platinum-group noble metals (Pt) involved in thermal catalytic processes are widely adopted to address energy and environmental issues[1–5]. However, a significant challenge persists in mitigating catalyst deactivation under harsh thermal conditions[3–6]. The primary mechanisms of deactivation include Pt sintering, Pt overoxidation, and Pt loss, which result in reduced active surface area, formation of less reactive platinum oxides, and physical loss of active sites[3–9]. In industrial applications, increasing Pt loading was adopted to preserve metallic Pt active sites post-thermal equilibrium, but suffering from the high cost of noble metals[10].

Recent advancements, using the atom trapping method, a series of Pt single-atom ($Pt_{SA}$) catalysts, such as $Pt_{SA}$/MgAl$_2$O$_4$, $Pt_{SA}$/Fe$_2$O$_3$, and $Pt_{SA}$/CeO$_2$, were obtained to resist Pt sintering[3,7,11–13]. Among them, the $Pt_{SA}$/CeO$_2$ system is potentially a good candidate in the field of tailpipe emission control due to its effective oxygen storage and redox activity[5,7]. These catalysts achieve nearly 100% utilization of Pt atoms and effectively resist Pt sintering at high temperatures (800 °C)[12–14]. Despite these advantages, $Pt_{SA}$ catalysts prepared through atom trapping are prone to overoxidation and excessive coordination due to strong binding with the support surface, which reduces the oxidation activity compared with Pt nanoclusters (~2 nm)[12,15,16]. Pt nanoclusters, however, could revert to a lower activity $Pt_{SA}$ state under oxidizing conditions above 550 °C[1,15]. Moreover, the efficiency of atom trapping at high temperatures has yet to be studied, and minimizing the loss of active components due to high-temperature airflow in industrial settings remains a key challenge. Although the atom trapping method has addressed the issue of Pt sintering, persistent challenges such as Pt overoxidation and Pt loss continue to undermine the activity-stability trade-off in Pt-based catalysts. To date, no catalyst design has successfully

[1]Institute of Environmental and Applied Chemistry, College of Chemistry, Central China Normal University, Wuhan, PR China. [2]Engineering Research Center of Photoenergy Utilization for Pollution Control and Carbon Reduction, Ministry of Education, Central China Normal University, Wuhan, PR China. [3]Wuhan Institute of Photochemistry and Technology, Wuhan, PR China. ✉e-mail: luo.z@ccnu.edu.cn; guoyanbing@mail.ccnu.edu.cn

suppressed all three deactivation mechanisms, limiting the development of highly durable catalysts for industrial applications.

Addressing these three deactivation challenges necessitates an innovative design strategy that accounts for both micron-scale and atomic-scale structures. Specifically, at the micron-scale, an ordered macroporous (OM) support, with its interconnected macropores and mesopores, can further improve Pt capture efficiency, reducing Pt loss under harsh temperatures[17,18]. Transitioning to the atomic scale, the enhanced redox properties of the $Ce_{0.8}Zr_{0.2}O_2$ support convert Pt into a highly active metallic state, which may modulate the coordination configuration of Pt atoms to prevent Pt overoxidation[19,20]. However, the implementation of strategies and their industrial application still face critical knowledge gaps and technological challenges. Particularly regarding how Zr doping affects the coordination dynamics of $Pt_{SA}$ sites during high-temperature reactions, and how to achieve scalable monolithic integration of this OM catalyst for industrial use.

Herein, we developed a Pt single-atom catalyst supported on an OM $Ce_{0.8}Zr_{0.2}O_2$ ($Pt_{SA}$/CeZrO$_2$) and successfully integrated it into a 3.4-liter commercial monolith, achieving its potential for industrial scalability. At the micron-scale, the OM structure plays an essential role in minimizing the loss of active components, effectively reducing Pt loss by 37.5% during the high-temperature atomization process. Shifting to the atomic scale, in contrast to the oversaturated six-coordinated $Pt_{SA}$ observed on pure $CeO_2$ after aging at 800 °C, the Zr-doped structure stabilizes a dynamically four-coordinated $Pt_{SA}$. This low-coordinated configuration retains more unoccupied Pt $d$ orbitals, enabling peroxide species to remain active at high temperatures and enhancing propane C−H bond activation. Therefore, the $Pt_{SA}$/CeZrO$_2$ exhibits ultra-stability compared with $Pt_{SA}$/CeO$_2$ and benchmark commercial diesel oxidation catalysts (DOC), even after 50 h of hydrothermal aging at 800 °C with 10 vol.% H$_2$O. This study unveils the dynamic evolution of the $Pt_{SA}$ configuration under oxidative conditions and introduces a dual-scale design strategy to tackle three critical deactivation challenges—Pt sintering, Pt overoxidation, and Pt loss—offering a new paradigm for ultra-stable applications in high-temperature and oxidizing environments.

## Results

### Monolithic integration of industrial-scale $Pt_{SA}$/OM CeZrO$_2$ catalyst

Compared with packed-bed reactors (a type of reactor where catalysts are packed into a column), honeycomb monolith-based structured catalytic devices offer higher heat and mass transport, reduction in uneven temperature distribution across the catalyst, and simpler scalability[14,21]. These advantages make them more cost-effective and efficient for purifying environmental pollutants. However, traditional washcoated monolithic catalysts, while benefiting from these structural advantages, often face challenges in material utilization and catalytic performance due to ineffective exposure of the reactive surface[20–22]. Recently, we successfully integrated three-dimensional ordered macroporous (OM) metal oxide catalysts onto channeled monolithic cordierite substrates. The OM architecture provides structural stability at high temperatures and under mechanical agitation, along with abundant macropores and mesopores that enhance gas-solid interactions and catalytic activity[18,20,23]. Despite the potential benefits of OM-structured catalysts, scaling up to industrially relevant levels is required to bridge this nanotechnology with industrial applications. Herein, we address the challenges associated with process complexity and report the first scalable integration of an OM catalyst onto industrial-scale commercial monolithic honeycombs, ranging in volume from 2.5 cm$^3$ to 3.4 L (Fig. 1a and Supplementary Fig. 1). These catalysts can be directly utilized in Diesel Oxidation Catalyst (DOC) devices for pollutant control.

We employed a template-directed method to grow OM CeZrO$_2$ and CeO$_2$ on honeycomb cordierite substrates, as illustrated in

Supplementary Fig. 2–5. To achieve uniform adhesion of the polystyrene (PS) spheres and sol within the long channels, we used a rapid drying process, followed by two repeated operations. Inductively Coupled Plasma (ICP) analysis determined that the CeZrO$_2$ loading was about 4.01 wt.% (40 g/L) after calcination. Large-area SEM investigation (Fig. 1b) confirmed the uniform distribution of the OM structure across the 3.4 L catalyst surface. SEM images were obtained from three different positions after device sectioning, as illustrated in Fig. 1b (1, 2, 3). The results demonstrated that the periodic macroporous architectures, with a pore size of 0.7 μm and a thickness of 20 μm, uniformly covered the entire surface of the monolithic honeycomb channels. This uniformity prevents localized overheating and uneven mechanical stress, thereby reducing the risk of coating delamination and cracking. Mercury Intrusion Porosimetry (MIP) analysis (Supplementary Fig. 6a) revealed that the OM samples exhibited higher peak intensities than traditional washcoated monolithic catalysts, indicating a greater quantity of mesopores and macropores (20–1000 nm). The increase in mesopores stemmed from the spaces between Ce−Zr solid solution nanoparticles (Supplementary Fig. 7), while macropore formation resulted from the decomposition of PS spheres. Meanwhile, both OM and Powder samples lost less than 1% of their weight after 30 min of ultrasonic treatment at 40 kHz, demonstrating that the OM structure did not damage the mechanical stability of $Pt_{SA}$/CeZrO$_2$ (Supplementary Fig. 6b).

Pt nanoparticles (Pt$_{NP}$) were loaded onto OM CeO$_2$ and CeZrO$_2$ using a microwave-assisted dip-coating method, achieving a Pt loading of 0.4 g$_{Pt}$/L (0.04 wt.%, ICP data)[14]. Rapid microwave drying ensured the uniform distribution of Pt. A thermal aging test was conducted under standard conditions (800 °C in air for 50 h) to evaluate the high-temperature stability of the Pt$_{NP}$/CeZrO$_2$ catalyst[7]. As illustrated in Fig. 1c, it rapidly transformed into a single-atom $Pt_{SA}$/CeZrO$_2$ structure within seconds through an atom trapping mechanism and remained stable, resisting Pt sintering, throughout the 50-h aging process[7]. EXAFS spectra in Fig. 1d demonstrate that the aged $Pt_{SA}$/CeZrO$_2$ catalyst exhibits a prominent Pt−O peak and small Pt−Ce peaks, while the Pt$_{NP}$ sample shows a significant Pt−Pt peak[15]. This demonstrated the conversion of Pt$_{NP}$/CeZrO$_2$ into $Pt_{SA}$/CeZrO$_2$ during aging at 800 °C. STEM images of Pt$_{NP}$/CeZrO$_2$ (Fig. 1e) revealed uniform Pt nanoparticles on the (111) plane of CeZrO$_2$, with an interplanar d-spacing of 0.31 nm and an average particle size of approximately 2.1 nm. STEM results (Fig. 1f) confirmed that the bright contrast spots localized on the Ce/Zr atomic rows, indicating isolated Pt atoms in the Ce vacancies. The single-atom nature of supported Pt was further confirmed by CO-DRIFTS (Supplementary Fig. 8), where, unlike Pt$_{NP}$/CeZrO$_2$ and Pt$_{NP}$/CeO$_2$, both $Pt_{SA}$/CeZrO$_2$ and $Pt_{SA}$/CeO$_2$ exhibited only a linear CO peak at 2101−2112 cm$^{-1}$, indicating CO adsorption on isolated Pt atoms[8].

The Pt loading, CO-DRIFTS, and propane catalytic combustion activity were tested at three different positions of the large-scale $Pt_{SA}$/CeZrO$_2$ catalytic device (Supplementary Fig. 9a). Results indicated similar activity across all three positions, with a Pt loading maintained of approximately 0.38 g$_{Pt}$/L (Supplementary Fig. 9b, c). Furthermore, Pt site maintained its single-atom dispersion across these positions, and no sintering was observed after aging at 800 °C for 50 h, as evidenced by only a linear CO peak at 2101−2112 cm$^{-1}$ in CO-DRIFTS (Supplementary Fig. 9d)[8]. Furthermore, the catalysts prepared in different batches exhibit good repeatability, with the $T_{90}$ error controlled within ±15 °C (Supplementary Fig. 10). The uniformity of the OM structure and Pt loading during the synthesis process is a key control step for performance. The effect of Zr content on catalytic activity was also investigated, and $Pt_{SA}$/Ce$_{0.8}$Zr$_{0.2}$O$_2$ was identified as the optimal formulation (Supplementary Fig. 11). XRD analysis (Supplementary Fig. 12a) showed all characteristic peaks corresponding to cordierite (Mg$_2$Al$_4$Si$_5$O$_{15}$, JCPDS 02-0646) and CeO$_2$ (JCPDS 34-0394), with no detectable peaks for ZrO$_2$ or Pt, suggesting the formation of a ceria-zirconium solid solution and high Pt dispersion[19]. XPS survey spectra

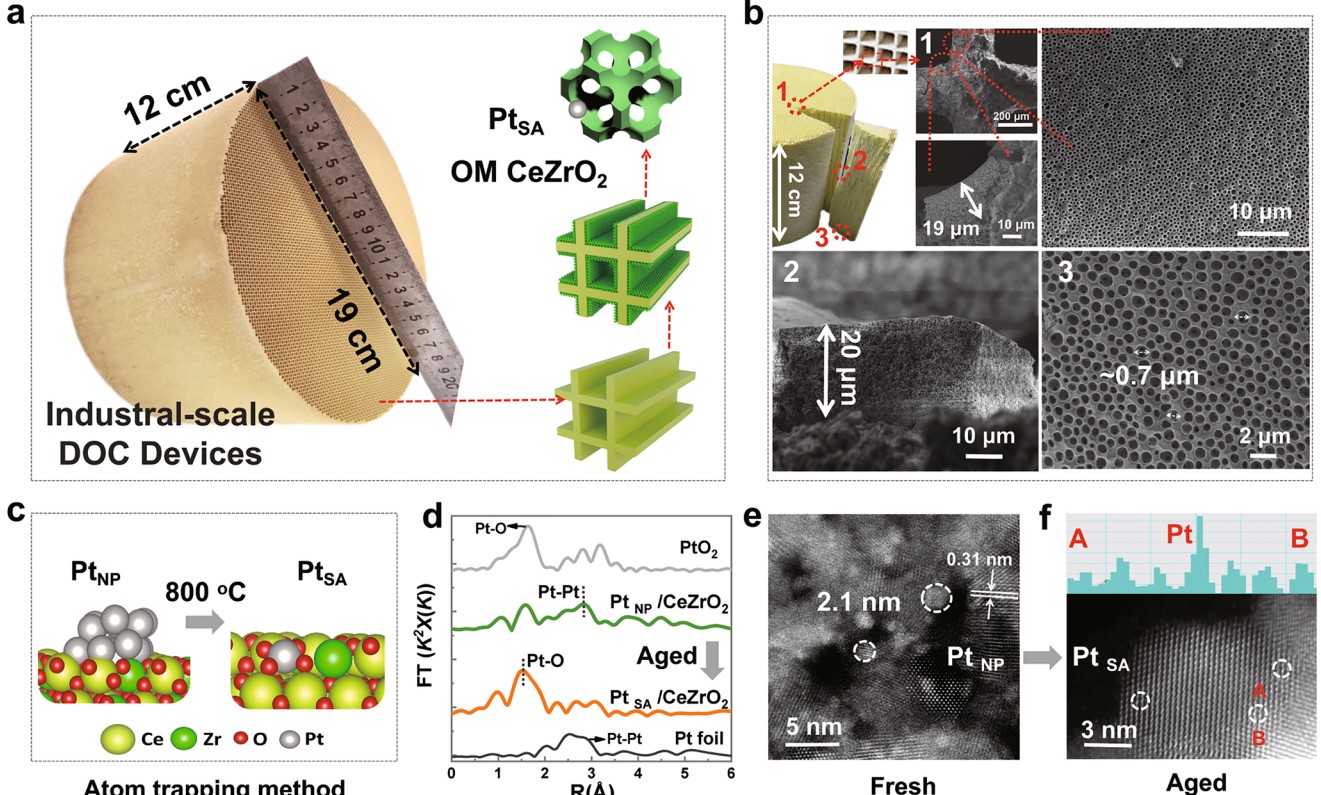

**Fig. 1 | Structure characterization of monolithic ordered macroporous (OM) $Pt_{SA}$/CeZrO$_2$. a** Photograph of an OM catalyst integrated commercial cordierite honeycomb monolith. **b** Large-area Scanning Electron Microscopy (SEM) investigation of the OM catalyst integrated inside the monolithic honeycomb channels, showcasing both cross-sectional and top views, which reveal a uniform coverage of the OM catalyst throughout the entire 12 cm-long honeycomb channels. **c** Schematic diagram illustrating the atom trapping method (Aging conditions:

800 °C for 50 h in air). **d** $k^2$-weighted Fourier-transformed Extended X-ray Absorption Fine Structure (EXAFS) spectra of fresh $Pt_{NP}$/CeZrO$_2$ and aged $Pt_{SA}$/CeZrO$_2$. Pt foil and PtO$_2$ are employed as references. **e, f** Annular Dark Field Scanning Transmission Electron Microscopy (ac-HADDF STEM) images of fresh $Pt_{NP}$/CeZrO$_2$ (**e**) and aged atomically dispersed $Pt_{SA}$/CeZrO$_2$ (**f**), where $Pt_{NP}$ and $Pt_{SA}$ represent Pt nanoparticles and Pt single atoms, respectively.

and EDS mapping (Supplementary Fig. 12b, c) further confirmed the uniform doping of Zr and Pt loading. Combining EXAFS, STEM, and CO-DRIFTS results confirmed that after aging at 800 °C, supported Pt nanoparticles were atomized, transforming into isolated Pt single-atom catalysts without sintering.

## Catalytic activity and stability

A thermal aging test at 800 °C was conducted to evaluate the catalyst's high-temperature endurance. Fresh $Pt_{NP}$/CeO$_2$ and $Pt_{NP}$/CeZrO$_2$ exhibit similar catalytic activity ($T_{50}$ = 340 °C vs $T_{50}$ = 360 °C) as shown in Fig. 2a. After 50 h of aging at 800 °C in air, the activity of aged $Pt_{SA}$/CeZrO$_2$ remained stable ($T_{50}$ = 349 °C) due to the Zr addition, whereas $Pt_{SA}$/CeO$_2$ showed a notable activity decline ($T_{50}$ = 451 °C), with a 100 °C increase in $T_{50}$. The normalized reaction rate of aged $Pt_{SA}$/CeO$_2$ at 450 °C decreased by 50% (460−227 μmol/(g$_{cat}$*s), Fig. 2b). The corresponding Arrhenius plots in Fig. 2c also reveal that $Pt_{SA}$/CeZrO$_2$ (48.4 kJ/mol) has a lower apparent activation energy (Ea) than $Pt_{SA}$/CeO$_2$ (103.3 kJ/mol). Notably, as illustrated in Fig. 2d, after 50 h of hydrothermal aging at 800 °C with 10 vol% H$_2$O, the conversion of $Pt_{SA}$/CeO$_2$ decreased significantly from 45% at 450 °C to just 12%. In contrast, $Pt_{SA}$/CeZrO$_2$ maintained a high conversion of 92% within 1000 h. Additionally, the $Pt_{SA}$/CeZrO$_2$ catalyst exhibited good catalytic stability under different temperatures from 250 to 450 °C, and maintained good cycling stability over three test rounds ranging from 100 to 800 °C, indicating its stable intrinsic activity after aging at 800 °C (Supplementary Fig. 13). As shown in Supplementary Fig. 14, $Pt_{SA}$/CeZrO$_2$ also demonstrated good activity and stability in the catalytic

oxidation reactions of other VOCs (100 ppm toluene). To better evaluate the stability of as-prepared catalysts, the commercial PtPd/CeZr-based catalyst (Supplementary Fig. 15a–d) was adopted as a control sample. As shown in Fig. 2e, after 50 h of thermal aging at 800 °C in air, the low Pt content $Pt_{SA}$/CeZrO$_2$ displayed a lower $T_{90}$ (0.4 g$_{Pt}$/L, $T_{90}$ = 400 °C) than the higher Pt-loaded commercial PtPd/CeZr-based catalyst (0.9 g$_{Pt}$/L, $_{90}$ = 455 °C). This highlights its enhanced precious metal utilization efficiency and superior durability.

A more rigorous aging test at 1100 °C was conducted to assess the extreme-temperature tolerance of the $Pt_{SA}$/CeZrO$_2$ catalyst, using commercial Pt/MnCoO$_x$ and PtPd/CeZr catalysts for comparison. After thermal aging at 800 and 1100 °C (Supplementary Fig. 15e–g), the $Pt_{SA}$/CeZrO$_2$ catalyst exhibited the smallest change in $T_{90}$ (fresh, 800 °C-aged, and 1100 °C-aged samples: $T_{90}$ = 400, 400, and 460 °C, respectively). In contrast, the activity decline was more pronounced for the commercial Pt/MnCoO$_x$ ($T_{90}$ = 420, 560, 650 °C) and Com-PtPd/CeZr catalysts ($T_{90}$ = 450, 455, 600 °C). These results indicate that Zr doping significantly improves the high-temperature resistance of the $Pt_{SA}$/CeZrO$_2$ catalyst. Supplementary Tables 1 and 2 show that $Pt_{SA}$/CeZrO$_2$ exhibits superior catalytic activity for C$_3$H$_8$ oxidation compared with other Pt-based catalysts reported in the literature, even after thermal aging at 800 °C. Moreover, the $Pt_{SA}$/CeZrO$_2$ catalyst shows excellent sulfur resistance, maintaining a stable propane conversion of 96% at 400 °C with 200 ppm SO$_2$ for 50 h (Supplementary Fig. 16).

To further investigate the high-temperature stability mechanism of the $Pt_{SA}$/CeZrO$_2$ catalyst, SEM, STEM, CO-DRIFTS, XPS, and XANES were applied to characterize the morphological structure and

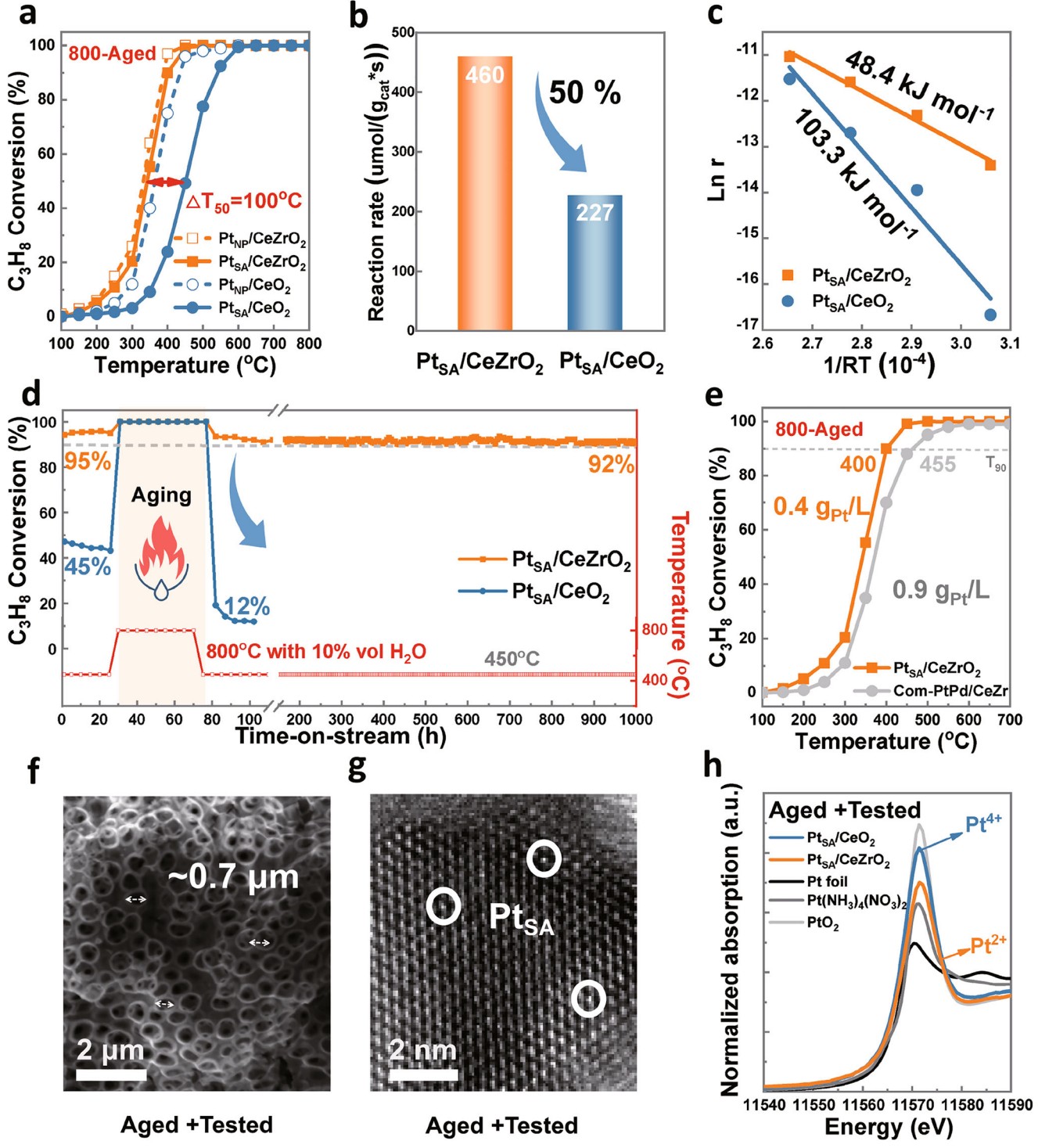

**Fig. 2 | Catalytic performance of aged Pt$_{SA}$/CeO$_2$ and Pt$_{SA}$/CeZrO$_2$. a** Light-off curves for fresh Pt$_{NP}$/CeO$_2$, Pt$_{NP}$/CeZrO$_2$, and aged Pt$_{SA}$/CeO$_2$, Pt$_{SA}$/CeZrO$_2$ (Aging conditions: 800 °C for 50 h in air). **b** Normalized reaction rate of aged catalysts at 450 °C. **c** Arrhenius plots and apparent activation energy of aged catalysts. **d** Durability performance before and after hydrothermal aging. The testing and aging cycling experiments consist of three stages: (1) Stability test at 450 °C for 25 h, (2) Aging in humid air (10 vol% H$_2$O) at 800 °C for 50 h (shadow part with torch), (3) Stability test in humid air (10 vol% H$_2$O) at 450 °C for ultralong 925 h. **e** Comparison of aged Pt$_{SA}$/CeZrO$_2$ (0.4 g$_{Pt}$/L) with an aged commercial PtPd/CeZr catalyst (0.9 g$_{Pt}$/L + 0.1g$_{Pd}$/L). **f** SEM and **g** STEM images of aged + tested Pt$_{SA}$/CeZrO$_2$ after aging and testing in Fig. 2d. **h** Normalized XANES spectra at the Pt L$_3$ edge of aged + tested Pt$_{SA}$/CeO$_2$ and Pt$_{SA}$/CeZrO$_2$ samples after aging and testing in Fig. 2d.

oxidation state of the aged + tested samples. SEM images (Fig. 2f and Supplementary Fig. 17a) revealed that, despite slight deformation, the OM structure retained its porous form, with pore sizes around 0.7 μm. STEM, along with in situ CO-DRIFTS analysis (Fig. 2g, Supplementary Fig. 17b and Supplementary Fig. 18), showed no Pt agglomeration, confirming the preservation of the Pt single-atom structure. These results confirm that both Pt$_{SA}$/CeO$_2$ and Pt$_{SA}$/CeZrO$_2$ retained their OM micron-structures and single-atom dispersion after the reaction. In contrast, the XANES spectra (Fig. 2h) demonstrated that the aged + tested Pt$_{SA}$/CeO$_2$ exhibited a high white-line intensity, similar to Pt$^{4+}$ in PtO$_2$. Zr doping significantly reduced the white-line intensity of aged + tested Pt$_{SA}$/CeZrO$_2$, resembling the characteristics of Pt$^{2+}$ in Pt (NH$_3$)$_4$

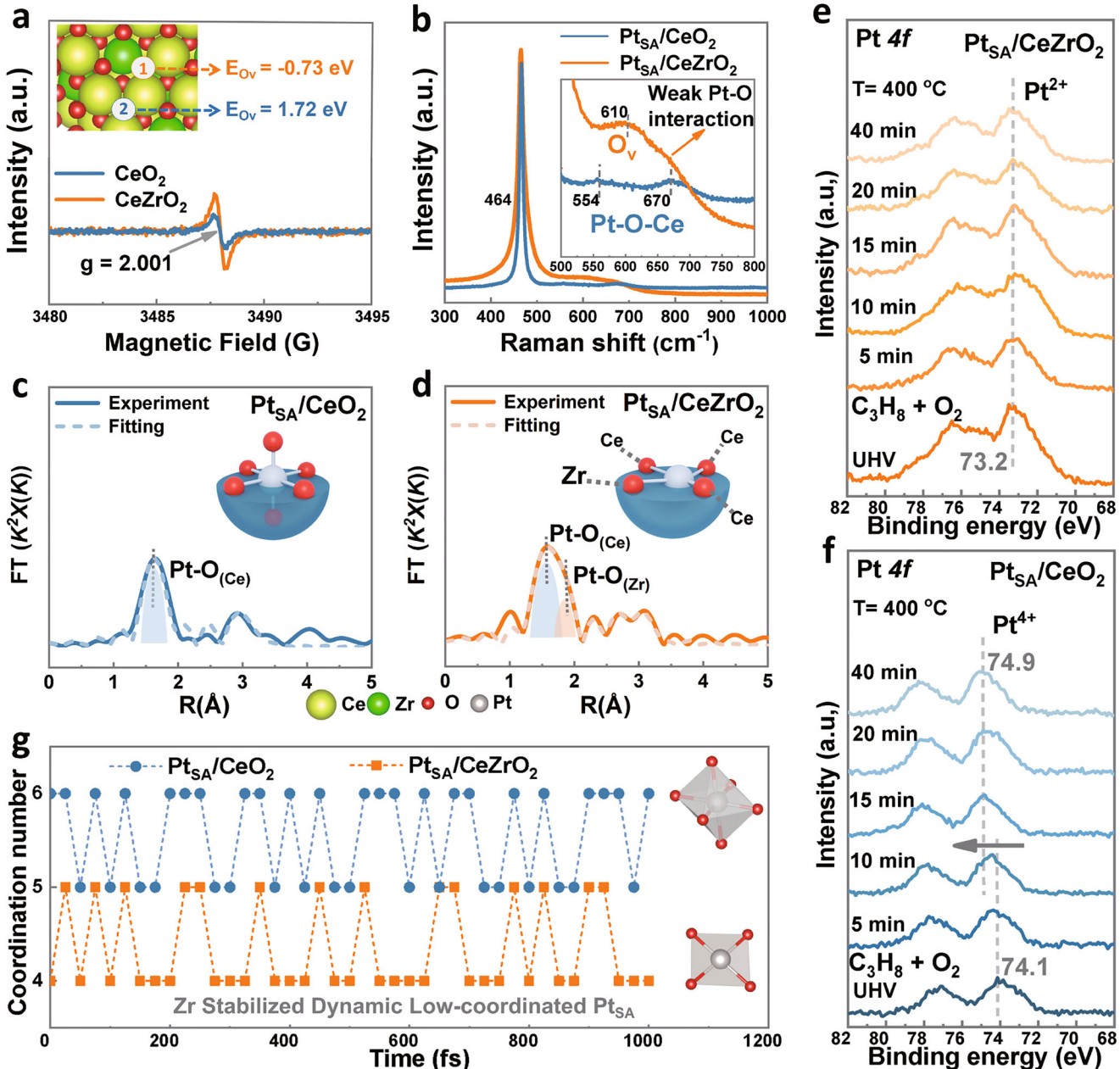

**Fig. 3 | Effect of Zr on the Pt$_{SA}$ coordination environment. a** Low-temperature Electron Paramagnetic Resonance (EPR) spectrum of CeO$_2$ and CeZrO$_2$ at 77 K under high vacuum, with an inside showing a DFT model of CeZrO$_2$ and the formation energies of oxygen vacancies at different positions. **b** Raman spectra of aged Pt$_{SA}$/CeO$_2$ and Pt$_{SA}$/CeZrO$_2$. Extended X-ray Absorption Fine Structure (EXAFS) data fitting for aged Pt$_{SA}$/CeO$_2$ (**c**) and Pt$_{SA}$/CeZrO$_2$ (**d**), the inside illustration shows the Pt$_{SA}$ coordination. In situ NAP-XPS Pt *4f* spectra for Pt$_{SA}$/CeZrO$_2$ (**e**) and Pt$_{SA}$/CeO$_2$ (**f**) before and after dosing C$_3$H$_8$ + O$_2$. **g** Selected snapshots of the ab initio molecular dynamics (AIMD) trajectory for Pt single-atom coordination number over Pt$_{SA}$/CeO$_2$ and Pt$_{SA}$/CeZrO$_2$ under actual reaction conditions (1 C$_3$H$_8$ and 20 O$_2$ molecules at 400 °C), the inside illustration shows the Pt$_{SA}$ coordination.

(NO$_3$)$_2$. The XPS Pt *4f* results further confirmed that Zr doping decreased the proportion of Pt$^{4+}$ species (74.0 eV B.E.) on the surface of aged + tested Pt$_{SA}$/CeZrO$_2$ from 37% to 25% (Supplementary Fig. 17c). It is evident that Zr doping effectively prevents Pt overoxidation during high-temperature aging and usage, enhancing the balance between activity and stability in Pt/CeO$_2$-based catalysts.

### Zr doping stabilized dynamic low-coordinated single-atom Pt

To reveal the anti-overoxidation mechanism of the Pt$_{SA}$/CeZrO$_2$ catalyst at the atomic scale, we employed Raman, XAS, and AIMD methods to investigate the impact of Zr on the local atomic coordination of Pt$_{SA}$, with particular focus on the dynamic evolution of atomic

configurations induced by high-temperature reactions. Firstly, the CeZrO$_2$ DFT model was established with the most exposed (111) surfaces (in Fig. 3a) based on the standard cards of XRD in Supplementary Fig. 12a. The calculated oxygen vacancy formation energy ($E_{Ov}$) in Fig. 3a indicates that oxygen vacancies (O$_v$) are more easily formed around Zr atoms (−0.73 eV) compared with other sites where the formation energy is higher (1.72 eV). Meanwhile, the stronger peak intensity at $g = 2.001$ in the EPR spectrum of CeZrO$_2$ (Fig. 3a) further demonstrated its higher concentration of O$_v$ than CeO$_2$[23]. Additionally, the stronger peak at 610 cm$^{-1}$ in the Raman spectrum (Fig. 3b) represents more O$_v$ in Pt$_{SA}$/CeZrO$_2$ induced by Zr doping[20]. The lattice distortion caused by the small-sized Zr doping could account for the

generation of $O_v$ near Zr atoms, which in turn would influence the position and chemical environment of $Pt_{SA}$[19]. It is also worth noting that the weaker Pt–O–Ce vibration peaks at 554 and 670 $cm^{-1}$ appear on $Pt_{SA}/CeZrO_2$, indicating that Zr doping weakens the excessively strong metal-support interaction in the $Pt/CeO_2$ system.

The $CeZrO_2$ DFT model and the loading position of $Pt_{SA}$ were optimized, as detailed in Supplementary Fig. 19. The most thermodynamically stable configurations of $Pt_{SA}/CeO_2$ and $Pt_{SA}/CeZrO_2$ are shown in Supplementary Fig. 20, where $Pt_{SA}$ bonds with six oxygen atoms on pure $CeO_2$, but coordinates with four oxygen atoms in $CeZrO_2$. Furthermore, thermodynamic calculations demonstrate that the lowest formation energy occurs when $Pt_{SA}$ is anchored near Zr atoms and oxygen vacancies ($O_v$). The STEM results (Supplementary Fig. 21) also show an asymmetric atomic distance on both sides of the $Pt_{SA}$ bright spots, which is caused by local lattice distortion induced by the differences in the atomic radii and electronegativity between Ce and Zr. This further indicates that Pt is more likely to be loaded around Zr atoms. To further confirm changes in the local coordination environment of $Pt_{SA}$ sites in the Zr-doped sample, EXAFS was fitted with the results shown in both R space and k space (Fig. 3c, d and Supplementary Fig. 22). In contrast to the symmetric Pt–O$_{(-Ce)}$ peak around 1.71 Å in $Pt_{SA}/CeO_2$ (Fig. 3c), the peak in $Pt_{SA}/CeZrO_2$ exhibits asymmetry (Fig. 3d and Supplementary Fig. 23), due to Pt loaded around Zr atom and the co-existence of Pt–O$_{(Ce)}$ and Pt–O$_{(Zr)}$ bonds with different but similar bond lengths[15]. As summarized from real-space fitting data (Supplementary Table 3), the coordination number of the Pt–O bond on $Pt_{SA}/CeO_2$ (~5.8) is higher than that on $Pt_{SA}/CeZrO_2$ (~4.1), demonstrating that a low-coordinated $Pt_{SA}$ catalyst is constructed by Zr doping. Meanwhile, an asymmetric square-planar $Pt_1O_4$ geometry of $Pt_{SA}/CeZrO_2$ was evidenced by three shorter Pt–O$_{(Ce)}$ of 1.67 Å and one longer Pt–O$_{(Zr)}$ of 1.97 Å. Based on the Raman, DFT, and EXAFS results, we concluded that the simple Zr doping strategy stabilized $Pt_{SA}/CeZrO_2$ in a low-coordination state even after aging at 800 °C, avoiding an oversaturated $Pt_{SA}$ supported in pure $CeO_2$.

However, during the temperature-driven reaction, the atomic structure of active sites may considerably change, resulting in unanticipated active states. We employed in situ NAP-XPS to characterize the chemical environment changes of $Pt_{SA}/CeZrO_2$ and $Pt_{SA}/CeO_2$ under reaction conditions (400 °C, $C_3H_8:O_2 = 1:20$, 0.2 mbar). As shown in Fig. 3e, f, the binding energies of $Pt_{SA}/CeZrO_2$ and $Pt_{SA}/CeO_2$ are 73.2 eV and 74.1 eV, respectively, representing $Pt^{2+}$ and $Pt^{4+}$ species, which are consistent with the XANES analysis (Fig. 2h), proving that Pt is closer to the metallic state with the assistance of Zr. Furthermore, as the reaction gas ($C_3H_8 + O_2$) was introduced, the binding energy of $Pt_{SA}/CeZrO_2$ remained at 73.2 eV, indicating that the catalyst can withstand high-temperature oxidative environments and maintain Pt in a low oxidation state without undergoing overoxidation. However, the binding energy of $Pt_{SA}/CeO_2$ shifted gradually to higher binding energies (from 74.1 to 74.9 eV) as the reaction proceeded (0 → 15 min), stabilizing at 74.9 eV from 15 to 40 min. This indicates that Pt underwent oxidation during the reaction and stabilized in a higher oxidation state, which is unfavorable for the adsorption and activation of reactants. In situ NAP-XPS results further demonstrate that $Pt_{SA}/CeZrO_2$ with Zr assistance has better oxidation resistance than $Pt_{SA}/CeO_2$, and confirm the oxidation resistance mechanism derived from EXAFS (Fig. 3c, d), where Zr helps stabilize Pt in a low-coordination state.

To further investigate the femtosecond-level dynamic evolution of $Pt_{SA}$ coordination configuration under actual reaction conditions, AIMD simulations were performed. (Fig. 3g, Supplementary Fig. 24 and Supplementary Fig. 25a, 1 $C_3H_8$: 20 $O_2$; at 400 °C). The results show that the $Pt_{SA}$ supported on $CeZrO_2$ does not always maintain a 4-coordinated state. As the reaction occurs, the Pt–O coordination number dynamically fluctuates between 4 and 5, as the adsorption and desorption of reactants. Similarly, the coordination number of $Pt_{SA}$ on

pure $CeO_2$ changes between 5 and 6. It is concluded that the $Pt_{SA}/CeZrO_2$ maintains a dynamic low-coordinated configuration, with Pt–O coordination number mostly lower than $Pt_{SA}/CeO_2$. The Pt $4f$ XPS results (Supplementary Fig. 25b) further demonstrated that the Pt oxidation state is almost unchanged in the used $Pt_{SA}/CeZrO_2$ sample (Fresh sample $Pt^0$: 40%; Aged and tested sample $Pt^0$: 38%). In conclusion, an efficient dynamic low-coordinated $Pt_{SA}$ structure stabilized by Zr doping enables the $Pt_{SA}/CeZrO_2$ to resist Pt overoxidation under harsh aging conditions at 800 °C. This dynamically low-coordinated configuration is the first to be proven as the active form to efficiently catalyze $C_3H_8$ to $CO_2$ under actual reaction conditions.

## Adsorption and activation of reactants

The distinct electronic state of $Pt_{SA}/CeZrO_2$ and $Pt_{SA}/CeO_2$, induced by changes in coordination structure, contributes to their significant difference in catalytic performance. Hence, the XPS valence spectra and DFT were carried out to systematically analyze the electronic orbital structure of these two $Pt_{SA}$ sites. The huge difference in oxidation state could be revealed by the calculated charge differential density (Supplementary Fig. 26a and 17b), indicating that the decreased number of Pt–O bonds allows fewer electrons to be transferred from Pt to $CeZrO_2$ (0.99 $e^-$), compared with 1.89 $e^-$ transferred from Pt to pure $CeO_2$. As shown in Fig. 4a, consistent with the reduction of Pt–O coordination number, the Pt $d$-band center of $Pt_{SA}/CeZrO_2$ is closer to the Fermi level compared to $Pt_{SA}/CeO_2$ catalyst. This shift is also reflected in the lower Pt $4f$ XPS binding energies for $Pt_{SA}/CeZrO_2$ (Supplementary Fig. 17c)[24]. Usually, as low coordination occurs, the decrease in electron orbital overlap would induce an upshift of the $d$-band center so as to activate the $d$-band filling, and then the $d$ electrons become more active[24]. Supplementary Fig. 26c depicts the ELF contour plots for $Pt_{SA}$ with different coordinated structures, and the electrons of low-coordinated $Pt_{SA}$ are closer to the state of electron vaporization (0.39 in $Pt_{SA}/CeZrO_2$ vs 0.28 in $Pt_{SA}/CeO_2$). These results suggest that as the coordination number decreases, reduced electron orbital overlap raises the $d$-band center, enhancing the reactivity of the $d$ electrons.

The low-coordinated local environment of $Pt_{SA}$, along with the active d-orbital electrons, further promotes the adsorption and activation of reactant molecules. Firstly, the adsorption energies of $O_2$ and $C_3H_8$ were calculated for the two models, with the four most stable adsorption configurations shown in Supplementary Fig. 27. In the $Pt_{SA}/CeZrO_2$ sample, $O_2$ is adsorbed at the oxygen vacancies around Zr and between near $Pt_{SA}$, while $C_3H_8$ is adsorbed on the molecular oxygen species already adsorbed. In contrast, the adsorption of $O_2$ and $C_3H_8$ on $Pt_{SA}/CeO_2$ is difficult. Specifically, for $Pt_{SA}/CeZrO_2$: $E_{ads}(O_2) = -2.33$ eV, $E_{ads}(C_3H_8) = -1.21$ eV; for $Pt_{SA}/CeO_2$: $E_{ads}(O_2) = -0.56$ eV, $E_{ads}(C_3H_8) = -0.65$ eV. This indicates that the low-coordinated environment of $Pt_{SA}$ facilitates the adsorption of $O_2$ and $C_3H_8$ molecules. Besides, we investigated the catalytic combustion mechanism of $C_3H_8$ using in situ DRIFTS (Fig. 4b). The peaks at 2905 and 2960 $cm^{-1}$ confirm $C_3H_8$ adsorption, with higher intensity observed on $Pt_{SA}/CeZrO_2$, indicating enhanced adsorption due to the low-coordinated $Pt_{SA}$[20,25]. Upon introducing $O_2$, peak intensity decreased, especially on $Pt_{SA}/CeZrO_2$, suggesting higher oxidation activity. The peak at 1667 $cm^{-1}$, unique to the $Pt_{SA}/CeZrO_2$ sample, corresponds to the vas C=C functional group ($CH_2 = CH - O^-$) in allylic alcohols and allyl species[20,25]. This indicates that low-coordination $Pt_{SA}$ effectively activates C–H bonds, shifting the oxidation pathway from propionate to a more efficient acrylate pathway. The transition state of the rate-limiting step (Fig. 4c) further reveals that the thermodynamic energy barrier for C–H cleavage of $C_3H_8$ on $Pt_{SA}/CeZrO_2$ (0.17 eV) is lower than that on $Pt_{SA}/CeO_2$ (0.98 eV), highlighting that low-coordinated $Pt_{SA}$ enhances C–H bond cleavage, thereby promoting the degradation of $C_3H_8$ more effectively.

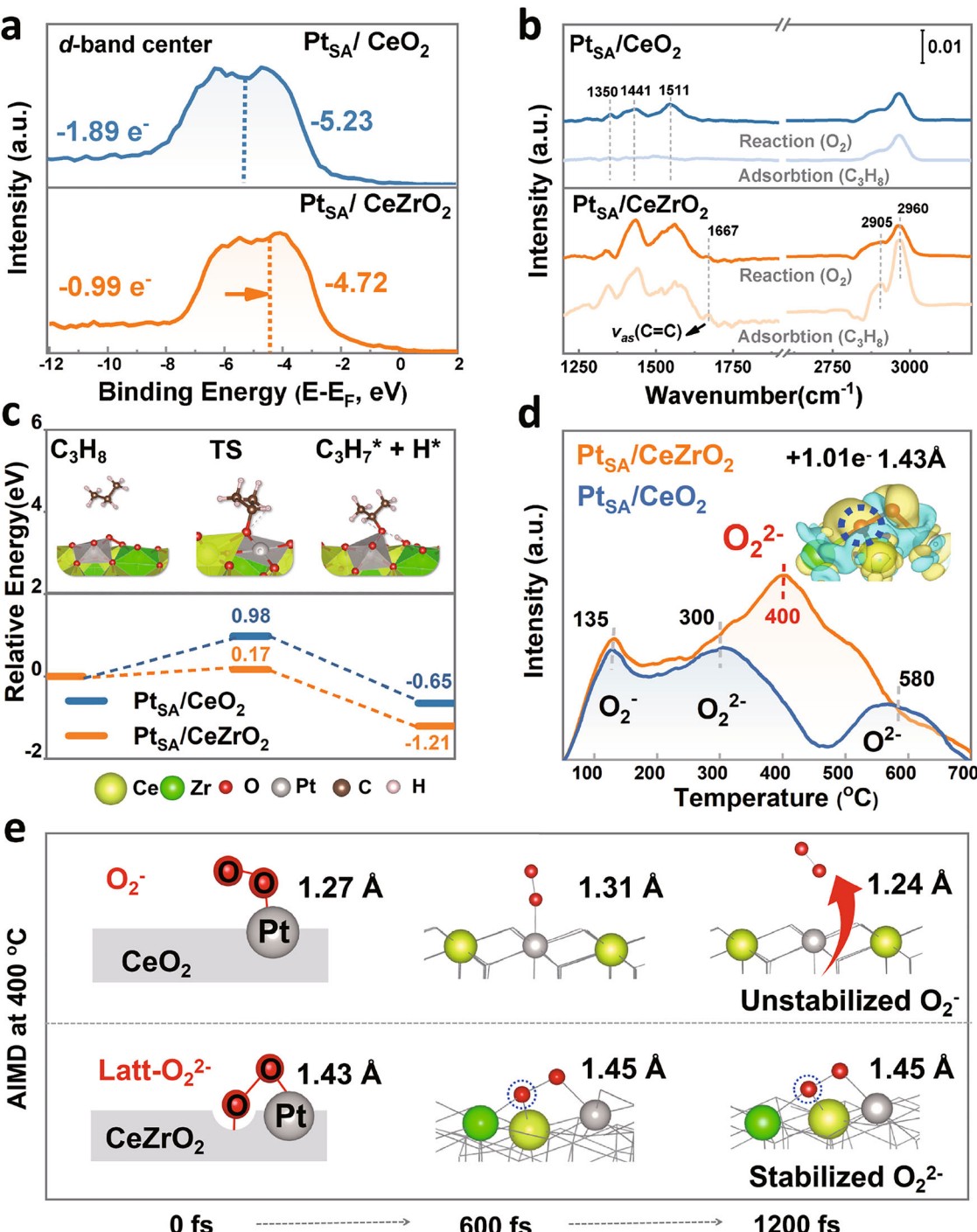

**Fig. 4 | Effect of Pt coordination environment on adsorption and activation behavior. a** X-ray photoelectron spectroscopy (XPS) valence band spectra of Pt, with the inner values representing the calculated charge differential density. **b** In situ DRIFTS results for $C_3H_8 + N_2$ and $C_3H_8 + O_2 + N_2$ at 200 °C. **c** Calculated energy barrier for $C_3H_8$ activation. **d** $O_2$-temperature-programmed desorption ($O_2$-TPD) profile of $Pt_{SA}/CeZrO_2$ and $Pt_{SA}/CeO_2$. **e** Selected snapshots from the Ab initio molecular dynamics (AIMD) trajectory illustrating $O_2$ activation on the catalysts at 400 °C.

Furthermore, we investigated the molecular oxygen species on $Pt_{SA}/CeZrO_2$ and $Pt_{SA}/CeO_2$ catalysts, focusing on their actual active state and dynamic evolutions at operating temperature (400 °C) using $O_2$-TPD, XPS, EPR, DFT, and AIMD methods. The $O_2$-TPD results (Fig. 4d) identified three oxygen species appearing with increasing temperature: superoxide species ($O_2^-$ at ~135 °C), peroxide species ($O_2^{2-}$ at ~400 °C), and lattice oxygen ($O_{latt}^{2-}$ at ~510 °C)[26–28]. Notably, $Pt_{SA}/CeZrO_2$ exhibited stronger peaks (0.10 vs. 0.06 mmol/g), indicating that Zr doping favors the activation of adsorbed oxygen species.

Specifically, the strongest peak at 400 °C suggests that the low-coordinated $Pt_{SA}$ promotes the formation of peroxide species ($O_2^{2-}$). XPS O 1s and EPR data further confirm the increased presence of peroxide species on $Pt_{SA}/CeZrO_2$ as described in Supplementary Fig. 28. Additionally, charge differential density analysis (Fig. 4d inset) shows that oxygen molecules adsorbed between $Pt_{SA}$ and nearby oxygen vacancies gain 1.01 e$^-$, forming chemisorbed peroxide $O_2^{2-}$ species with a bond length of 1.43 Å. Unlike typical adsorbed oxygen on $Pt_{SA}/CeO_2$, the peroxide species on $Pt_{SA}/CeZrO_2$ has one oxygen

atom embedded in a lattice vacancy (Fig. 4e). Due to its combination of the bond length of peroxide species and the position of lattice oxygen, we refer to it as a lattice peroxide species (Latt-$O_2^{2-}$). This unique Latt-$O_2^{2-}$ could potentially combine the reactivity advantage of molecular oxygen with the stability advantage of lattice oxygen. AIMD simulation further demonstrated the stability of Latt-$O_2^{2-}$ species at realistic catalytic environments (400 °C). From the snapshots in Fig. 4e, after 1000 fs of simulation, superoxide ($O_2^-$) desorbed from the $Pt_{SA}$/$CeO_2$ surface, while the Latt-$O_2^{2-}$ species on $Pt_{SA}$/$CeZrO_2$ remained in an activated state with a bond length of 1.45 Å. These findings first suggest that the Latt-$O_2^{2-}$ species, with its enhanced thermal stability, is the most promising reactive oxygen species for triggering $C_3H_8$ oxidation at higher operating temperatures (300–400 °C).

Combining the above atomic-scale results, although $Pt_{SA}$/$CeO_2$ prepared by the atom capture method maintains a non-sintering monoatomic dispersion, its oversaturated coordination leads to excessive electron occupation by Pt–O bonds, resulting in reduced reactivity. In contrast, $Pt_{SA}$/$CeZrO_2$, doped with Zr to prevent Pt overoxidation, features dynamically low-coordinated $Pt_{SA}$ sites with more unoccupied $d$ orbitals. This allows peroxide species to stay active at high temperatures, enhancing the activation of propane C–H bonds and improving catalytic performance, even after aging at 800 °C in flowing air. In summary, while Zr doping is an established methodology, this study provides atomic-scale insights into the stabilization mechanism of catalysts, elucidating how Zr modulates the coordination dynamics of metal centers and regulates reactive oxygen species during high-temperature catalysis. The proposed Zr-assisted atom trapping strategy effectively mitigates both Pt sintering and Pt overoxidation.

## Impact of integrated OM micron-structure on catalyst stability

Transitioning to the micron-scale realm, we further explored the influence of the OM structure on catalyst stability. Firstly, computational fluid dynamics (CFD) was used to examine gas flow patterns across three systems: an OM monolithic catalyst, a powdered catalyst, and a blank monolith. Figure 5a shows the OM monolith and its simulated airflow, revealing laminar flow through the orifice at a face velocity of 5 m/s[29]. Compared with powdered catalysts, the monolithic design significantly reduces the pressure drop from 502 to 97 Pa, with the OM catalyst coating adding negligible extra pressure. This low-pressure drop characteristic could stabilize the gas flow within the catalytic device. The efficiency of a monolithic catalyst, however, is limited by kinetic resistance at low temperatures, internal mass transfer resistance at intermediate temperatures, and external mass transfer resistance at high temperatures. Enhancing catalyst activity can minimize kinetic resistance, while optimizing substrate design can reduce external resistance. Nevertheless, decreasing internal mass transfer resistance is more challenging due to its dependence on reactant diffusivity within the washcoat layer[22]. In Fig. 5b, the internal mass transfer resistances are quantified as a function of the monolith temperature using the low-dimensional model[30]. Notably, the internal mass transfer resistance of OM catalysts is two orders of magnitude lower than that of conventional powder-coated catalysts. This reduction in resistance facilitates enhanced mass transfer of reactant molecules within the OM catalyst, thereby exposing active sites more effectively and promoting gas-solid interactions through a much shorter diffusion path. To further validate the mass transport properties of the OM structure, catalytic activity for $C_3H_8$ oxidation over OM and powder $Pt_{SA}$/$CeZrO_2$ catalysts was evaluated at different gas hourly space velocities (GHSV), as shown in Fig. 5c. As GHSV increases from 1000 to 30,000 $h^{-1}$, the powder catalyst shows a larger performance drop ($\triangle T_{90} = 100$ °C) compared with the OM catalyst ($\triangle T_{90} = 50$ °C), highlighting the OM structure's higher tolerance to space velocity due to its lower internal mass transfer resistance.

We previously achieved efficient and stable dynamically low-coordinated $Pt_{SA}$ sites through Zr-doping-assisted atom trapping. However, the efficiency of atom trapping at high temperatures remains unstudied, and minimizing the loss of active components due to high-temperature airflow in industrial settings is a key challenge. This work first proposes a scalable strategy for designing OM catalysts to enhance atom trapping efficiency under high temperatures. To evaluate the OM structure's resistance to Pt loss, Pt loading in OM Pt/$CeZrO_2$, powder Pt/$CeZrO_2$, and commercial PtPd/CeZr-based catalysts was measured before and after aging at 800 °C and 1100 °C in flowing air (ICP data in Fig. 5d). After aging, Pt loading in the powder and commercial catalysts decreased by 62.5% and 72.5%, respectively, while the OM Pt/$CeZrO_2$ sample showed only a 25% reduction. As shown in Supplementary Fig. 29, the powder Pt/$CeZrO_2$ and commercial Pt-based catalysts lost 15% and 11% more Pt in flowing air at 800 °C compared to static air. However, the OM Pt/$CeZrO_2$ sample had similar Pt loss in both flowing and static conditions, showing better tolerance to flowing air. These results demonstrate that the abundant macropores and mesopores in the OM support can effectively capture Pt atoms, reducing Pt loss by 37.5% under extreme temperatures (Fig. 5e).

Furthermore, we compared OM $Pt_{SA}$/$CeZrO_2$, OM $Pt_{SA}$/$CeO_2$, and Powder $Pt_{SA}$/$CeZrO_2$ together, as shown in Supplementary Fig. 30. The Powder $Pt_{SA}$/$CeZrO_2$ catalyst without the OM structure experienced a 75 °C increase in $T_{90}$ (400 °C → 475 °C) after aging due to Pt loss, and the reaction rate decreased from 460 to 321 μmol/($g_{cat}$*s). The OM $Pt_{SA}$/$CeO_2$ catalyst without Zr assistance saw a significant 140 °C increase in $T_{90}$ (400 °C → 540 °C) after aging due to $Pt_{SA}$ overoxidation (6-coordinate Pt–O), with the reaction rate decreasing from 460 to 227 μmol/($g_{cat}$*s). These results demonstrate the synergistic effect of the OM structure integrated technology and Zr-assisted atom trapping technology. Figure 2d shows that OM $Pt_{SA}$/$CeZrO_2$ can withstand hydrothermal aging at 800 °C for 50 h with 10 vol% $H_2O$, maintaining over 90% propane conversion for 1000 h at 450 °C. Supplementary Fig. 15g indicates that after 2 h of thermal aging at 1100 °C, the $T_{90}$ of OM $Pt_{SA}$/$CeZrO_2$ increases slightly (from 400 °C to 460 °C) but stays below 500 °C, while the $T_{90}$ of OM $Pt_{SA}$/$CeO_2$ and two commercial Pt-based catalysts rise above 600 °C. These results demonstrate that OM $Pt_{SA}$/$CeZrO_2$ can withstand extreme conditions−800 °C for 50 h, 450 °C for 1000 h, and 1100 °C for 2 h−showing superior high-temperature stability compared to Pt-based DOC catalysts in the literature (Fig. 5f). The synergy of the OM structure and Zr doping helps resist Pt sintering, Pt overoxidation, and Pt loss, ensuring OM $Pt_{SA}$/$CeZrO_2$'s durability in extreme high-temperature oxidative environments.

We further compared the OM $Pt_{SA}$/$CeZrO_2$ with the commercial PtPd/CeZr-based catalyst. First, the commercial PtPd/CeZr has noble metal particles around 15 nm (Supplementary Fig. 15c), while the OM $Pt_{SA}$/$CeZrO_2$ maintains single-atom dispersion even after aging at 800 °C, offering better Pt sintering resistance and noble metal utilization. Moreover, under high-temperature flowing conditions, the OM $Pt_{SA}$/$CeZrO_2$ shows better resistance to Pt loss, with a 37.5% reduction in Pt loss compared to the commercial PtPd/CeZr (Fig. 5d, e). Therefore, despite having a lower Pt loading, the OM $Pt_{SA}$/$CeZrO_2$ (0.4 $g_{Pt}$/L) exhibits comparable catalytic combustion activity to the commercial PtPd/CeZr-based catalyst (0.9 $g_{Pt}$/L + 0.1 $g_{Pd}$/L). Ultimately, the oxidation activities of carbon monoxide, propylene, and nitric oxide (CO, $C_3H_6$, and NO) for both OM $Pt_{SA}$/$CeZrO_2$ and a Pt-based commercial catalyst, tested as diesel oxidation catalysts (DOCs), are shown in Fig. 5g. After aging at 800 °C for 50 h, the OM $Pt_{SA}$/$CeZrO_2$ catalyst maintains strong performance metrics: the CO light-off temperature ($T_{50}$) remains below 200 °C, the $T_{50}$ of $C_3H_6$ stays under 250 °C, and the NO conversion exceeds 30%. The raw material consumption costs of OM $Pt_{SA}$/$CeZrO_2$ and the commercial PtPd/CeZr-based catalyst were also compared, including the raw material cost of PS

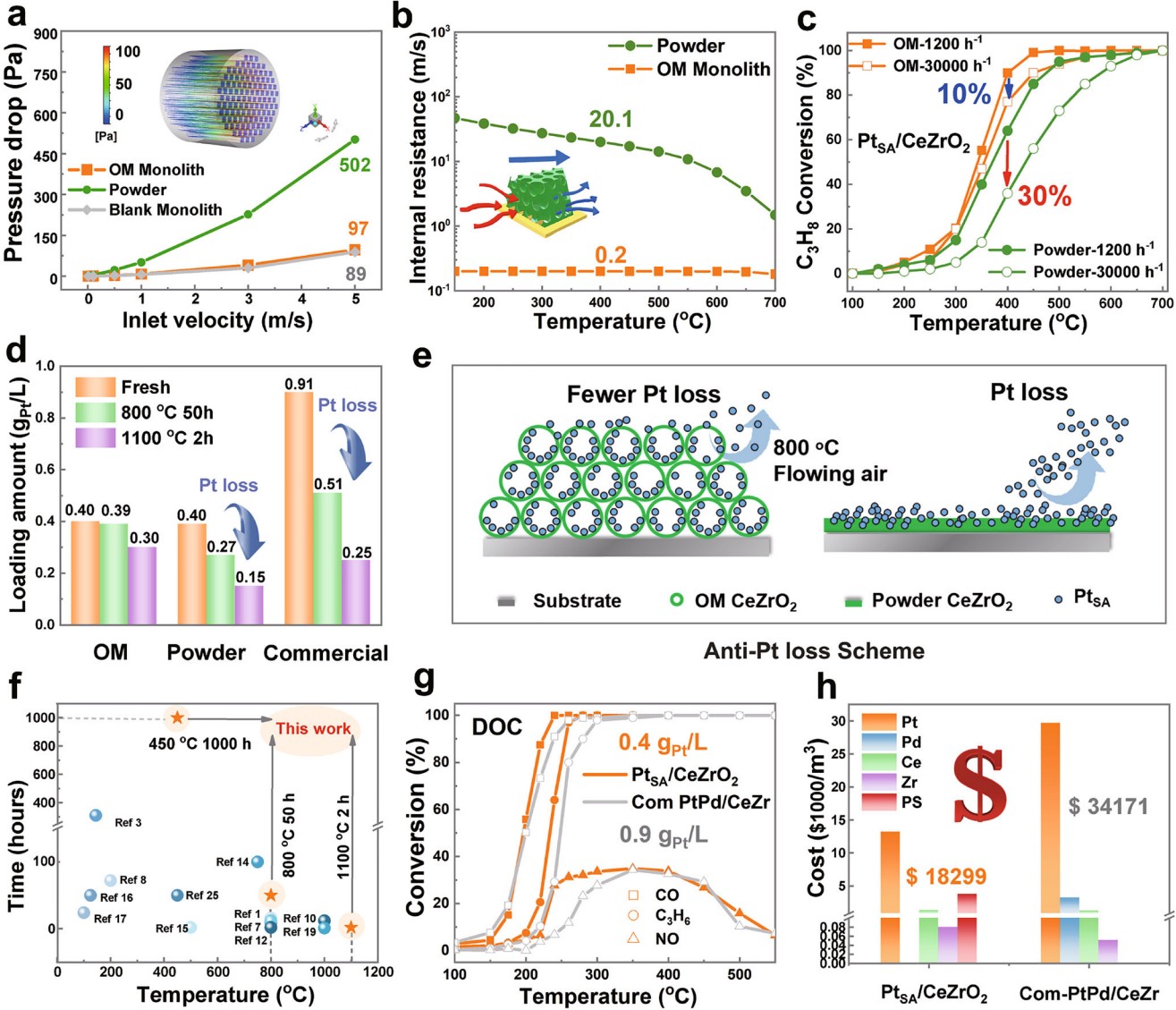

**Fig. 5 | Impact of ordered macroporous (OM) structure on internal diffusion resistance and Pt loss. a** Computational fluid dynamics (CFD) model and pressure drop at different face velocities. **b** Calculated internal resistance as a function of monolithic temperature for OM and traditional powder catalysts. **c** Conversion profiles of $C_3H_8$ oxidation for powder $Pt_{SA}/CeZrO_2$ and OM $Pt_{SA}/CeZrO_2$ catalysts under different gaseous hourly space velocity (GHSV) conditions. **d** Changes in Pt loading of catalysts after aging at 800 °C and 1100 °C (20% $O_2$, $N_2$ balance, flow rate 50 mL/min, 800 °C for 50 h/1000 °C for 2 h). **e** Schematic diagram illustrating the inhibitory effect of the OM structure on Pt loss. **f** The comparison of the tolerance temperature and time of $Pt_{SA}/CeZrO_2$ in this work with Pt-based diesel oxidation catalysts in the literature. **g** Comparison of performance with commercial diesel oxidation catalysts (DOC). **h** Material consumption costs comparison between the OM $Pt_{SA}/CeZrO_2$ integrated monolith (0.4 $g_{Pt}$/L) and commercial PtPd/CeZr-based catalysts (0.9 $g_{Pt}$/L + 0.1 $g_{Pd}$/L).

microspheres, as shown in Fig.5h and Supplementary Fig. 31. The raw material consumption cost of OM $Pt_{SA}/CeZrO_2$ was reduced by 46%, from \$34,171/m³ to \$18,299/m³. While OM $CeZrO_2$ support coatings are established catalytic materials, this study develops the first one-step synthesis protocol and enables industrial-scale fabrication on 3.4 L cordierite substrates. These findings indicate that the OM support not only promotes mass transfer and diffusion of reactants but also mitigates Pt loss under high temperature, thereby enhancing the long-term durability and performance of the $Pt_{SA}/CeZrO_2$ catalyst in industrial settings.

## Discussion

The outstanding durability of Pt isolated on an ordered macroporous (OM) $CeZrO_2$ can be attributed to its enhanced resistance to Pt sintering, Pt overoxidation and Pt loss: (1) Following aging at 800 °C, Pt nanoparticles transform into isolated Pt single atoms ($Pt_{SA}$), effectively

preventing Pt sintering and boosting Pt utilization efficiency through a Zr-assisted atom trapping method. (2) Zr doping reduces the Pt–O coordination number from 6 to about 4 during high-temperature atomization. This decrease in coordination leaves more $d$ orbitals of the low-coordinated $Pt_{SA}$ site available, enhancing its capacity to adsorb and activate reactants in high-temperature oxidative environments. The Zr-stabilized dynamic low-coordinated $Pt_{SA}$ configuration was demonstrated as the active form in oxidation reaction at high temperatures. (3) A novel, scalable integrated OM support design strategy was introduced, effectively reducing Pt loss by 37.5% under extreme-temperature conditions.

In summary, an ultra-stable, dynamically low-coordinated $Pt_{SA}/CeZrO_2$ ordered macroporous structure industrial-scale integrated monolithic catalyst was constructed. This catalyst maintained a high conversion (92% at 450 °C) even after enduring 50 h of hydrothermal aging at 800 °C with 10 vol% $H_2O$. Additionally, with half the amount

of Pt-group metals compared with a commercial PtPd/CeZr-based diesel oxidation catalyst ($0.4\,g_{Pt}/L$ vs $0.9\,g_{Pt}/L + 0.1\,g_{Pd}/L$), the $Pt_{SA}/CeZrO_2$ catalyst achieved a lower $T_{90}$ conversion temperature for propane oxidation (400 vs 450 °C). Crucially, the precise tailoring of $Pt_{SA}$ coordination structure and scalable integration of OM $CeZrO_2$ were achieved over a 3.4-liter commercial monolith to directly meet industrialization requirements. This work revealed the dynamic evolution of $Pt_{SA}$ configuration under oxidative conditions and presented a dual-scale design strategy to address three key deactivation issues—Pt sintering, Pt overoxidation, and Pt loss—thereby enabling the development and scaling of robust catalytic converters for high-temperature heterogeneous Pt-based catalysts.

## Methods

### Catalyst preparation

**Scalable synthesis of OM $CeZrO_2$ catalysts.** We employed a template-directed method[28,31] to grow ordered macroporous $Ce_{0.8}Zr_{0.2}O_2$ (Abbreviated as OM $CeZrO_2$) on a monolithic honeycomb cordierite substrate, enabling scalable fabrication (as shown in Supplementary Figs. 2 and 3). Initially, a well-optimized blend of polystyrene (PS) microsphere dispersion and metal precursor sol was prepared as described in the Supplementary Method. Subsequently, the honeycomb cordierite substrates were vertically immersed in the PS microsphere and sol mixture for 30 min. After soaking, these substrates were subjected to a vacuum (about −1 kPa) to eliminate excess solution and improve the uniformity of the sol-gel attachment, as shown in Supplementary Fig. 3b. Next, a rapid drying process was conducted at 150 °C for 2.5–3 min to ensure uniform adhesion of the PS spheres and sol onto the substrate surface, followed by thorough drying at 50 °C for above 10 h. This drying method ensures the uniform adhesion of the sol on the substrate surface while preventing the PS microspheres from prematurely degrading before the $CeZrO_2$ crystallizes. It is important to pre-adjust the oven temperature before placing the samples inside and precisely control the drying time in the first step to be between 2.5 and to 3 min. These steps were repeated twice to ensure coating uniformity and sufficient loading of the catalyst. Subsequently, calcination at 250 °C for 2 h was performed to remove the PS template, followed by further calcination at 550 °C for 4 h to facilitate the formation of the OM $CeZrO_2$ solid solution structure. It is important to note that if the temperature is directly raised to 550 °C, the sudden decomposition of the PS template and the shrinkage of the metal oxide will occur simultaneously, causing the pore walls to collapse due to stress concentration. Stepwise calcination allows the template decomposition and material densification to occur in stages, ensuring the mechanical stability of the pore structure. For comparative purposes, OM $CeO_2$ was synthesized using a similar protocol. The synthesis of the powder $Ce_{0.8}Zr_{0.2}O_2$ sample, which lacks an OM structure, was carried out without the addition of PS sphere templates but followed the same procedural steps. Detailed chemical materials are provided in the Supplementary Method.

**Pt-loading.** Pt is loaded using a simple microwave-assisted dip-coating method[14]. Initially, the integral OM $CeZrO_2$ is immersed in a 75% ethanol solution of Pt salt, $Pt(NH_3)_4(NO_3)_2$, with a certain concentration of 1.7 g/L. After being taken out, it is microwave-dried for 30 s (1000 W) and then calcined at 550 °C for 2 h to obtain $Pt_{NP}/CeZrO_2$. The rapidity of microwave drying ensures the uniformity of Pt loading. Subsequently, $Pt_{NP}/CeZrO_2$ is aged in a tubular furnace under an air atmosphere at 800 °C for 50 h (50 mL/min), during which the Pt nanoparticles transform into isolated Pt single atoms, resulting in $Pt_{SA}/$ OM $Ce_{0.8}Zr_{0.2}O_2$ (Abbreviated as $Pt_{SA}/CeZrO_2$). The comparison samples $Pt_{SA}/$OM $CeO_2$ (Abbreviated as $Pt_{SA}/CeO_2$) and Pt/Powder $CeZrO_2$ are prepared using the same method.

### Catalytic activity evaluation

The catalytic oxidation activity of the catalyst samples for $C_3H_8$ is assessed using a simulated fixed-bed reactor with a quartz tube of 23 mm diameter. The required volume of the integral catalyst sample ($1.3 \times 1.3 \times 1.5\ cm^3$) is loaded into the middle of the quartz tube, and the amount of catalyst used for testing is kept consistent. The composition of the reaction gas is 3000 ppm $C_3H_8$, 12% $O_2$, and $N_2$ balance, with a total flow rate of 50 mL/min, corresponding to a volumetric space velocity of about 1100 $h^{-1}$. The temperature programming process is as follows: heating from 200 °C to 550 °C at 5 °C/min with a 50 °C interval per stage, each temperature stage is maintained for 30 min. The composition of the exhaust gas from the catalytic reaction is monitored in real-time online using gas chromatography (GC, 9790, Taizhou), with $N_2$ as the carrier gas. The conversion of $C_3H_8$ is calculated as follows, Eq. (1), where $X_{propane}$ represents the conversion of propane, $F_{propane,in}$ represents the inlet concentration of propane, and $F_{propane,out}$ represents the outlet concentration of propane after the reaction. $F_{propane,in} - F_{propane,out}$ represents the concentration of propane converted during the reaction. $T_{50}$ and $T_{90}$, respectively, represent the temperatures corresponding to 50% and 90% conversion of propane.

$$X_{\text{propane}} = \frac{F_{\text{propane, in}} - F_{\text{propane, out}}}{F_{\text{propane, in}}} \times 100\% \qquad (1)$$

To compare the kinetic characteristics of catalysts, the mass-specific rate of $C_3H_8$ catalytic oxidation is calculated throughout the conversion process. In Eq. (2), $r$ represents the mass-specific rate of the catalyst, $X$ [%] represents the conversion of $C_3H_8$ at a certain temperature, $F$ [L/s] represents the total flow rate of the reaction gas, and $m$ represents the mass of the catalyst.

$$r = \frac{(1\% \times X[\%] \times F[L/s])}{(22.4\,L/mol \times m[g])} [mol/(g\,s)] \qquad (2)$$

The apparent activation energy when the $C_3H_8$ conversion is below 10% is calculated. In the formula, Ea represents the reaction activation energy, A represents the pre-exponential factor, a and b are constants, $[C_3H_8]$ and $[O_2]$ represent the concentrations of propane and oxygen, respectively. The calculation Eq. (3) for the Arrhenius plot is as follows:

$$\ln r = \frac{Ea}{RT} + \ln A + A\ln[C_3H_8] + b\ln[O_2] \qquad (3)$$

The calculation Eq. (4) for turnover frequency (TOF) is as follows. Here, M represents the molar mass of Pt; σPt denotes the platinum content, which is determined by ICP-OES.

$$TOF = \frac{r \times M}{\sigma_{Pt}} \qquad (4)$$

**DOC (CO, NO, $C_3H_6$) catalytic oxidation activity Test.** The catalyst sample ($4 \times 4 \times 20$ mm) is wrapped in quartz wool and loaded into a quartz tube (inner diameter $\varphi = 8$ mm) within a tubular furnace for catalytic performance testing. The DOC activity test conditions are as follows: 1500 ppm CO, 600 ppm NO, 900 ppm $C_3H_6$, 12% $O_2$, with nitrogen as the balance gas, and total flow rates of 340 mL/min. The volumetric space velocity is GHSV = 7000 $h^{-1}$. The catalyst is subjected to programmed heating via the tubular furnace's setup. The entire reaction process is monitored in real-time online using a Thermo-NICOLET iS50 FT-IR Fourier Transform Infrared Spectrometer to analyze the exhaust gas. The concentrations of the three gases in DOC are represented by the peak area of the characteristic peaks in the

spectrum. The conversion Eqs. (5–7) for propylene, carbon monoxide, and nitrogen monoxide are as follows:

$$XC_3H_6 = \frac{(C_{(C_3H_6)in} - C_{(C_3H_6)out})}{C_{(C_3H_6)in}} \times 100\% \qquad (5)$$

$$X_{CO} = \frac{(C_{(CO)in} - C_{(CO)out})}{C_{(CO)in}} \times 100\% \qquad (6)$$

$$X_{NO} = \frac{(C_{(NO)in} - C_{(NO)out})}{C_{(NO)in}} \times 100\% \qquad (7)$$

## Catalyst characterization

The catalysts' morphology and structure were analyzed using various electron microscopy techniques. Scanning electron microscopy (SEM) images were captured employing the FEI Teneo low vacuum SEM and the JEOL 6335 F field emission SEM, both operated at 10–20 kV. For transmission electron microscopy (TEM), inclusive of bright field and high angular annular dark field (HAADF) images, as well as energy-dispersive X-ray spectroscopy for compositional mapping, the imaging was conducted utilizing the FEI Talos STEM and Tecnai F30 STEM. Regarding aberration-corrected HAADF STEM imaging, two distinct STEM/TEM systems were employed. Firstly, the JEOL JEM 2200FS, coupled with a CEOS (Center for Electron Optics and Spectroscopy) probe corrector, offers a nominal image resolution of 0.07 nm. Secondly, the JEOL JEM-ARM200F, also equipped with a CEOS probe corrector, ensuring a guaranteed resolution of 0.08 nm.

The X-ray absorption fine structure data were obtained from the BL14W1 station at the Shanghai Synchrotron Radiation Facility (SSRF), which operates with a storage ring energy of 3.5 GeV. The beamline utilizes a monochromatic X-ray source with a specific energy range appropriate for the absorption edge of Pt. The experimental setup was calibrated by simultaneously measuring the spectrum of a reference Pt foil to ensure accurate data for each sample. To ensure proper calibration and reproducibility, we carefully controlled the experimental conditions. The beamline was operated under a vacuum environment to minimize the influence of atmospheric scattering. Subsequent to data collection, the X-ray absorption spectroscopy (XAS) data underwent processing and fitting procedures using the ATHENA module in the IFEFFIT software packages. The raw data were first corrected for background, energy shifts, and normalization. For the Pt L3 edge data, the fitting process involved the first coordination shell based on Fourier-transformed k3-weighted χ(k) functions within the k-range of 3.0–13.9 Å$^{-1}$. This range was selected based on the characteristics of the Pt L3 edge and the need to focus on the first coordination shell, as it provides the most relevant structural information. The coordination numbers were determined by fixing the amplitude reduction factor $(S_0^2)$, which was derived from fitting the reference Pt foil. This approach was employed to maintain consistency in the analysis and improve the reliability of the results. In cases where the number of independent data points was limited due to experimental constraints, the Debye–Waller factor ($\sigma^2$) was set to a reasonable value of 0.003.

The crystalline structure of the sample was analyzed using X-ray diffraction (XRD) with CuKα radiation (wavelength = 1.540598 Å) on a BRUKER AXS D5005 instrument. The scan was performed in the 2θ range of 10°–90°, with a step size of 0.02° and a scan rate of 0.5° per minute. This setup was chosen to ensure sufficient resolution in detecting peaks corresponding to various crystal planes. The XRD patterns were analyzed to determine the phase composition. Raman spectra were obtained using a Horiba LabRAM HR Evolution equipped with an Ar+ laser (wavelength = 325 nm, power = 100 mW) over a wavenumber range of 200–1000 cm$^{-1}$. Spectra were acquired in three consecutive runs, with 20-s exposure intervals between each scan to minimize laser-induced heating effects. The laser spot size was approximately 1 μm to ensure high spatial resolution.

Specific surface area via Brunauer-Emmett-Teller (BET) analysis was determined using Quantachrome NOVA 1000 Gas Sorption Analyzer and Micromeritics ASAP 2020 physisorption analyzer. Samples were degassed at 200 °C for 6 h before measurement. Mercury Intrusion Porosimetry (MIP) measurements were conducted using a Micromeritics AutoPore IV 9500 apparatus, covering a pressure range of 0–10,000 psia. The technique was used to measure the pore size distribution and total porosity of the sample. Prior to measurement, samples were degassed at 200 °C for 6 h under vacuum to remove surface contaminants.

CO adsorption was monitored using a Nicolet iS50 FT-IR spectrometer (Thermo Scientific) equipped with a diffuse reflectance attachment. Prior to CO exposure, samples were purged with nitrogen at 200 °C for 0.5 h to remove adsorbed impurities. Afterward, the samples were exposed to CO gas (5000 ppm) at room temperature/400 °C for 40 min. DRIFT spectra were collected after purging the sample under vacuum for 1 h to remove gaseous CO peaks, ensuring that only the adsorbed CO signals were detected. Quantitative analysis of metallic elements was performed via Inductively Coupled Plasma Optical Emission Spectrometry (ICP-OES) using an Agilent ICP-OES 730 instrument. Samples were dissolved in a mixture of hydrofluoric acid and nitric acid at high temperatures (about 120 °C) to break down the matrix. The resulting liquid was then analyzed for its elemental composition. Calibration standards were prepared using known concentrations of the target elements.

X-ray photoelectron spectra (XPS) were obtained using a Thermo ESCALAB 250XI equipped with Al Kα (hv = 1486.6 eV) for excitation. Binding energies were adjusted with reference to C 1 s to 284.8 eV. The center of gravity calculation for valence band spectra was determined within the range of 0--10 eV, utilizing the density of states. Oxygen Temperature-Programmed Desorption (O$_2$-TPD) experiments were conducted on all catalysts using a Chemisorb tp-5080 (Xianquan, Tianjin). Prior to each experiment, catalysts (50 mg) underwent pretreatment in flowing 5% O$_2$/N$_2$ (50 mL/min) at 400 °C for 1 h, followed by cooling to room temperature and a subsequent N$_2$ purging for 30 min. Electron Paramagnetic Resonance (EPR) spectra were recorded with a Bruker EMX EPR spectrometer (Billerica, MA) at 77 K in the X-band. To identify oxygen vacancies, EPR measurements were carried out in high vacuum after pretreating samples in N$_2$ at 400 °C to eliminate surface-adsorbed oxygen species. Additionally, for discerning surface-adsorbed oxygen species and tracking their evolution, EPR measurements were conducted after exposure to air.

The Near-Ambient Pressure X-ray Photoelectron Spectrometer (NAP-XPS) measurements were performed at the Shanghai Synchrotron Radiation Facility beamline, using a differentially-pumped Al Kα source (Specs model XR50) with a photon energy of 1486.6 eV. Emitted photoelectrons (and Auger electrons) were detected with a near-ambient pressure hemispherical analyzer (Specs model Phoibos 150), mounted in a custom-designed system capable of measuring XPS under sample gas pressures up to 10 Torr. The specific testing procedure is as follows: The sample was placed into the sample chamber and sputter-cleaned for 30 min. The temperature was set to 673 K, and spectra were collected under two conditions: UHV and C$_3$H$_8$ + O$_2$ (with a partial pressure of 1/10). Full spectra were acquired, along with specific C 1 s and Pt $4f$ spectra.

The in situ analysis of propane oxidation through Diffuse Reflectance Infrared Transform Spectroscopy (DIRFTS) was conducted using a THERMO/Nicolet iS50 spectrometer equipped with an MCT detector, coupled with a Praying Mantis High-Temperature Reaction Chamber featuring ZnSe windows (Hidden). In preparation for the in situ DIRFTS examination, the samples underwent incremental heating to 400 °C

under a 20% $O_2/N_2$ flow for 15 min, followed by purging with $N_2$ at 400 °C for an additional 30 min. Simultaneously, background measurements were obtained at 400 °C under $N_2$ flow. Subsequently, the pre-treated catalyst surface was exposed to a sequential flow of 3000 ppm $C_3H_8/N_2$ and 3000 ppm $C_3H_8/20\%$ $O_2/N_2$, each held for 30 min at 200 °C. The spectra were collected concurrently, with a total gas flow of 50 mL/min.

**Density functional theory (DFT) calculations.** DFT, as implemented in the Vienna ab initio simulation package (VASP), was used to carry out the calculations presented here[32]. The projector augmented wave (PAW) method[33] was used to treat the effective interaction of the core electrons and nucleus with the valence electrons, while exchange and correlation were described using the Perdew-Burke-Ernzerhof (PBE) functional[34]. The kinetic energy cutoff for the plane-wave basis set is 400 eV, and all atoms are allowed to relax until the force and energy are less than 0.05 eV Å$^{-1}$ and $10^{-4}$ eV, respectively.

**Model construction.** The unit cells were built using $(2 \times 2 \times 1)$ supercells of $CeO_2$. For our studies, we chose the most thermodynamically stable (111) surface of $CeO_2$, with a 15 Å vacuum gap between the slabs. To create a $Ce_{0.8}Zr_{0.2}O_2$ model, we substituted 1/5 of the Ce atoms with Zr, distributed evenly throughout the $CeO_2$ structure. Monoatomic Pt can be introduced in two ways: lattice doping and surface adsorption. The loading position of a single-atom Pt was optimized, and $Pt_{SA}/CeO_2$ and $Pt_{SA}/CeZrO_2$ models were constructed.

The oxygen vacancy formation energy was calculated by Eq. (8):

$$E_f = E_{tot-Ov-}E_{tot} + 1/2E_{O2} \qquad (8)$$

Where $E_{tot}$ and $E_{tot-Ov}$ are the energy of the perfect and the corresponding oxygen-defective $Ce_{0.8}Zr_{0.2}O_2$ (111) surface, $E_{O2}$ corresponds to the total energy of the $O_2$ molecule in the gas phase.

The adsorption energies of $O_2$ and $C_3H_8$ were calculated as follows Eqs. (9–10):

$$E_{ads}(O_2) = E_{tot} - E_{surf} - E_{O2} \qquad (9)$$

$$E_{ads}(C_3H_8) = E_{tot} - E_{surf} - E_{C_3H_8} \qquad (10)$$

Where $E_{tot}$ is the total energy of the $O_2/C_3H_8$ after adsorption on $Ce_{0.8}Zr_{0.2}O_2$ surface, $E_{surf}$ is the sum of the surface energy. Climbing images nudged elastic band (CL-NEB) calculations were performed to locate the TS with the assistance of special functional scripts embedded in the transition state tools (VTST) software package compiled in VASP.

Differential charge can be used to study the valence electron transfer. The charge transfer caused by metal doping is given by the following Eq. (11):

$$\Delta\rho = \rho(Pt - support) - \rho(support) - \rho(Pt) \qquad (11)$$

where $\rho$(Pt-support), $\rho$(Pt), and $\rho$(support) are the doping model, the corresponding single Pt ions, and the total support ($CeO_2$ or $CeZrO_2$) charge density, respectively.

**Molecular dynamics analysis**
We detail the outcomes of our ab initio molecular dynamics (AIMD) studies, utilizing density functional theory (DFT) to investigate the dynamics and kinetics of catalysts in the $C_3H_8$ oxidation process. These AIMD simulations were executed in a canonical ensemble (NVT) framework, employing a Nose-Hoover thermostat, with a temporal resolution of 1 fs for each step. The simulations initiated from geometrically optimized structures at a starting temperature of 673 K.

They were subjected to a thermal annealing process over a duration of 1 ps, during which the potential energies were observed to stabilize progressively. Subsequently, trajectories extending up to 1 ps were compiled and analyzed.

**Computational fluid dynamics (CFD) method.** The flow field of a monolithic catalyst was simulated using ANSYS Fluent 2020[35]. The cylindrical catalyst ($d = 23$ mm, $h = 10$ mm) was discretized into 90,000 tetrahedral cells. The SIMPLEC algorithm and second-order windward scheme were used to calculate the average pressure drop from inlet to outlet. Boundary conditions were applied to each surface[36].

**Low-dimensional model constructing.** To quantify internal mass transfer resistances, a low-dimensional model for a single washcoat monolith channel, based on Joshi et al.'s work, was employed[30]. This model simplifies the convection–diffusion-reaction equations by averaging in the transverse direction, yielding an overall mass transfer coefficient for transverse diffusion and reaction. Key assumptions are: (a) laminar flow is assumed within the monolith channel, characterized by a Reynolds number of less than 2300. (b) Axial diffusion and heat conduction in the fluid phase are considered negligible relative to convection, with justification based on the Prandtl and Schmidt numbers being small for typical monolith flows. and (c) constant physical properties, including viscosity and diffusivity, were assumed over the temperature and concentration ranges explored. The model includes transport equations for species mass balances in the gas phase, washcoat phase, and gas-solid interface.

## Data availability
The source data generated in this study are provided in the Supplementary Information/Source Data file. Source data are provided with this paper.

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

## Acknowledgements

The authors are grateful for the financial support from National Natural Science Foundation of China (No. 22076060, 22376074, 22376075), The Recruitment Program of Global Young Experts start-up funds, The Program of Introducing Talents of Discipline to Universities of China (111 program, B17019), Natural Science Foundation for Distinguished Young Scholars of Hubei province (NO. 2021CFA085), the Cultivation Program of Wuhan Institute of Photochemistry and Technology (GHY2023KF010, GHY2023KF002), Longyan Major Science and Technology Project (2022LYF1006), Wuhan Institute of Photochemical Technology incubation project(GHY2023FH03), Wuhan Knowledge Innovation Special Basic Research Project (2023020201010123), the Central China Normal University project (CCNU24JCPT017, 30106220497).

## Author contributions

Y.G. and Z.L. conceived and supervised the research. B.Z. (Baojian Zhang) and Y.G. designed the experiments and wrote the manuscript. B.Z. (Baojian Zhang) performed most of the catalyst evaluation, data collection, and analyses. R.L. and B.Z. (Biluan Zhang) carried out the DFT calculations and AIMD simulations. L.L. synthesized the catalysts. W.Y. performed TEM experiments. W.G. and B.C. carried out the CFD calculations. P.L. and S.Z. helped with the characterization analysis. J.Y. and J.W. provided suggestions on the project design. All the authors contributed to the overall scientific interpretation and edited the manuscript.

## Competing interests

The authors declare no competing interests.
