## [Transparent Peer Review file · Nature Communications]

Ultra-stable Low-coordinated PtSA/CeZrO₂ Ordered Macroporous structure Integrated Industrial-scale Monolithic Catalysts for High-Temperature Oxidation

Corresponding Author: Professor Yanbing Guo

Version 0:

Reviewer comments:

Reviewer #1

(Remarks to the Author)

Herein isolated Pt single atom (PtSA) was constructed on Ce_{0.8}Zr_{0.2}O₂ support with an ordered macroporous (OM) nanostructure. The dynamic low-coordinated PtSA possess more free d-orbital electrons that are not excessively occupied by Pt-O bonds, which enhanced the activation of the propane C-H bonds and enables peroxide species to remain active at high temperatures. Consequently, the PtSA/CeZrO₂ catalyst exhibited a high activity and stability in propane oxidation, even superior than commercial diesel oxidation catalysts Pt/Al₂O₃. Furthermore, PtSA/CeZrO₂ was integrated into commercial cordierite monolith for potential commercial applications. The research results are interesting, and propose a meaningful routine to design and synthesize alkane oxidation catalysts with excellent catalytic performances. However, the innovation of the paper should to be further strengthened. Based on overall considerations, I recommend revising and resubmitting for reconsideration for publication. The detail comments are as follows:

- (1) As far as I know, atomic trapping method has been proposed and widely applied to prepare single atom catalysts on CeO₂-based support, since Abhaya K. Datye reported thermally stable single-atom platinum-on-ceria catalysts via atom trapping (Science, 2016, 353, 150-154). Subsequently, the mechanism of trapping was also proposed (ACS Catalysis, 2019, 9, 3978-3990). More significantly, locate platinum single sites on CeO₂ and the variation of the active state under reaction conditions were successfully addressed (Nature Catalysis, 2020, 3, 824-833). Therefore, it is essential to distinguish where is the platinum single site location on OM nanostructured Ce_{0.8}Zr_{0.2}O₂? What's the effect of Zr doping on the Pt location, besides coordinated configuration of Pt? Why did select of the composition of Ce_{0.8}Zr_{0.2}O₂ support? If varying the molar ratio of Ce/Zr, can single atom dispersion still be achieved?
- (2) The effect of Zr doping on the thermal stability of CeO₂ and the corresponding mechanism were entirely revealed early, especially in three-way catalysts for vehicle exhaust purification. Therefore, the idea of applying thermally stable CeZrO₂ instead of CeO₂ is a natural state of being. The conclusion about the effect of zirconium doping on CeO₂ lacks sufficient novelty. Please provide more discussion about the novelty of Zr doping in the related section.
- (3) Where did commercial diesel oxidation catalysts Pt/Al₂O₃ come from? In fact, the commercial diesel oxidation catalysts also include Pt/CeZrO_x catalysts. Therefore, the comparison of the activity between PtSA/CeZrO₂ and commercial Pt/CeZrO_x catalysts should be conducted. On the other hand, the activity of PtSA/CeZrO₂ should be also compared with noble metal catalysts in references.
- (4) What's aging atmosphere conditions for the experiments of Pt loss? If it was conducted under static conditions at high temperature, how is the Pt loss for the PtSA/CeZrO₂ catalyst after aging under reactive conditions?
- (5) This paper solved the problems of Pt sintering, overoxidation and loss by preparing the PtSA/CeZrO₂ catalyst. However, in actual applications of DOC, the catalyst's resistance to sulfur poisoning also needs to be considered. How is the sulfur resistance of this catalyst?

Reviewer #2

(Remarks to the Author)

The article presents a significant advancement in the field of catalysis, particularly in the development of platinum (Pt)

single-atom catalysts (SACs) supported on ordered macroporous (OM) structures. The findings have important implications for industrial applications, especially in emission control technologies, as they address critical challenges such as catalyst deactivation, which is a major concern in high-temperature catalytic processes. The integration of the catalyst into a commercial monolith demonstrates its practical applicability, enhancing its potential impact on reducing environmental pollutants.

The research addresses three major deactivation mechanisms (Pt sintering, Pt overoxidation, and Pt loss) that limit the efficiency and durability of catalytic converters. By introducing a novel Zr-doping strategy and demonstrating the scalability of the catalyst design with the catalyst able to maintain high conversion rates even under extreme conditions.

Although the article is well-structured, with a clear introduction, methodology, results, and discussion sections, some sections, particularly those detailing complex experimental techniques and results, may benefit from simplification or additional explanations to enhance understanding for a broader audience. Visual aids such as diagrams and charts effectively complement the text, but further clarification of specific terms and concepts could improve accessibility.

Specifically,

- 1) Some sections of the article contain complex jargon (first time - likely coined by authors) and intricate descriptions that may hinder understanding and limit the article's accessibility to a broader audience.
- 2) While figures and diagrams are included, some of them may require clearer labeling and explanations. The significance of certain results could be better illustrated through visual means.
- 3) The discussion section, although comprehensive, could benefit from a more in-depth exploration of the implications of the findings. For example, discussing potential real-world applications and scenarios where this catalyst could be implemented would enhance the relevance of the study.
- 4) The article could provide a more thorough comparison with existing literature. While it mentions previous work, a more detailed analysis of how this research builds upon or diverges from prior studies would strengthen the argument for its novelty.
- 5) Certain methodological aspects, particularly regarding the characterization techniques, could be elaborated upon. Providing more detail on the experimental setup, conditions, and rationale for chosen methods would enhance reproducibility and understanding.
- 6) The focus on a dynamic low-coordinated Pt single-atom configuration is technically significant. This configuration enhances the reactivity of the catalyst by allowing more unoccupied d-orbitals, which facilitates the activation of reactants, particularly under high-temperature conditions. It would be good if a modeling could be done to confirm this argument.

These are some suggestions that could further enhance the quality of the submission.

- 7) While the catalyst demonstrates high conversion rates and stability after 50 hours of hydrothermal aging at 800 °C, the article does not provide extensive data on long-term stability over extended operational periods. The performance under real-world conditions over months or years is crucial for industrial applications. It is important to demonstrate that the test conditions are sufficient to make this extrapolation.
- 8) The article does not sufficiently address the reproducibility of the catalyst's performance across different batches or synthesis methods. Variability in catalyst preparation could lead to inconsistent results. This will require a more explicit disclosure of the synthetic approach.
- 9) Although the article identifies and addresses key deactivation mechanisms (sintering, overoxidation, and loss), there is limited empirical evidence or data presented to demonstrate how these mechanisms specifically affect the catalyst over time in practical applications. Operando techniques could establish SAR for the catalyst.
- 10) While the article discusses the dynamic low-coordinated configuration of Pt single atoms and their reactivity, there is a lack of detailed experimental data supporting the claims about the behavior of these active sites under varying operational conditions.
- 11) Finally, the article does not discuss the environmental impact or economic feasibility of using this catalyst in industrial applications. Factors such as the cost of materials, energy consumption during synthesis, and potential recycling methods for spent catalysts are not addressed.

Reviewer #3

(Remarks to the Author)

Reviewer #4

(Remarks to the Author)

Version 1:

Reviewer comments:

Reviewer #1

(Remarks to the Author)

I have re-reviewed the manuscript and find that all the comments have been well-addressed. I have no further comments and think it can be accepted as is.

Reviewer #2

(Remarks to the Author)

I thank the authors for fully addressing my concerns and carrying out the advices.

Reviewer #3

(Remarks to the Author)

The authors have significantly improved the work following reviewers' comments and the overall quality of this work is enhanced. The efforts to respond to reviewers have triggered meaningful and insightful explorations and remarkable new results. There are still several points that remain unclear as follows. I think this paper can be considered for publication after carefully addressing these comments.

1. Fig. 3c and d. I am still having trouble understanding the inlet cartoons showing the Pt structure. At first glance, both Pt structures seem symmetric (although an asymmetric nature is mentioned in the text), and the difference between Pt-O(Ce) and Pt-O(Zr) is not clear. Further clarification is required.
2. It is interesting that Pt can still form single atoms with a high Zr doping content in PtSA/Ce_{0.6}Zr_{0.4}O₂. Then, with a wide range of Zr doping from 0 to 40%, why does Pt/Ce_{0.8}Zr_{0.2}O₂ show the best performance?
3. The comparison of PtSA/CeZrO₂ and PtPd/CeZr in Fig. 2e and Fig. 5f is really remarkable. Can you reiterate better why PtSA/CeZrO₂ shows amazingly better performance compared to the commercial sample? I fully understand the exact recipe for the industrial catalyst is confidential. However, there should be reasons behind its complicated composition with the presence of multiple elements Pt, Pd, Ce, Zr, Y, Ba, Al, and Si. Each element may play a role here in enhancing the performance (e.g. Pd assists with low-temperature performance and other metals act as promoters). The most interesting results come from Fig. 2e, in which the catalyst is in powder form without integration of the OM structure based on my understanding. The performance of PtSA/CeZrO₂ is better than the commercial catalyst at all temperatures. Why does PtSA/CeZrO₂ (even without the OM structure) outperform the commercial sample in all aspects with a much simpler composition and structure?
4. It is interesting to see the activity of PtSA/CeZrO₂ is even higher in the presence of SO₂, according to Supplementary Fig. 16 (both a and b). Any explanation for this?
5. In Fig. 3g, I am having trouble understanding how the trajectory for the Pt coordination number aligns with scattered data points in the AIMD simulation. Could you please clarify this more?
6. Supplementary Fig. 7 shows the particle size of Ce-Zr ranging from 5-30 nm. However, these particles seem to be clearly separated from each other based on the interpretation of panel c. Does it really support the original claim that mesopores stemmed from "the spaces" between Ce-Zr solid solution nanoparticles? Or are Ce-Zr particles with porous structure? It is still not clear here.
7. Despite the valid argument from the authors, the claim still seems an overstatement: "the efficiency of the atomic trapping method for preparing PtSA catalysts was investigated for the first time". There should be more descriptive language in this sentence to narrow down and better frame the contributions of this work, and make sure that "first time" holds true here.

Reviewer #4

(Remarks to the Author)

I co-reviewed this manuscript with one of the reviewers who provided the listed reports. This is part of the Nature

Communications initiative to facilitate training in peer review and to provide appropriate recognition for Early Career Researchers who co-review manuscripts.

Version 2:

Reviewer comments:

Reviewer #3

(Remarks to the Author)

Authors have addressed all my comments and the paper is recommended to be accepted for publication.

Reviewer #4

(Remarks to the Author)

Main revisions in the revised manuscript

1. We included a comparison with previous work in the Introduction, Results and Discussion sections, emphasizing the innovative contributions of this paper.
2. We separated the discussion of Pt_{SA}/CeZrO₂ and catalyst engineering by OM structure, reorganized the Abstract and Discussion sections, and clarified the synergistic effect of dual-scale structural control for a clearer storyline.
3. We revised the Title and Discussion, defined the application scope more clearly, and made it easier to understand.
4. We added stability test data for Pt_{SA}/CeZrO₂ at 450°C for 1000 hours (**Fig. 2d**) to better assess the catalyst's durability and potential applications.
5. We included *in situ* CO DRIFTS and *in situ* NAP XPS data (**Supplementary Fig. 1** and **Fig. 3e-f**) to better understand the catalyst's anti-sintering and anti-overoxidation mechanisms under reaction conditions and clarify its high-temperature stability.
6. We added Pt loss data after high-temperature aging of the catalyst in static air (**Supplementary Fig. 1**) and compared it with Pt loss in flowing gas conditions to better explain the OM structure's role in preventing Pt loss (**Page 14, Line 334-341**). A schematic was added to simplify understanding (**Fig. 5e**).
7. We included data on how the Zr doping ratio affects catalytic performance (**Supplementary Fig. 1**) and re-analyzed STEM, DFT, and EXAFS results to explain the impact of Zr on Pt_{SA} loading position (**Page 14, Line 334-341**).
8. We compared the catalytic oxidation activity (T₉₀, TOF in **Supplementary Table 1 and Table 2**) and high-temperature stability (**Fig. 5f**) with literature-reported catalysts and included comparisons with commercial catalysts (**Fig. 5g-h**).
9. We added performance data from different catalyst batches and error analysis (**Supplementary Fig. 1**), and clarified the key steps and principles in the catalyst preparation process (**Page 14, Line 334-341**) to improve reproducibility.
10. We discussed material costs, energy consumption during synthesis, and potential recycling methods for spent catalysts (**Supplementary Fig. 1** and **Fig. 5h**).

11. We included snapshots from AIMD simulations of the Pt_{SA}/CeZrO₂ model at different time intervals in **Supplementary Fig. 9** to illustrate the dynamic changes in the Pt_{SA} coordination structure during high-temperature reactions.
12. We included new SO₂ resistance performance data (**Supplementary Fig. 9**) to highlight the improved stability of the catalyst under harsh conditions. Similarly, we added mechanical stability test data (**Supplementary Fig. 9**) to demonstrate the robustness of the OM structure in real-world applications.
13. We enhanced the manuscript's clarity and reproducibility by refining the language, correcting grammar and spelling, clarifying first-time terms, and improving the catalyst preparation and characterization sections, along with adding explanations for some Supplementary data.

Point-to-point response to the reviewers' comments

Reviewer: 1

General comments: Herein isolated Pt single atom (Pt_{SA}) was constructed on $\text{Ce}_{0.8}\text{Zr}_{0.2}\text{O}_2$ support with an ordered macroporous (OM) nanostructure. The dynamic low-coordinated Pt_{SA} possess more free d-orbital electrons that are not excessively occupied by Pt-O bonds, which enhanced the activation of the propane C-H bonds and enables peroxide species to remain active at high temperatures. Consequently, the $\text{Pt}_{\text{SA}}/\text{CeZrO}_2$ catalyst exhibited a high activity and stability in propane oxidation, even superior than commercial diesel oxidation catalysts $\text{Pt}/\text{Al}_2\text{O}_3$. Furthermore, $\text{Pt}_{\text{SA}}/\text{CeZrO}_2$ was integrated into commercial cordierite monolith for potential commercial applications. The research results are interesting, and propose a meaningful routine to design and synthesize alkane oxidation catalysts with excellent catalytic performances. However, the innovation of the paper should to be further strengthened. Based on overall considerations, I recommend revising and resubmitting for reconsideration for publication. The detail comments are as follows:

Reply: Thanks for raising up the valuable comment. We have focused on strengthening the description of the paper's innovation and carefully addressed each of the following comments, making the necessary revisions to improve the manuscript's quality. In response to your comment, we have expanded the discussion in both the introduction and results sections to provide a more detailed analysis of how our research builds upon and diverges from prior studies in the field. Please see our responses and detail revisions as listed below.

Comment 1. As far as I know, atomic trapping method has been proposed and widely applied to prepare single atom catalysts on CeO_2 -based support, since Abhaya K. Datye reported thermally stable single-atom platinum-on-ceria catalysts via atom trapping (Science, 2016, 353, 150-154). Subsequently, the mechanism of trapping was also proposed (ACS Catalysis, 2019, 9, 3978-3990). More significantly, locate platinum single sites on CeO_2 and the variation of the active state under reaction conditions were successfully addressed (Nature Catalysis, 2020, 3, 824-833). Therefore, it is essential to distinguish where is the platinum single site location on OM nanostructured $\text{Ce}_{0.8}\text{Zr}_{0.2}\text{O}_2$? What's the effect of Zr doping on the Pt location, besides coordinated configuration

of Pt? Why did select of the composition of $\text{Ce}_{0.8}\text{Zr}_{0.2}\text{O}_2$ support? If varying the molar ratio of Ce/Zr, can single atom dispersion still be achieved?

Reply:

(1) Based on your suggestion, we have reanalyzed the position of Pt in $\text{Ce}_{0.8}\text{Zr}_{0.2}\text{O}_2$ through STEM, EXAFS and DFT calculations. Firstly, through STEM observation (**Supplementary Fig. 21**), the atomic distances on both sides of the Pt_{SA} bright spots were found to be asymmetric, indicating that the atoms on both sides may be Ce and Zr atoms, respectively. Due to the differences in the radii and electronegativity of these two atoms, local lattice distortion occurs. Furthermore, we performed DFT calculations on the Pt loading position in $\text{Ce}_{0.8}\text{Zr}_{0.2}\text{O}_2$, as shown in **Supplementary Fig. 19**. The calculation results indicate that oxygen vacancies (O_v) are more likely to form around Zr, and when Pt_{SA} is loaded by substituting a Ce atom adjacent to Zr and O_v , the formation energy is the lowest. EXAFS characterization further confirmed the results of STEM and DFT, as shown in **Fig. 3c-d**. It can be seen that the Pt-O peak in $\text{Pt}_{\text{SA}}/\text{CeO}_2$ has good symmetry, while the Pt-O peak in $\text{Pt}_{\text{SA}}/\text{CeZrO}_2$ exhibits significant asymmetry, indicating the presence of both Pt-O-Ce and Pt-O-Zr. The difference in bond lengths between these two bonds causes the asymmetry in the first-shell Pt-O peak. This further supports that Pt is loaded around Zr atoms. In conclusion, Zr facilitates the capture of Pt atoms by the $\text{Ce}_{0.8}\text{Zr}_{0.2}\text{O}_2$ support.

Supplementary Fig. 19: (d) CeZrO_2 model; (e,f) The formation energy of oxygen vacancies (O_v) at different positions on the surface of CeZrO_2 ; (g) The Pt_{SA} loading position optimize; (h) The $\text{Pt}_{\text{SA}}/\text{CeZrO}_2$ model.

Fig. 3 (c,d) Extended X-ray Absorption Fine Structure (EXAFS) data fitting for aged $\text{Pt}_{\text{SA}}/\text{CeO}_2$ (c) and $\text{Pt}_{\text{SA}}/\text{CeZrO}_2$.

(2) Regarding the determination of the Zr doping ratio, we first conducted a literature survey. Most studies (*Environ. Sci. Technol.* **2021**,55,12607–12618; *Small* **2019**,15,1903058; *Applied Catalysis A: General* **2013**,450,131–142) suggest that a small amount of Zr doping (10%-30%) significantly promotes the catalytic activity and stability of the Pt/CeO₂ catalyst. A small amount of Zr doping can inhibit grain growth, exposing more active surface area, and also promotes the formation of surface oxygen vacancies, which enhances catalytic oxidation activity. However, excessive Zr doping (> 0.5) leads to a reduction in the surface Ce³⁺ species with variable valence, inhibiting the catalytic oxidation reaction (*Small* **2019**,15,1903058). At the same time, we also prepared catalysts with different Zr doping ratios: $\text{Pt}_{\text{SA}}/\text{Ce}_{0.9}\text{Zr}_{0.1}\text{O}_2$, $\text{Pt}_{\text{SA}}/\text{Ce}_{0.8}\text{Zr}_{0.2}\text{O}_2$, $\text{Pt}_{\text{SA}}/\text{Ce}_{0.7}\text{Zr}_{0.3}\text{O}_2$, $\text{Pt}_{\text{SA}}/\text{Ce}_{0.6}\text{Zr}_{0.4}\text{O}_2$. The activity test results (**Supplementary Fig. 11a**) show that $\text{Pt}_{\text{SA}}/\text{Ce}_{0.8}\text{Zr}_{0.2}\text{O}_2$ exhibits the best C₃H₈ catalytic oxidation activity as shown in **Revision (3)**. Therefore, $\text{Pt}_{\text{SA}}/\text{Ce}_{0.8}\text{Zr}_{0.2}\text{O}_2$ was selected for subsequent research. To study the effect of the Zr doping ratio on the Pt_{SA} structure, we characterized $\text{Pt}_{\text{SA}}/\text{Ce}_{0.8}\text{Zr}_{0.2}\text{O}_2$ and $\text{Pt}_{\text{SA}}/\text{Ce}_{0.6}\text{Zr}_{0.4}\text{O}_2$ using CO-DRIFTS (**Supplementary Fig. 11b**), as shown in **Revision (3)**. This indicates that even when the Zr doping ratio is increased to 0.4, the single-atom structure of Pt can still be maintained.

Revision:

(1) We have added and revised the corresponding analysis on **Page 13 Lines 249-256** as follows: The most thermodynamically stable configurations of $\text{Pt}_{\text{SA}}/\text{CeO}_2$ and $\text{Pt}_{\text{SA}}/\text{CeZrO}_2$ are shown in

Supplementary Fig. 20, where Pt_{SA} bonds with six oxygen atoms on pure CeO₂, but coordinates with four oxygen atoms in CeZrO₂. Furthermore, thermodynamic calculations demonstrate the lowest formation energy occurs when Pt_{SA} is anchored near Zr atoms and oxygen vacancies (O_v). The STEM results (**Supplementary Fig. 21**) also show an asymmetric atomic distance on both sides of the Pt_{SA} bright spots, which is caused by local lattice distortion induced by the differences in the atomic radii and electronegativity between Ce and Zr. This further indicates that Pt is more likely to be loaded around Zr atoms.

Supplementary Fig. 21: Annular Dark Field Scanning Transmission Electron Microscopy (ac-HAADF STEM) images of aged atomically dispersed Pt_{SA}/CeZrO₂.

(2) We have added and revised the corresponding analysis on **Page 13 Lines 260** as follows: In contrast to the symmetric Pt-O peak around 1.71 Å in Pt_{SA}/CeO₂ (**Fig. 3c**), the peak in Pt_{SA}/CeZrO₂ exhibits asymmetry (**Fig. 3d**), due to Pt loaded around Zr atom and the co-existence of Pt-O-Zr and Pt-O-Ce bonds in the Zr-doped sample. (*Angew. Chem. Int. Ed.* **60**, 26054–26062 (2021)).

(3) On **Page S15**, we added the the activity test and CO DRIFTS toward samples with different Zr content as follow (**Supplementary Fig. 11**):

Supplementary Fig. 11: (a) Light-off curves for $Pt_{SA}/Ce_{0.9}Zr_{0.1}O_2$, $Pt_{SA}/Ce_{0.8}Zr_{0.2}O_2$, $Pt_{SA}/Ce_{0.7}Zr_{0.3}O_2$ and $Pt_{SA}/Ce_{0.6}Zr_{0.4}O_2$; (b) CO DRIFTS for $Pt_{SA}/Ce_{0.8}Zr_{0.2}O_2$ and $Pt_{SA}/Ce_{0.6}Zr_{0.4}O_2$.

We also prepared catalysts with different Zr doping ratios: $Pt_{SA}/Ce_{0.9}Zr_{0.1}O_2$, $Pt_{SA}/Ce_{0.8}Zr_{0.2}O_2$, $Pt_{SA}/Ce_{0.7}Zr_{0.3}O_2$, $Pt_{SA}/Ce_{0.6}Zr_{0.4}O_2$. The activity test results (**Supplementary Fig. 11a**) show that $Pt_{SA}/Ce_{0.8}Zr_{0.2}O_2$ exhibits the best C_3H_8 catalytic oxidation activity ($T_{90} = 400$ $^{\circ}C$), while $Pt_{SA}/Ce_{0.6}Zr_{0.4}O_2$ shows the poorest catalytic oxidation activity ($T_{90} = 530$ $^{\circ}C$). It is evident that as the Zr doping ratio increases to over 30 %, the catalytic activity decreases significantly. Therefore, $Pt_{SA}/Ce_{0.8}Zr_{0.2}O_2$ was selected for subsequent research.

To study the effect of the Zr doping ratio on the Pt_{SA} structure, we characterized $Pt_{SA}/Ce_{0.8}Zr_{0.2}O_2$ and $Pt_{SA}/Ce_{0.6}Zr_{0.4}O_2$ using CO-DRIFTS, as shown in the **Supplementary Fig. 11b**. Both samples exhibit CO adsorption peaks only in the range of 2101-2112 cm^{-1} , corresponding to the characteristic peak of CO adsorbed at Pt single-atom sites. This indicates that even when the Zr doping ratio is increased to 0.4, the single-atom structure of Pt can still be maintained.

Comment 2. The effect of Zr doping on the thermal stability of CeO_2 and the corresponding mechanism were entirely revealed early, especially in three-way catalysts for vehicle exhaust purification. Therefore, the idea of applying thermally stable $CeZrO_2$ instead of CeO_2 is a natural state of being. The conclusion about the effect of zirconium doping on CeO_2 lacks sufficient novelty. Please provide more discussion about the novelty of Zr doping in the related section.

Reply: We appreciate the valuable comments from the reviewers. Indeed, Pt/CeZrO₂-based catalysts have been studied. For example, Liu et al. found that $CeZrO_2$ has more abundant oxygen

defects than CeO₂, which helps maintain better dispersion of Pt particles (*Environ. Sci. Technol.* **55**, 12607–12618, (2021)). Li et al. reported that Zr doping introduced more oxygen vacancies (O_v) and Ce³⁺ species in the support, promoting the formation of highly active metallic Pt⁰ (*Chem. Eng. J.* **322**, 234–245, (2017)). In summary, previous studies have demonstrated the promoting effect of Zr on the activity and stability of catalysts, revealing that Zr doping facilitates the formation of oxygen vacancies and Pt⁰ species. However, current research has mostly focused on the effects of Zr doping on the acidity of the catalyst support, Pt oxidation state, and redox activity.

In contrast, this study has for the first time revealed the impact of Zr doping on the atomic local coordination structure of Pt single atom (Pt_{SA}) active sites, and for the first time studied the dynamic changes of the Pt_{SA} coordination structure during the reaction process. This work advances our understanding of the mechanism by which Zr doping affects the catalyst performance at the atomic level and explores the dynamic structure-activity relationship changes during real reaction process. In response to the suggestion, to better highlight the contribution of this paper, we have made changes to the Introduction and Results, as detailed in the following **Revision**.

Revision:

- (1) On **Page 5 Line 68**, we added a description at the Introduction to emphasize the novelty of this work, as follows: “However, the implementation of strategies and their industrial application still face critical knowledge gaps and technological challenges. Particularly regarding how Zr doping affects the coordination dynamics of Pt_{SA} sites during high-temperature reactions, and how to achieve scalable monolithic integration of this OM catalyst for industrial use.”
- (2) On **Page 17 Line 367**, we added a description at the end of the **Results** in **Fig 3** and **Fig 4** to emphasize the novelty of this work, as follows: “In summary, while Zr doping is an established methodology, this study provides atomic-scale insights into the stabilization mechanism of catalysts, elucidating how Zr modulates the coordination dynamics of metal centers and regulates reactive oxygen species during high-temperature catalysis.”
- (3) On **Page 20 Line 446**, we added a description at the end of the **Results** in **Fig 5** to emphasize the novelty of this work, as follows: “While OM CeZrO₂ support coatings are established catalytic materials, this study develops the first one-step synthesis protocol and enables industrial-scale fabrication on 3.4L cordierite substrates. These findings indicate that the OM support not only

promotes mass transfer and diffusion of reactants but also mitigates Pt loss under high temperature, thereby enhancing the long-term durability and performance of the Pt_{SA}/CeZrO₂ catalyst in industrial settings.”

Comment 3. Where did commercial diesel oxidation catalysts Pt/Al₂O₃ come from? In fact, the commercial diesel oxidation catalysts also include Pt/CeZrO_x catalysts. Therefore, the comparison of the activity between Pt_{SA}/CeZrO₂ and commercial Pt/CeZrO_x catalysts should be conducted. On the other hand, the activity of Pt_{SA}/CeZrO₂ should be also compared with noble metal catalysts in references.

Reply:

(1) The Pt-based commercial catalyst we used is sourced from DONGFENG MOTOR CORPORATION. In fact, this catalyst (**Figure R1a, b**) contains a more complex composition, as indicated by the XPS (**Figure R1c**), which shows the presence of Pt, Pd, Ce, Zr, Y, Ba, Al, Si, and O. The ICP data (**Figure R1d**) reveals the contents of Ce and Zr as 30.5g/L and 5.1g/L, respectively, allowing it to be classified as a Pt/CeZrO_x-based catalyst. However, due to confidentiality agreements with our partner regarding proprietary information, we have not disclosed all the elements of the commercial DOC catalyst in the manuscript. Following your suggestion, in order to emphasize the comparability with commercial catalysts, we have revised the manuscript to refer to the Pt-based commercial catalyst as Com-PtPd/CeZr.

(2) We have supplemented the comparison of the T₉₀ and TOF of the propane catalytic oxidation catalysts with literature reports, as shown in **Supplementary Table 1-2**.

(3) Additionally, we have provided a comparison of the tolerance temperature and time of the Pt-based DOC catalyst from the reference literature, as shown in **Fig. 5f** in following revision.

Figure.R1 (a, b) Actual photographs of the commercial PtPd/CeZrYBaO_x (Com-PtPd/CeZr); (c) The full XPS spectrum of Com-PtPd/CeZr; (d) The ICP data of Com-PtPd/CeZr.

Revision:

(1) we have revised the manuscript to refer to the Pt-based commercial catalyst as Com-PtPd/CeZr as follow **Figures** and corresponding analysis.

Fig. 2e, Comparison of aged Pt_{SA}/CeZrO₂ (0.4 g_{Pt}/L) with an aged commercial PtPd/CeZr catalyst (0.9 g_{Pt}/L + 0.1 g_{Pd}/L).

Fig. 5 f, Diesel oxidation performance comparison between the OM $\text{Pt}_{\text{SA}}/\text{CeZrO}_2$ integrated monolith ($0.4 \text{ g}_{\text{Pt}}/\text{L}$) and commercial PtPd/CeZr catalysts ($0.9 \text{ g}_{\text{Pt}}/\text{L} + 0.1 \text{ g}_{\text{Pd}}/\text{L}$).

(2) On Supplementary Page 35, we have added the comparison of the T_{90} and TOF of the propane catalytic oxidation catalysts with literature reports, as shown in Supplementary Table 1-2.

Supplementary Table 1. Comparative activity of catalysts for the catalytic oxidation of C_3H_8 .

Catalysts	concentration (ppm)	Space velocity ($\text{mL}\cdot\text{g}^{-1}\cdot\text{h}^{-1}$)	T_{50}/T_{90} ($^{\circ}\text{C}$)	Ref
$\text{Pt}_{\text{SA}}/\text{CeZrO}_2$	3000	37500	345/400	This work (800 $^{\circ}\text{C}$ -Aged)
$\text{Pt}_{\text{SA}}/\text{CeO}_2$	3000	37500	450/550	This work (800 $^{\circ}\text{C}$ -Aged)
PtPd/CeZr	3000	37500	370/455	Commercial (800 $^{\circ}\text{C}$ -Aged)
Pt/MnCoO_x	3000	37500	490/560	Commercial (800 $^{\circ}\text{C}$ -Aged)
$\text{Pt}/\text{TiO}_{2-x}$	10000	60000	550/650	1
Pt/CeO_2	1000	12000	375/465	3
Pd/CeO_2	2000	30000	325/530	4
Pt/LaCoO_3	8000	30000	390/500	5
$\text{Pt}/\text{CeO}_2\text{-HA}$	2000	36000	320/450	6

Supplementary Table 1 shows that $\text{Pt}_{\text{SA}}/\text{CeZrO}_2$ exhibits the lowest T_{90} for C_3H_8 oxidation compared with other Pt-based catalysts reported in the references, even after thermal aging at 800 $^{\circ}\text{C}$.

Supplementary Table 2. Comparative TOF of catalysts for the catalytic oxidation of C₃H₈.

Catalysts	Pt loading (wt %) ^a	Reaction rate ($\mu\text{mol}/(\text{g}_{\text{cat}}*\text{s})$) ^b	TOF*10 ³ (s ⁻¹) ^b	Ref
Pt _{SA} /CeZrO ₂	0.04	102.7	501	This work (800 °C-Aged)
Pt _{SA} /CeO ₂	0.04	4.5	22	This work (800 °C-Aged)
Pt/MnCoO _x	0.04	3.5	17	Commercial (800 °C-Aged)
Pt/TiO _{2-x}	0.52	182.1	68	1
Pt/CeO ₂	0.23	112.8	96	3
Pt/LaCoO ₃	0.29	4.5	3	5
Pt/CeO ₂ -HA	0.84	258.9	60	6

^a Measured by ICP.

^b Calculated by the reaction rate at 300 °C.

Supplementary Table 2 shows that Pt_{SA}/CeZrO₂ exhibits the largest TOF for C₃H₈ oxidation compared with other Pt-based catalysts reported in the references, even after thermal aging at 800 °C.

(3) **On Page 19**, we have provided a comparison of the tolerance temperature and time of the Pt-based DOC catalyst from the reference literature, as shown in **Fig. 5f**

Fig. 5f, The comparison of tolerance temperature and time of the Pt-based DOC catalyst from the reference literature.

Fig. 2d shows that Pt_{SA}/OM CeZrO₂ can withstand hydrothermal aging at 800°C for 50h with 10 vol% H₂O, maintaining over 90% propane conversion for 1000h at 450°C. **Supplementary Fig. 11** indicates that after 2h of thermal aging at 1100°C, the T₉₀ of Pt_{SA}/OM CeZrO₂ increases slightly (from 400°C to 460°C) but stays below 500°C, while the T₉₀ of Pt_{SA}/OM CeO₂ and two commercial Pt-based catalysts rise above 600°C. These results demonstrate that Pt_{SA}/OM CeZrO₂ can withstand extreme conditions—800°C for 50h, 450°C for 1000h, and 1100°C for 2h—showing superior high-temperature stability compared to Pt-based DOC catalysts in the reference (**Fig. 5f**). The synergy of the OM structure and Zr doping helps resist Pt-sintering, Pt-overoxidation, and Pt-loss, ensuring Pt_{SA}/OM CeZrO₂'s durability in extreme high-temperature oxidative environments.

Comment 4. What's aging atmosphere conditions for the experiments of Pt loss? If it was conducted under static conditions at high temperature, how is the Pt loss for the Pt_{SA}/CeZrO₂ catalyst after aging under reactive conditions?

Reply: Thank you for your suggestion. We have added the specific parameters and methods of the experiment. The aging conditions for the Pt loss experiment are in a flowing air atmosphere (20% O₂, N₂ balance, flow rate 50 mL/min, 800 °C for 50 hours / 1000 °C for 2 hours). Additionally, based on your suggestion, we have also included the ICP data for the samples aged in static air. As

shown in **Revision**.

Revision:

- (1) We have added the specific parameters and methods of the experiment on **Page 19 Lines 415-416** as follows: 20% O₂, N₂ balance, flow rate 50 mL/min, 800 °C for 50 hours/1000 °C for 2 hours.
- (2) We have also included the ICP data for the samples aged in static air as shown in **Supplementary Fig. 28**:

Supplementary Fig. 28: (a, b) Changes in Pt loading of catalysts after aging at 800 °C and 1100 °C in static air (a) and flowing air (b). (c) Schematic diagram illustrating the inhibitory effect of the OM structure on Pt loss (20% O₂, N₂ balance, flow rate 50 mL/min, 800 °C for 50 hours / 1000 °C for 2 hours).

As shown in **Supplementary Fig. 28**, the powder Pt/CeZrO₂ and commercial Pt-based catalysts lost 15% and 11% more Pt in flowing air at 800 °C compared to static air. However, the OM Pt/CeZrO₂ sample had similar Pt loss in both flowing and static conditions, showing better tolerance to flowing air. These results demonstrate that the abundant macropores and mesopores in the OM nanostructure can effectively capture Pt atoms, reducing Pt loss by 37.5% under extreme temperatures (**Fig. 5e** and **Supplementary Fig. 28**).

Comment 5. This paper solved the problems of Pt sintering, overoxidation and loss by preparing the Pt_{SA}/CeZrO₂ catalyst. However, in actual applications of DOC, the catalyst's resistance to sulfur poisoning also needs to be considered. How is the sulfur resistance of this catalyst?

Reply: Based on your constructive suggestion, we have added the sulfur resistance test. The results

show that Pt_{SA}/CeZrO₂ can withstand 200 ppm SO₂, with no decline in catalytic oxidation activity over 50 hours.

Revision:

(1) We have added the corresponding analysis on **Page 9 Lines 195-197** as follows: Moreover, the Pt_{SA}/CeZrO₂ catalyst shows excellent sulfur resistance, maintaining a stable propane conversion rate of 96% at 400 °C with 200 ppm SO₂ for 50 hours (**Supplementary Fig. 16**).

(2) On **Page S19**, we have added the sulfur resistance test as follow (**Supplementary Fig. 16**):

Supplementary Fig. 16 (a) C₃H₈ catalytic oxidation activity of Pt_{SA}/CeZrO₂ catalyst under different reaction conditions (with or without SO₂); **(b)** C₃H₈ catalytic oxidation sulfur resistance lifetime test of Pt_{SA}/CeZrO₂ catalyst at 400 °C with 200 ppm SO₂ in the reaction atmosphere.

Supplementary Fig. 16 shows the sulfur resistance performance of the Pt_{SA}/CeZrO₂ catalyst in C₃H₈ catalytic oxidation. Notably, after introducing 200 ppm SO₂, the catalytic oxidation conversion rate increased, with the T₉₀ decreasing from 400 °C to 350 °C (**Supplementary Fig. 16a**). Additionally, at 400 °C, the conversion rate improved from 90% to 96% upon the introduction of 200 ppm SO₂, maintaining stable performance for 50 hours without any degradation (**Supplementary Fig. 16b**). These results highlight the excellent sulfur resistance of the Pt_{SA}/CeZrO₂ catalyst.

Reviewer: 2

General comments: The article presents a significant advancement in the field of catalysis, particularly in the development of platinum (Pt) single-atom catalysts (SACs) supported on ordered macroporous (OM) structures. The findings have important implications for industrial applications, especially in emission control technologies, as they address critical challenges such as catalyst deactivation, which is a major concern in high-temperature catalytic processes. The integration of the catalyst into a commercial monolith demonstrates its practical applicability, enhancing its potential impact on reducing environmental pollutants.

The research addresses three major deactivation mechanisms (Pt sintering, Pt overoxidation, and Pt loss) that limit the efficiency and durability of catalytic converters. By introducing a novel Zr-doping strategy and demonstrating the scalability of the catalyst design with the catalyst able to maintain high conversion rates even under extreme conditions.

Although the article is well-structured, with a clear introduction, methodology, results, and discussion sections, some sections, particularly those detailing complex experimental techniques and results, may benefit from simplification or additional explanations to enhance understanding for a broader audience. Visual aids such as diagrams and charts effectively complement the text, but further clarification of specific terms and concepts could improve accessibility.

Reply: We sincerely appreciate the reviewer's positive feedback and thoughtful comments on our manuscript. We have simplified some of the more intricate details and added further explanations where necessary to enhance understanding. Additionally, we have included a few more visual aids to support the text and help illustrate key concepts. Please see our responses and detail revisions as listed below.

Comment 1. Some sections of the article contain complex jargon (first time - likely coined by authors) and intricate descriptions that may hinder understanding and limit the article's accessibility to a broader audience.

Reply: We have revisited the complex jargon and simplified intricate descriptions as follow revision.

Revision:

- (1) On **Page 5 Line 88**, we have revised the phrasing “Compared with packed-bed reactors (a type of reactor where catalysts are packed into a column), honeycomb monolith-based structured catalytic devices offer higher heat and mass transport, reduction in uneven temperature distribution across the catalyst, and simpler scalability.”.
- (2) On **Page 7 Line 117**, we have revised the jargon as “Inductively Coupled Plasma (ICP)”.
- (3) On **Page 11 Line 230**, we have revised the phrasing “The calculated oxygen vacancy formation energy (E_{Ov}) in **Fig. 3a** indicates that oxygen vacancies (O_v) are more easily formed around Zr atoms (-0.73 eV) compared with other sites where the formation energy is higher (1.72 eV).”
- (4) On **Page 17 Line 351**, we have added description “Unlike typical adsorbed oxygen on Pt_{SA}/CeO_2 , the peroxide species on $Pt_{SA}/CeZrO_2$ has one oxygen atom embedded in a lattice vacancy (**Fig. 4e**). Due to its combination of the bond length of peroxide species and the position of lattice oxygen, we refer to it as a lattice peroxide species ($Latt-O_2^{2-}$). This unique $Latt-O_2^{2-}$ could potentially combine the reactivity advantage of molecular oxygen with the stability advantage of lattice oxygen.”

Comment 2. While figures and diagrams are included, some of them may require clearer labeling and explanations. The significance of certain results could be better illustrated through visual means.

Reply: We have revised the captions for **Fig. 2**, **Fig. 5**, and **Supplementary Fig. 13**, **Fig. 15** to make them more detailed and added explanations for **Supplementary Fig. 1** to **Supplementary Fig. 7**. Additionally, a schematic diagram has been added to **Fig. 5** to better illustrate the role of the OM structure in suppressing Pt loss as follow revision.

Revision:

- (1) We have revised the captions as follow “**Fig. 2b**, Normalized reaction rate of aged catalysts at 450 °C.”.
- (2) We have revised the captions as follow “**Fig. 5d**, Changes in Pt loading of catalysts after aging at 800 °C and 1100 °C (20% O_2 , N_2 balance, flow rate 50 mL/min, 800 °C for 50 hours/1000 °C for 2 hours).”.
- (3) We have revised the captions as follow “**Supplementary Fig. 6b** Mechanical stability of $Pt_{SA}/OM CeZrO_2$, the inside shows the SEM images of the samples after ultrasound treatment for

different durations.”.

(4) We have added the explanations for **Supplementary Fig. 6b** as follow “As shown in **Supplementary Fig. B**, the mass loss of the OM sample during ultrasound treatment is slightly higher than that of the Powder sample, possibly due to minor damage to the OM pore structure. However, the mass loss difference between the two samples is minimal, with both being less than 1%. This suggests that the OM coating exhibits good mechanical stability, with no significant difference compared to traditional coatings. Furthermore, the SEM images (**Supplementary Fig. 6b** inside) of the Pt_{SA}/OM CeZrO₂ catalyst after ultrasound treatment for different durations show that the OM structure undergoes slight deformation, but the overall OM framework structure remains stable, further confirming the excellent mechanical stability of the Pt_{SA}/OM CeZrO₂ catalyst. The good mechanical stability of the Pt_{SA}/OM CeZrO₂ catalyst may be attributed to its uniform and ordered pore structure, which allows stress to be evenly distributed across the surface and within the pores, rather than concentrating in a specific area, thereby better maintaining overall stability and reducing local collapse.”.

(5) We have added the explanations for **Supplementary Fig. 13b** as follow “As shown in **Supplementary Fig. 13b**, Pt_{NP}/CeZrO₂ exhibits good cyclic stability, while the activity of Pt_{NP}/CeO₂ significantly decreases during the cycling tests. This could be attributed to the highest testing temperature of 800 °C, at which both catalysts may transform into Pt_{SA}. As shown in **Fig. 2a**, after thermal aging at 800 °C, Pt_{SA}/CeO₂ shows poor activity, whereas Pt_{SA}/CeZrO₂ maintains activity similar to that of Pt_{NP}/CeZrO₂. These results further confirm the transformation of Pt_{NP} to Pt_{SA} at 800 °C and highlight the poor high-temperature resistance of Pt_{NP}/CeO₂. As seen in **Supplementary Fig. 13c**, both Pt_{SA} catalysts obtained after 800 °C thermal aging exhibit good cyclic stability, but the activity of Pt_{SA}/CeO₂ is significantly lower than that of Pt_{SA}/CeZrO₂, with a T₅₀ increase of approximately 100 °C, further demonstrating its poor resistance to 800 °C.”.

(6) A schematic diagram has been added to **Fig 5** to better illustrate the role of the OM structure in suppressing Pt loss as follow:

Fig. 5e Schematic diagram illustrating the inhibitory effect of the OM structure on Pt loss.

Comment 3. The discussion section, although comprehensive, could benefit from a more in-depth exploration of the implications of the findings. For example, discussing potential real-world applications and scenarios where this catalyst could be implemented would enhance the relevance of the study.

Reply: Thanks for your valuable comment. We have added real-world applications and scenarios in the discussion section to highlight the relevance of this study as follow revision.

Revision: On Page 21 Line 470-474, We have revised the last sentence of the discussion as follows: "This work revealed the dynamic evolution of Pt_{SA} configuration under oxidative conditions and presented a dual-scale design strategy to address three key deactivation issues—Pt sintering, Pt overoxidation, and Pt loss—thereby enabling the development and scaling of robust catalytic converters for high-temperature heterogeneous Pt-based catalysts."

Comment 4. The article could provide a more thorough comparison with existing literature. While it mentions previous work, a more detailed analysis of how this research builds upon or diverges from prior studies would strengthen the argument for its novelty.

Reply: Thanks for your valuable feedback. We agree that a more thorough comparison with existing literature would enhance the manuscript and further highlight the novelty of our work. In response to your comment, we have expanded the discussion in both the introduction and results sections to provide a more detailed analysis of how our research builds upon and diverges from prior studies in the field. We believe these revisions strengthen the argument for the originality and significance

of our work, as following:

Revision:

- (1) On **Page 5 Line 68**, we added a description at the Introduction to emphasize the novelty of this work, as follows: “However, the implementation of strategies and their industrial application still face critical knowledge gaps and technological challenges. Particularly regarding how Zr doping affects the coordination dynamics of Pt_{SA} sites during high-temperature reactions, and how to achieve scalable monolithic integration of this OM catalyst for industrial use.”
- (2) On **Page 17 Line 367**, we added a description at the end of the **Results** in **Fig 3** and **Fig 4** to emphasize the novelty of this work, as follows: “In summary, while Zr doping is an established methodology, this study provides atomic-scale insights into the stabilization mechanism of catalysts, elucidating how Zr modulates the coordination dynamics of metal centers and regulates reactive oxygen species during high-temperature catalysis.”
- (3) On **Page 20 Line 446**, we added a description at the end of the **Results** in **Fig 5** to emphasize the novelty of this work, as follows: “While OM CeZrO₂ support coatings are established catalytic materials, this study develops the first one-step synthesis protocol and enables industrial-scale fabrication on 3.4L cordierite substrates. These findings indicate that the OM support not only promotes mass transfer and diffusion of reactants but also mitigates Pt loss under high temperature, thereby enhancing the long-term durability and performance of the Pt_{SA}/CeZrO₂ catalyst in industrial settings.”

Comment 5. Certain methodological aspects, particularly regarding the characterization techniques, could be elaborated upon. Providing more detail on the experimental setup, conditions, and rationale for chosen methods would enhance reproducibility and understanding.

Reply: Thanks for your valuable feedback. In the revised manuscript, we have added a more detailed description of the experimental setup and the conditions under which the characterization techniques were conducted as following revision.

Revision: On **Page 24 Line562-627**, we have revised the section of Catalyst characterization as follow:

(1) The X-ray absorption fine structure data were obtained from the BL14W1 station at the Shanghai Synchrotron Radiation Facility, which operates with a storage ring energy of 3.5 GeV. The beamline utilizes a monochromatic X-ray source with a specific energy range appropriate for the absorption edge of Pt. The experimental setup was calibrated by simultaneously measuring the spectrum of a reference Pt foil to ensure accurate data for each sample. To ensure proper calibration and reproducibility, we carefully controlled the experimental conditions. The beamline was operated under a vacuum environment to minimize the influence of atmospheric scattering. Subsequent to data collection, the X-ray absorption spectroscopy (XAS) data underwent processing and fitting procedures using the ATHENA module in the IFEFFIT software packages. The raw data were first corrected for background, energy shifts, and normalization. For the Pt L3 edge data, the fitting process involved the first coordination shell based on Fourier-transformed k^3 -weighted $\chi(k)$ functions within the k -range of 3.0-13.9 \AA^{-1} . This range was selected based on the characteristics of the Pt L3-edge and the need to focus on the first coordination shell, as it provides the most relevant structural information. The coordination numbers were determined by fixing the amplitude reduction factor (S_0^2), which was derived from fitting the reference Pt foil. This approach was employed to maintain consistency in the analysis and improve the reliability of the results. In cases where the number of independent data points was limited due to experimental constraints, the Debye-Waller factor (σ^2) was set to a reasonable value of 0.003.

(2) The Near-Ambient Pressure X-ray Photoelectron Spectrometer (NAP-XPS) measurements were performed at the Shanghai Synchrotron Radiation Facility beamline, using a differentially-pumped Al $K\alpha$ source (Specs model XR50) with a photon energy of 1486.6 eV. Emitted photoelectrons (and Auger electrons) were detected with a near-ambient pressure hemispherical analyzer (Specs model Phoibos 150), mounted in a custom-designed system capable of measuring XPS under sample gas pressures up to 10 Torr. The specific testing procedure is as follows: The sample was placed into the sample chamber and sputter-cleaned for 30 minutes. The temperature was set to 673 K, and spectra were collected under two conditions: UHV and $C_3H_8 + O_2$ (with a partial pressure of 1/10). Full spectra were acquired, along with specific C 1s and Pt 4f spectra.

(3) The crystalline structure of the sample was analyzed using X-ray diffraction (XRD) with $CuK\alpha$ radiation (wavelength = 1.540598 \AA) on a BRUKER AXS D5005 instrument. The scan was

performed in the 2θ range of 10° to 90° , with a step size of 0.02° and a scan rate of 0.5° per minute. This setup was chosen to ensure sufficient resolution in detecting peaks corresponding to various crystal planes. The XRD patterns were analyzed to determine the phase composition.

(4) Mercury Intrusion Porosimetry (MIP) measurements were conducted using a Micromeritics AutoPore IV 9500 apparatus, covering a pressure range of 0-10000 psia. The technique was used to measure the pore size distribution and total porosity of the sample. Prior to measurement, samples were degassed at 200°C for 6 hours under vacuum to remove surface contaminants.

(5) This model simplifies the convection–diffusion–reaction equations by averaging in the transverse direction, yielding an overall mass transfer coefficient for transverse diffusion and reaction. Key assumptions are: (a) laminar flow is assumed within the monolith channel, characterized by a Reynolds number of less than 2300. (b) Axial diffusion and heat conduction in the fluid phase are considered negligible relative to convection, with justification based on the Prandtl and Schmidt numbers being small for typical monolith flows. and (c) constant physical properties, including viscosity and diffusivity, were assumed over the temperature and concentration ranges explored. The model includes transport equations for species mass balances in the gas phase, washcoat phase, and gas-solid interface.

Comment 6. The focus on a dynamic low-coordinated Pt single-atom configuration is technically significant. This configuration enhances the reactivity of the catalyst by allowing more unoccupied d-orbitals, which facilitates the activation of reactants, particularly under high-temperature conditions. It would be good if a modeling could be done to confirm this argument.

Reply: Thanks for your valuable suggestion. To better illustrate the dynamic changes in the Pt_{SA} coordination structure during high-temperature reactions, we have included snapshots of $\text{Pt}_{\text{SA}}/\text{CeZrO}_2$ model from the AIMD simulation at different time intervals in the **Supplementary Fig. 23**.

Revision: we have included snapshots of $\text{Pt}_{\text{SA}}/\text{CeZrO}_2$ model from the AIMD simulation at different time intervals in the **Supplementary Fig. 23** as follow:

Supplementary Fig. 23 The snapshots of $\text{Pt}_s/\text{CeZrO}_2$ model from the AIMD simulation at different time (400 °C, $1\text{C}_3\text{H}_8$, 20O_2).

As shown in **Supplementary Fig.23**, the coordination number of the Pt-O bond fluctuates between 4 and 5 throughout most of the reaction under 400 °C.

These are some suggestions that could further enhance the quality of the submission.

Comment 7. While the catalyst demonstrates high conversion rates and stability after 50 hours of hydrothermal aging at 800 °C, the article does not provide extensive data on long-term stability over extended operational periods. The performance under real-world conditions over months or years is crucial for industrial applications. It is important to demonstrate that the test conditions are sufficient to make this extrapolation.

Reply: Thanks for your valuable suggestion. In industry, catalysts are required to operate stably for at least several months to several years. However, to more efficiently evaluate the stability of catalysts in the laboratory, researchers often subject the catalysts to extreme environmental conditions to simulate catalyst deactivation after long-term use. Based on the USDRIVE's protocol (*Emiss. Control Sci. Technol.* 5, 183-214, (2019)), we selected aging conditions of 800 °C for 50 hours with 10 vol% H_2O , which can simulate the condition of catalysts after long-term use and is widely adopted by researchers (*ACS Catal.* 13, 5456-5471, (2023); *Science* 358, 1419-1423 (2017)). Additionally, a 1000-hour stability test was also added to better study the catalyst's stability, as

shown in Fig 2d. It can be seen that the catalyst achieved 1000 hours of operation without obvious degradation at 450 °C, with the conversion rate maintained above 92%, demonstrating the catalyst's excellent stability.

Revision:

(1) On Page 10 Fig 2, we added a 1000-hour stability test of Pt_{SA}/CeZrO₂ and corresponding analysis as follow:

Fig. 2 d, Durability performance before and after hydrothermal aging. The testing and aging cycling experiments consist of three stages: (1) Stability test at 450 °C for 25 hours, (2) Aging in humid air (10 vol% H₂O) at 800 °C for 50 hours, (3) Stability test in humid air (10 vol% H₂O) at 450 °C for 925 hours.

Notably, as illustrated in Fig. 2d, after 50 hours of hydrothermal aging at 800 °C with 10 vol% H₂O, the conversion of Pt_{SA}/CeO₂ decreased significantly from 45% at 450 °C to just 12%. In contrast, Pt_{SA}/CeZrO₂ maintained a high conversion of 92% within 1000 hours.

Comment 8. The article does not sufficiently address the reproducibility of the catalyst's performance across different batches or synthesis methods. Variability in catalyst preparation could lead to inconsistent results. This will require a more explicit disclosure of the synthetic approach.

Reply: Thank you for your valuable feedback. To enhance the analysis of the catalyst's performance reproducibility, we have added the following content to the paper: (1) In the experimental section,

we have provided a more detailed description of the catalyst synthesis process, clarifying the raw materials, synthesis conditions, and reaction steps, to ensure that other researchers can replicate our experimental procedure. (2) We have added performance testing data for multiple catalysts from different batches and conducted statistical analysis on these data. The experimental results show that the catalyst performance between different batches exhibits good reproducibility, with the error margin being controllable. In the paper, we also discuss the factors that may affect the catalyst's performance and emphasize the key influencing factors. (3) We usually use catalysts prepared from the same batch for comparative experiments to minimize experimental errors. Specific details can be found in the following **Revision**.

Revision

(1) On **Page 21 Line 477**, we have modified the catalyst preparation section, improved some details, and emphasized key steps as following: “**Scalable synthesis of OM CeZrO₂ catalysts**.Initially, a well-optimized blend of polystyrene (PS) microsphere dispersion and metal precursor sol was prepared **as described in Supplementary Method**.After soaking, these substrates were subjected to a vacuum (about -1 kPa) to eliminate excess solution **and improve the uniformity of the sol-gel attachment, as shown in Supplementary Fig. 3c**..... This drying method ensures the uniform adhesion of the sol on the substrate surface while preventing the PS microspheres from prematurely degrading before the CeZrO₂ crystallizes. It is important to pre-adjust the oven temperature before placing the samples inside and precisely control the drying time in the first step to be between 2.5 to 3 minutes. It is important to note that if the temperature is directly raised to 550 °C, the sudden decomposition of the PS template and the shrinkage of the metal oxide will occur simultaneously, causing the pore walls to collapse due to stress concentration. Stepwise calcination allows the template decomposition and material densification to occur in stages, ensuring the mechanical stability of the pore structure.**Pt-loading**. the integral OM CeZrO₂ is immersed in a **75% ethanol** solution of Pt salt, Pt(NH₃)₄(NO₃)₂, with a certain concentration of 1.7 g/L. it is microwave-dried for 30 seconds (**1000 W**) **The rapidity of microwave drying ensures the uniformity of Pt loading**. a tubular furnace under an air atmosphere at 800 °C for 50 hours (**50 mL/min**),.....”

(2) On **Supplementary Page 14**, we have added performance testing data for multiple catalysts

from different batches and conducted statistical analysis on these data:

Supplementary Fig. 10 Performance testing data for multiple catalysts from different batches.

(3) On **Supplementary Page14**, we have added analysis about **Supplementary Fig. 10** as follow:

“To investigate the reproducibility of catalyst preparation, we evaluated the propane catalytic oxidation performance across four catalyst batches, showing T_{90} values around 405 °C. While achieving complete consistency in performance across different batches remains challenging due to difficulties in precisely controlling the uniformity of the monolithic catalyst's ordered macroporous (OM) pore structure and Pt loading, rigorous optimization of synthesis protocols enabled us to constrain the T_{90} error margin within ± 15 °C. This result demonstrates satisfactory reproducibility of the catalyst.”

(4) **On Page 8 Line 153-156**, we have added analysis about **Supplementary Fig. 10** as follow:

“Furthermore, the catalysts prepared in different batches exhibit good repeatability, with the T_{90} error controlled within ± 15 °C (**Supplementary Fig. 10**). The uniformity of the OM structure and Pt loading during the synthesis process is a key control step for performance.”

Comment 9. Although the article identifies and addresses key deactivation mechanisms (sintering, overoxidation, and loss), there is limited empirical evidence or data presented to demonstrate how these mechanisms specifically affect the catalyst over time in practical applications. Operando techniques could establish SAR for the catalyst.

Reply: Thanks for your insightful comment. To better reveal the Pt-sintering, Pt-overoxidation, and Pt-loss resistance mechanisms of the Pt_{SA}/OM CeZrO₂ catalyst, we have supplemented *in situ* NAP XPS, *in situ* CO-DRIFTS, and *ex situ* ICP data based on your suggestions. Firstly, we conducted *in situ* CO-DRIFTS experiments on the aged and tested Pt_{SA}/CeZrO₂ catalyst at the T₉₀ reaction temperature (400 °C). Secondly, to further verify the effect of Zr-assisted atomic trapping on the catalyst's oxidation resistance, we employed *in situ* NAP XPS to characterize the chemical environment changes of Pt_{SA}/CeZrO₂ and Pt_{SA}/CeO₂ under reaction conditions (400 °C, C₃H₈:O₂ = 1:20, 0.2 mbar). Finally, for the study of the OM structure's effect on Pt-loss suppression in high-temperature flowing air, we have supplemented experiments on the influence of static and flowing air high-temperature aging on Pt loss, and corresponding *ex situ* ICP characterization.

Revision:

(1) On Supplementary Page 22, Fig. S18, we added the *in situ* CO DRIFTS of Pt_{SA}/CeZrO₂ and corresponding analysis as follow:

Supplementary Fig. 18 The *in situ* CO DRIFTS of Pt_{SA}/CeZrO₂ after 400 °C, 1C₃H₈, 20 O₂).

We conducted *in situ* CO-DRIFTS experiments on the aged and tested Pt_{SA}/CeZrO₂ catalyst at the T₉₀ reaction temperature (400 °C). As shown in **Supplementary Fig. 18**, after switching off the CO gas supply, only a peak appears at 2097 cm⁻¹, indicating that CO is chemically adsorbed on the Pt single-atom sites and remains stable at 400 °C, without showing any peak related to CO bridging adsorption on Pt nanoparticles. The *in situ* CO-DRIFTS further corroborated the conclusions drawn

from STEM and *ex situ* CO-DRIFTS (Fig. 2g and Supplementary Fig. 17b), showing that both Pt_{SA}/CeZrO₂ and Pt_{SA}/CeO₂ catalysts have excellent sintering resistance.

(2) On Page 12, Fig. 3e-f, we added the *in-situ* NAP-XPS Pt 4f spectra for Pt_{SA}/CeZrO₂ and Pt_{SA}/CeO₂ catalysts before and after dosing C₃H₈ + O₂ as follow:

Fig. 3 Effect of Zr on the Pt_{SA} coordination environment. **e, f**, *In-situ* Near-Ambient Pressure X-ray Photoelectron Spectroscopy (NAP-XPS) Pt 4f spectra for Pt_{SA}/CeZrO₂ (e) and Pt_{SA}/CeO₂ (f) before and after dosing C₃H₈ + O₂.

We employed *in-situ* NAP XPS to characterize the chemical environment changes of Pt_{SA}/CeZrO₂ and Pt_{SA}/CeO₂ under reaction conditions (400 °C, C₃H₈:O₂ = 1:20, 0.2 mbar). As shown in Fig. 3. e-f, the binding energies of Pt_{SA}/CeZrO₂ and Pt_{SA}/CeO₂ are 73.2 eV and 74.1 eV, respectively, representing Pt²⁺ and Pt⁴⁺ species, which are consistent with the XANES analysis (Fig.

2h), proving that Pt is closer to the metallic state with the assistance of Zr. Furthermore, as the reaction gas ($C_3H_8+O_2$) was introduced, the binding energy of $Pt_{SA}/CeZrO_2$ remained at 73.2 eV, indicating that the catalyst can withstand high-temperature oxidative environments and maintain Pt in a low oxidation state without undergoing overoxidation. However, the binding energy of Pt_{SA}/CeO_2 shifted gradually to higher binding energies (from 74.1 to 74.9 eV) as the reaction proceeded (0 \rightarrow 15 minutes), stabilizing at 74.9 eV from 15 to 40 minutes. This indicates that Pt underwent oxidation during the reaction and stabilized in a higher oxidation state, which is unfavorable for the adsorption and activation of reactants. *In situ* NAP XPS results further demonstrate that $Pt_{SA}/CeZrO_2$ with Zr assistance has better oxidation resistance than Pt_{SA}/CeO_2 , and confirm the oxidation resistance mechanism derived from EXAFS (Fig 3. c-d), where Zr helps stabilize Pt in a low-coordination state.

(3) On Supplementary Page 32, we have also included the *ex-situ* ICP data for the samples aged in static air:

Supplementary Fig. 28: (a, b) Changes in Pt loading of catalysts after aging at 800 °C and 1100 °C in static air (a) and flowing air (b). (c) Schematic diagram illustrating the inhibitory effect of the OM structure on Pt loss (20% O_2 , N_2 balance, flow rate 50 mL/min, 800 °C for 50 hours / 1000 °C for 2 hours).

As shown in Supplementary Fig. 28, the powder $Pt/CeZrO_2$ and commercial Pt-based catalysts lost 15% and 11% more Pt in flowing air at 800 °C compared to static air. However, the OM $Pt/CeZrO_2$ sample had similar Pt loss in both flowing and static conditions, showing better tolerance to flowing air. These results demonstrate that the abundant macropores and mesopores in

the OM nanostructure can effectively capture Pt atoms, reducing Pt loss by 37.5% under extreme temperatures (Fig. 5e and Supplementary Fig. 28).

Comment 10. While the article discusses the dynamic low-coordinated configuration of Pt single atoms and their reactivity, there is a lack of detailed experimental data supporting the claims about the behavior of these active sites under varying operational conditions.

Reply: Thank you for your suggestion. We considered experimentally verifying the fast oscillations in the coordination structure of Pt single atoms (Pt_{SA}) during the reaction process. However, we regret that we have not yet found an appropriate monitoring method. During the reaction, the coordination structure of Pt_{SA} changes rapidly on the femtosecond timescale due to temperature variations and reactant adsorption/desorption. Currently, there is a lack of in situ techniques capable of monitoring such fast atomic structure changes. For example, *in situ* NAP XPS can detect stable oxidation states during the reaction, but does not observe the oscillatory behavior seen in AIMD simulations, primarily due to its slower signal collection rate. Femtosecond time-resolved synchrotron radiation techniques could, in theory, monitor these oscillations in Pt_{SA} coordination during the reaction process. However, such techniques are still under development and are not widely accessible at this time. We are aware that millisecond-scale in situ synchrotron techniques (*Nat Commun* 16, 726 (2025). *Angew. Chem. Int. Ed.* 63, e202404213, (2024).) can be used to detect rapid atomic structure changes in materials, but unfortunately, these techniques are not currently within our reach. This technological bottleneck presents objective challenges for direct experimental validation.

Although we are currently unable to experimentally confirm the dynamic oscillations in Pt_{SA} coordination predicted by AIMD simulations, we believe this behavior is theoretically plausible. As temperature and reactant adsorption/desorption drive the reaction, it is reasonable to expect that the coordination number of Pt_{SA} will change dynamically, potentially exhibiting cyclic behavior, which aligns with the definition of catalytic reactions. We recognize that monitoring rapid atomic structure changes in catalysts during reactions (on the millisecond or femtosecond timescale) is a key area of research. In our future studies, we will focus on technological advancements to further confirm the conclusions of this paper.

Comment 11. Finally, the article does not discuss the environmental impact or economic feasibility of using this catalyst in industrial applications. Factors such as the cost of materials, energy consumption during synthesis, and potential recycling methods for spent catalysts are not addressed.

Reply: Thank you for your valuable feedback. We agree that the environmental impact and economic feasibility of using this catalyst in industrial applications are important factors to consider. In the revised manuscript, we will add a discussion on the cost of materials, energy consumption during synthesis, and potential recycling methods for spent catalysts. Due to the proprietary nature of large-scale catalyst production knowledge, the lack of accessible information has made the manufacturing cost assessment more complex. Moreover, accurate cost and energy consumption estimates typically require expensive software, specialized expertise, and non-public pricing data and production information. Therefore, we have only quantitatively compared the raw material consumption costs of Pt/OM CeZrO₂ and the commercial catalyst based on catalyst loading, including the consumption costs of Pt, Pd, Ce, Zr, and PS microsphere templates. We have also briefly discussed processing and equipment costs, energy consumption during synthesis, and potential recycling methods for spent catalysts in SI. This will help provide a more comprehensive assessment of its practical applications. For further details, please refer to the revision.

Revision:

(1) **On Supplementary Page34**, we have compared the raw material consumption costs of Pt/OM CeZrO₂ and the commercial catalyst based on catalyst loading as follow:

Supplementary Fig. 30: (a) Loading amount and (b) Material consumption costs comparison

between the Pt_{SA}/OM CeZrO₂ integrated monolith (0.4 g_{Pt}/L) and commercial PtPd/CeZr based catalysts (0.9 g_{Pt}/L+ 0.1g_{Pd}/L).

(2) **On Supplementary Page 34**, we have added a discussion on the cost of materials, energy consumption during synthesis, and potential recycling methods for spent catalysts as follow:

First, we compared the raw material consumption cost per cubic meter for the Pt_{SA}/OM CeZrO₂ integrated monolith and commercial PtPd/CeZr-based catalysts based on their catalyst loadings (**Figure a**) and metal prices as of March 29, 2025 (Pt: \$33/g, Pd: \$33/g, Ce: \$0.04/g, Zr: \$0.01/g), including the consumption cost of PS microsphere materials (styrene, polyvinylpyrrolidone, and azo-bis-isobutyronitrile) at \$3.73/L. As shown in **Figure b**, the raw material consumption cost of Pt_{SA}/OM CeZrO₂ was reduced by 46%, from \$34,171/m³ to \$18,299/m³.

This technology can use conventional coating process equipment, so the equipment cost is similar to that of traditional catalysts. However, the drying and calcination steps may be more complex than traditional coating methods (to ensure uniformity in the OM structure and Pt loading), requiring more precise control, which could lead to a slight increase in processing costs. The production cost of PS microsphere raw materials is relatively low and has industrial production potential, but since large-scale production has not yet been realized, it may lead to higher production costs. In the future, it will be necessary to develop large-scale PS microsphere production technology or explore more economical alternatives such as activated carbon microspheres.

The energy consumption during the drying and calcination steps in this technology does not differ significantly from that of traditional washcoat methods. Microwave drying, though more power-intensive, only lasts for half a minute. The PS template removal process is integrated into the crystallization process of CeZrO₂ and only requires a slightly extended time during the low-temperature phase. However, the cost savings from the reduced amount of precious metal Pt should offset these additional consumptions, making the catalyst more economical.

The catalyst reuse and recovery methods are the same as those of traditional catalysts (mainly high-temperature reduction and acid leaching), and due to the excellent stability performance of this catalyst, significant savings in catalyst regeneration and replacement costs can be achieved.

(3) **On Page 34 Line 443-446**, we have added a discussion on the cost of materials, energy

consumption during synthesis, and potential recycling methods for spent catalysts as follow: “The raw material consumption costs of $\text{Pt}_{\text{SA}}/\text{OM CeZrO}_2$ and the commercial PtPd/CeZr -based catalyst were also compared, including the raw material cost of PS microspheres, as shown in **Fig.5h** and **Supplementary Fig. 30**. The raw material consumption cost of $\text{Pt}_{\text{SA}}/\text{OM CeZrO}_2$ was reduced by 46%, from $\$34,171/\text{m}^3$ to $\$18,299/\text{m}^3$.”

Reviewer:3

General comments: This paper comprehensively explored the Pt_{SA}/CeZrO₂ catalyst coated on an ordered macroporous structure for diesel oxidation. However, there are major issues (particularly points 1 and 2) detailed below that prevent its publication in its current form.

Reply: Thank you for your thoughtful and detailed feedback on our manuscript. We believe these revisions improve the clarity and quality of the manuscript and we hope that they meet your expectations. Please see our responses and detail revisions as listed below.

Comment 1. The authors need to justify the claim “a transformative approach for the scalable integration and rational manipulation of robust Pt_{SA} catalytic converters”. Both the scalable integration and production of robust Pt_{SA} have been reported. This is clear even from the reference list in this manuscript (3, 7, 11-13, 18). Then, how is this work transformative?

Reply:

Thank you for your in-depth review of this article. **Firstly, the transformative aspect of this work lies in its innovative solution to Pt loss, sintering, and overoxidation in Pt-based catalysts, achieved through precise control at both micron- and atomic- scale, overcoming the limitations of traditional approaches.** As a result, the Pt_{SA}/CeZrO₂ catalyst can withstand extreme conditions—800°C for 50 hours, 1100°C for 2 hours, 450°C for 1000 hours, and 10 vol% H₂O—while maintaining high catalytic oxidation activity comparable to commercial Com-PtPd/CeZr catalysts, even with 55% less precious metal loadings. This work also explores structure dynamics of low-coordination Pt_{SA} sites during high-temperature reactions and elucidates the suppressive effect of the OM structure coating on Pt loss. Moreover, our study shows that Pt_{SA}/CeZrO₂ could be integrated into a 3.4-liter commercial cordierite monolith using template and atomic capture techniques, which holds great potential for commercial applications. This work offers new insights and pave the way for the development of high-temperature-resistant catalysts.

(1) For production of robust Pt_{SA}: To prevent Pt sintering, Abhaya et al. developed the atom capture method (*Science* 353, 150-154 (2016)) to prepare single atom Pt_{SA}/CeO₂ catalyst. Subsequent studies have confirmed the universality of this method, leading to the preparation of Pt_{SA}/TiO₂, Pt_{SA} /Fe₂O₃, Pt_{SA}/MgAl₂O₄, and other single-atom catalysts. They also revealed the

mechanism of the atom capture technology, where the strong interaction between the support and Pt results in Pt being captured by the support in a single atom state, preventing Pt aggregation and sintering (*Science* 358, 1419-1423 (2017); *Nat. Commun.*10,234(2019); *ChemCatChem* 14, e202200919 (2022); *Nat. Commun.*13,7070 (2022)). As research progressed, researchers discovered that, in addition to preventing Pt-sintering, the oxidation state of Pt_{SA} sites also significantly impacts their catalytic activity (*Nat. Catal.*3, 1-10(2020); *Angew. Chem. Int. Ed.* 59,20691-20696(2020); *Angew. Chem. Int. Ed.* 60,26054-26062 (2021); *Nat. Commun.*13,7070 (2022)). In this study, we employed a simple Zr-assisted atom capture approach to prepare Pt_{SA}/Ce_{0.8}Zr_{0.2}O₂ catalysts. We study the effect of Zr doping on the local coordination structure of Pt_{SA}, and elucidate the stabilization mechanism of Pt_{SA}. **Our study deepens the understanding of the catalyst's stabilization mechanism at the atomic level. Besides, we conducted the first study on the dynamic oscillation of the coordination structure of robust Pt_{SA} under high-temperature reaction conditions.** Notably, previous studies lack systematic investigation of Pt loss dynamics during high-temperature preparation, as existing works merely employed post-aging ICP analysis without quantifying initial Pt loading or capture efficiency in atom-trapping processes (*Science* 353, 150-154 (2016); *Science* 358, 1419-1423 (2017); *Nat. Catal.* 3, 1-10 (2020); *Nat. Commun.* 11, 1062 (2020); *Angew. Chem. Int. Ed.* 59, 20691-20696 (2020); *Angew. Chem. Int. Ed.* 60, 26054–26062 (2021); *Chem. Eng. J.* 426, 131855 (2021); *Nat. Commun.* 13, 7070 (2022)). For the preparation of robust Pt_{SA} catalysts, we are **the first to propose a method to improve atom capture efficiency by constructing an OM-structured support coating, which suppresses 37.5% Pt loss.** This approach has significant implications for the efficient utilization of precious metals in high-temperature catalysis.

(2) Furthermore, this study develops a simple, one-step preparation method for scaling up Pt_{SA}/OM CeZrO₂ catalysts, overcoming challenges of complex processes and uniform growth. Unlike our previous work (*Environ.Sci.Technol.*58,8096-8108(2024)), which involved complicated methods and catalyst preparation of size of 0.5L, this approach is streamlined and successfully **scales up to 3.4L, meeting industrial DOC size requirements.** While previous research focused on soot filtration and regeneration, this study emphasizes DOC activity and high-temperature stability. Given the different research objectives, Pt loading was achieved using microwave-assisted

impregnation for uniform distribution. In conclusion, **this work presents a simple and scalable preparation method for the integrated diesel oxidation catalyst that can directly applied to industrial-scale diesel exhaust treatment.** We believe that the term "transformative" is justified. And we have revised the Introduction and Results to better highlight these innovations, as detailed in the Revision.

Revision:

- (1) On **Page 5 Line 68**, we added a description at the Introduction to emphasize the novelty of this work, as follows: “However, the implementation of strategies and their industrial application still face critical knowledge gaps and technological challenges. Particularly regarding how Zr doping affects the coordination dynamics of Pt_{SA} sites during high-temperature reactions, and how to achieve scalable monolithic integration of this OM catalyst for industrial use.”
- (2) On **Page 17 Line 367**, we added a description at the end of the **Results** in **Fig 3** and **Fig 4** to emphasize the novelty of this work, as follows: “In summary, while Zr doping is an established methodology, this study provides atomic-scale insights into the stabilization mechanism of catalysts, elucidating how Zr modulates the coordination dynamics of metal centers and regulates reactive oxygen species during high-temperature catalysis.”
- (3) On **Page 20 Line 446**, we added a description at the end of the **Results** in **Fig 5** to emphasize the novelty of this work, as follows: “While OM CeZrO₂ support coatings are established catalytic materials, this study develops the first one-step synthesis protocol and enables industrial-scale fabrication on 3.4L cordierite substrates. These findings indicate that the OM support not only promotes mass transfer and diffusion of reactants but also mitigates Pt loss under high temperature, thereby enhancing the long-term durability and performance of the Pt_{SA}/CeZrO₂ catalyst in industrial settings.”

Comment 2. The two concepts in the manuscript (ordered macropore structure to enhance stability and Zr doping to enhance activity) seem to be separated. Namely, one can probably use an ordered macropore structure to improve the stability of any catalyst, and load Pt_{SA}/CeZrO₂ to a conventional industrial catalyst carrier to enhance activity. No clear synergy between the two aspects is discussed. It could be meaningful exploration here for industrial applications, but from an academic

perspective, the storyline is thus not clear. It is recommended to separate the discussion of Pt_{SA}/CeZrO₂ and catalyst engineering by OM structure, otherwise, the work can be off-focus and distractive.

Reply: We sincerely appreciate the reviewer's insightful suggestions. We have improved the discussion of the effects of Zr-doping and ordered macropores (OM) construction on performance enhancement:

(1) We compared and analyzed three samples—Pt_{SA}/OM CeZrO₂ (proposed design), Pt_{SA}/OM CeO₂ (OM structure without Zr), and Pt_{SA}/Powder CeZrO₂ (Zr-doped without OM)—in the same figure (**Supplementary Fig. 29**) to clearly demonstrate the synergistic effects. Detailed explanations can be found in the following **Revisions**.

(2) We agree with your suggestion to separate the discussion of these two aspects, as it would enhance the clarity of the manuscript's storyline. Therefore, in **Fig 3 and Fig 4**, we have focused on the anti-overoxidation mechanism of the Zr-assisted atomic capture method, revealing the dynamic changes of Pt_{SA} during high-temperature reactions. In **Fig 5**, we have highlighted the influence mechanism of the OM structure on the catalyst's stability, revealing its anti-Pt loss mechanism.

(3) Additionally, we have revised the **Abstract** to separately address these two aspects, making the structure of the paper clearer, as detailed in following **Revisions**.

(4) Furthermore, we provide a detailed explanation to the reviewer regarding the manuscript's storyline, including the challenges addressed, the innovative solutions, the catalyst performance, and the proposed new mechanisms. This work focuses on the addressing the challenges of Pt loss, Pt sintering, and Pt overoxidation deactivation in Pt-based industrial catalysts triggered by high temperatures. We found that traditional approaches are not able to address all three deactivation problems simultaneously. However, in this study, we innovatively solved all three deactivation issues through precise structure regulations at both of micro and atomic level. Moreover, we thoroughly investigated the anti-overoxidation and anti-loss mechanisms of Pt, and for the first time, revealed the dynamic oscillation phenomenon of the Pt_{SA} coordination structure under high-temperature reaction conditions.

Specifically, at the atomic-scale, we used Zr-assisted atomic capture method to prepare anti-

sintering and anti-overoxidation Pt_{SA} active sites. We further studied the anti-overoxidation mechanism of Pt sites. Zr atom doping promotes the formation of surface oxygen vacancies (O_v) and low-coordination Pt_{SA} active sites, which prevent the d-orbital free electrons from being excessively occupied by Pt-O bonds. Additionally, we explored for the first time the dynamic oscillation of the coordination structure of Pt_{SA} sites under high-temperature reaction conditions, proving that dynamically low-coordinated Pt_{SA} is the true active center for propane oxidation. Transitioning to the micron-scale, we constructed an OM structure support, which effectively suppressed Pt loss under high-temperature flow atmospheres. This reduced Pt loss by 37.5% under extreme conditions, achieving comparable activity and stability to commercial PtPd/CeZr-based catalysts at half the Pt loading. The micron- and atomic-level structure regulation allowed us to solve all three deactivation problems and obtain a high-stability Pt_{SA} catalyst that withstands temperatures up to 800 °C. Ultimately, we achieved both scalable integration and rational manipulation of robust Pt_{SA} catalytic converters, demonstrating its industrial application potential.

Revision:

(1) On Supplementary Page 33, we have added a comparison of the performance of Pt_{SA}/OM CeZrO₂, Pt_{SA}/OM CeO₂, and Pt_{SA}/Powder CeZrO₂ catalysts to study the synergistic effect of Zr doping and the OM structure on the catalyst stability.

Supplementary Fig. 29: Catalytic performance of aged Pt_{SA}/OM CeZrO₂, Pt_{SA}/OM CeO₂ and Pt_{SA}/Powder CeZrO₂. **a**, Light-off curves. **b**, Normalized reaction rate at 450 °C. (Aging conditions: 800 °C for 50 hours in air).

(2) On Page 19 Line 421-428, we have added analysis about the synergistic effect of the OM

structure integrated technology and Zr-assisted atom trapping technology: “ Furthermore, we compared Pt_{SA}/OM CeZrO₂, Pt_{SA}/OM CeO₂, and Pt_{SA}/Powder CeZrO₂ together, as shown in **Figure a**, the Pt_{SA}/Powder CeZrO₂ catalyst without the OM structure experienced a 75°C increase in T₉₀ (400°C → 475°C) after aging due to Pt loss, and the reaction rate decreased from 460 to 321 μmol/(g_{cat}*s). The Pt_{SA}/OM CeO₂ catalyst without Zr assistance saw a significant 140°C increase in T₉₀ (400°C → 540°C) after aging due to Pt_{SA} overoxidation (6-coordinate Pt-O), with the reaction rate decreasing from 460 to 227 μmol/(g_{cat}*s). These results demonstrate that the synergistic effect of the OM structure integrated technology and Zr-assisted atom trapping technology.”

(3) **On Page 2 Line 15-31**, we have modified the **Abstract** to separately discuss the impact mechanisms of Zr doping and the OM structure, making the storyline clearer. The specific modifications are as follows: “.....suffer from catalyst deactivation, such as Pt sintering, Pt overoxidation, and Pt loss.....Addressing these challenges necessitates an innovative design strategy that accounts for both micron-scale and atomic-scale structures. Specifically, we present an advanced integrated industrial-scale monolithic catalyst, Pt_{SA}/CeZrO₂, featuring..... At the atomic-scale, the dynamic low-coordinated Pt_{SA}, stabilized by Zr doping, Transitioning to the micron-scale, the OM structure helps prevent Pt loss..... As a result,.....This work reveals the dynamic evolution of Pt_{SA} configuration under oxidative conditions and provides a dual-scale design strategy for developing and scaling robust catalytic converters.”

(4) **On Page 17 Line 373**, we have added " Transitioning to the micron-scale realm, we further explored the influence of the OM structure on catalyst stability." to better transition and make the storyline of the paper clearer.

Comment 3. In the title, “ultra-stable” and “dynamic” seem conflicting. How can an “ultra-stable”structure easily change in a dynamic way? Besides, the specific application as a diesel oxidation catalyst is not mentioned in the title, and it is unfairly general unless this catalyst system has been proven effective for a variety of applications.

Reply:

(1) We understand the reviewer’s concern about the contradiction between "ultra-stable" and "dynamic." While the catalyst structure remains stable overall, the coordination number of Pt_{SA}

fluctuates during the reaction, as bond formation and breaking naturally occur. AIMD calculations show that the coordination number fluctuates between 4 and 5, without reaching an over-oxidized 6-coordinated Pt-O structure. A similar dynamic stability phenomenon has also been observed in other catalytic reactions (*J. Am. Chem. Soc.* **144**, 17140 (2022); *Nat Commun* **14**, 2512 (2023)). This dynamic behavior, which reflects stability over time, is similar to a person's weight fluctuating throughout the day but remaining stable over the long term. To avoid any confusion, we revised the title as following Revision.

(2) We agree with the reviewer's suggestion to include a more specific application context. However, we believe "High-Temperature Oxidation" better reflects the broader applicability of this catalyst system beyond diesel oxidation, as it may also effective in VOCs catalytic oxidation, catalytic desulfurization, and methanol oxidation. We have demonstrated the catalyst's stability and activity in toluene oxidation (**Supplementary Fig. 14**).

(3) To address the reviewer's concerns, we have revised the title to avoid potential misunderstandings and clarify the application scope.

Revision:

(1) The **Title** has been revised to "Ultra-stable Low-coordinated Pt_{SA}/CeZrO₂ Ordered Macroporous structure Integrated Industrial-scale Monolithic Catalysts for High-Temperature Oxidation"

(2) **On Supplementary Page 18**, we have added the following results of the activity and stability tests of toluene:

Supplementary Fig. 14: (a) Light-off curves for aged Pt_{SA}/CeO₂, Pt_{SA}/CeZrO₂ (Aging conditions:

800 °C for 50 hours in air). (b) Durability performance at 160 °C.

As shown in **Supplementary Fig. 14a**, the activity test demonstrates that the T_{90} for toluene oxidation over $\text{Pt}_{\text{SA}}/\text{CeZrO}_2$ after aging is approximately 25 °C lower than that of $\text{Pt}_{\text{SA}}/\text{CeO}_2$ (176 °C → 150 °C). Additionally, in the stability test at 160 °C (**Supplementary Fig. 14b**), $\text{Pt}_{\text{SA}}/\text{CeZrO}_2$ also exhibited good activity and stability, achieving 100 % operation with no decay over 24 hours.

Comment 4. Page 4, line 65, the term “macroporous nanostructure support at the nanoscale” is itself confusing. What exactly is the scale discussed here? In general, macropores refer to the pores in the material with more than 50 nm or even a few microns in diameter.

Reply: Thanks for your insightful comment. Upon revisiting the definition and scope of the nanoscale, we found that it is generally considered to be in the range of 1 to 100 nm, though in some cases, the range may be slightly extended depending on the context (*Mansoori, G.A. (2017). An Introduction to Nanoscience and Nanotechnology*). In this study, the OM structure coating, with a pore size of 0.7 microns and a coating thickness of approximately 20 microns, should be categorized within the micron-scale rather than the nano-scale. Therefore, we have revised the relevant term in the manuscript to "macroporous support at the micron-scale". This change clarifies the scale being discussed without affecting the core content or conclusions of the paper.

Revision:

- (1) We have revised the corresponding analysis on **Page 4 Line 63** as follows: Specifically, at the **micron-scale**, an ordered macroporous (OM) support, with its interconnected macropores and mesopores, can further improve Pt capture efficiency, reducing Pt loss under harsh temperatures.
- (2) On **Page 17 Line 372** as follows: “Impact of integrated OM **micron-structure** on catalyst stability”.

Comment 5. The two statements seem conflicting (page 5, line 87): “Compared with packed-bed reactors, honeycomb monolith-based structured catalytic devices offer higher heat and mass transport rates at a given pressure drop, reduction in transverse temperature gradients, and simpler scalability.” and “However, traditional washcoated monolithic catalysts often reduce material utilization efficiency and catalytic performance due to ineffective exposure of the reactive surface

and poor mass transport of reactants.” Why is the mass transport higher but there is still poor mass transport of reactants on the monolith-based structures?

Reply: Thanks for your insightful comment. The structure of honeycomb monolith catalytic devices can indeed improve heat and mass transfer rates, but mass transfer issues on the coated catalyst surface still exist, especially when the coating is thick or the surface exposure is insufficient. Therefore, the two statements are not contradictory. However, the wording may cause confusion, so, following the reviewer's suggestion, we have removed "and poor mass transport of reactants" from the second sentence to improve the clarity of the expression.

Revision: We have revised the corresponding analysis on **Page 5 Lines 91-94** as follows: “Compared with packed-bed reactors,... However, traditional washcoated monolithic catalysts, while benefiting from these structural advantages, often face challenges in material utilization and catalytic performance due to ineffective exposure of the reactive surface.”

Comment 6. Page 7, line 124, why “the increase in mesopores stemmed from the spaces between Ce-Zr solid solution nanoparticles”? What is the particle size discussed here that results in mesopores?

Reply: We have supplemented the TEM image of OM CeZrO₂ as suggested, which shows a particle aggregation structure with particle sizes ranging from 5 to 20 nm.

Revision: **On Supplementary Page 11**, we have included the TEM for OM CeZrO₂:

Supplementary Fig.7: TEM images of OM CeZrO₂ at different magnifications.

As shown in **Supplementary Fig.7**, OM CeZrO₂ is formed by the aggregation of Ce-Zr solid solution nanoparticles with particle sizes ranging from 5 to 30 nm.

Comment 7. Supplementary Fig. 6b. There is no comparison with a baseline sample to support that the catalyst here demonstrates excellent mechanical stability.

Reply: Thank you for your suggestions. To better compare the mechanical stability of the Pt_{SA}/OM CeZrO₂ catalyst, we also tested the Pt_{SA}/Powder CeZrO₂ sample (Supplementary Fig. B). Furthermore, the SEM images (Supplementary Fig. B inside) of the Pt_{SA}/OM CeZrO₂ catalyst after ultrasound treatment was added.

Revision: On Supplementary Page 10, we have included the mechanical stability for Pt_{SA}/Powder CeZrO₂ sample and SEM images of Pt_{SA}/OM CeZrO₂ after ultrasound treatment for different durations as follow:

Supplementary Fig. 6b Mechanical stability of Pt_{SA}/OM CeZrO₂, the inside shows the SEM images of the samples after ultrasound treatment for different durations.

As shown in **Supplementary Fig. 6b**, the mass loss of the OM sample during ultrasound treatment is slightly higher than that of the Powder sample, possibly due to minor damage to the OM pore structure. However, the mass loss difference between the two samples is minimal, with both being less than 1%. This suggests that the OM coating exhibits good mechanical stability, with no significant difference compared to traditional coatings. Furthermore, the SEM images (**Supplementary Fig. 6b inside**) of the Pt_{SA}/OM CeZrO₂ catalyst after ultrasound treatment for different durations show that the OM structure undergoes slight deformation, but the overall OM

framework structure remains stable, further confirming the excellent mechanical stability of the Pt_{SA}/OM CeZrO₂ catalyst. The good mechanical stability of the Pt_{SA}/OM CeZrO₂ catalyst may be attributed to its uniform and ordered pore structure, which allows stress to be evenly distributed across the surface and within the pores, rather than concentrating in a specific area, thereby better maintaining overall stability and reducing local collapse.

Comment 8. Page 8, line 164, it is recommended to change “conversion rate” to “conversion” to reduce possible confusion.

Reply: We have reviewed and revised the entire text based on the suggestions.

Revision: We have revised the entire text, for example, on **Page 8 Line 174**: “the conversion of Pt_{SA}/CeO₂ decreased significantly from 45% at 450 °C to just 12%.”

Comment 9. Supplementary Figs. 13b and c are confusing, and no caption or explanation is available there. Why do you need to test the Pt_{NP}/CeZrO₂ catalyst if it will transform into Pt_{SA}/CeZrO₂? Why does the activity of Pt_{NP}/CeO₂ always become lower but Pt_{NP}/CeZrO₂ does not? Also, the stability of Pt_{SA}/CeZrO₂ and Pt_{SA}/CeO₂ seems to be equally good. Then why is Pt_{SA}/CeZrO₂ claimed to show higher stability?

Reply:

(1) Thank you for your comment. We have revised the captions and explanations for **Supplementary Figs. 13b** and **13c** to clearly illustrate the experimental procedure and the comparison of the catalysts under different conditions. The confusion was likely due to the lack of explanation regarding the aging process and catalyst transformation, which we have now clarified in the updated captions.

(2) In this study, we test both Pt_{NP}/CeZrO₂ and Pt_{SA}/CeZrO₂ to understand the catalyst's behavior under high-temperature aging. The transformation from Pt_{NP} to Pt_{SA} is an important aspect of the catalyst's performance, and by testing the Pt_{NP}/CeZrO₂ catalyst, we can evaluate its initial activity and compare it to the performance of the transformed Pt_{SA} form. **This approach helps us understand the catalyst's resistance to high-temperature aging, which is crucial for its long-term application.** Fig. 2a show minimal activity change for Pt_{NP}/CeZrO₂ and Pt_{SA}/CeZrO₂ catalysts, while the aged Pt_{SA}/CeO₂ experiences a significant decline, with a T₅₀ increase of about

90 °C compared to Pt_{NP}/CeO₂. Although Pt_{SA}/CeO₂ avoids Pt sintering, it suffers considerable activity loss due to over-oxidation. These results highlight the importance of considering both sintering and over-oxidation resistance in catalyst design for long-term performance.

(3) The activity of Pt_{NP}/CeO₂ decreased during the three-cycle testing due to overoxidation of the Pt catalyst. In contrast, Pt_{NP}/CeZrO₂, with the assistance of Zr, maintains a lower oxidation state. Additionally, since the testing temperature reaches up to 800 °C, both catalysts could potentially convert to single-atom forms during the testing process. Pt_{SA}/CeO₂ shows a highly oxidized 6-coordinate Pt-O structure, while Pt_{SA}/CeZrO₂ shows a 4-coordinate Pt-O structure, leading to a significant difference in activity between Pt_{NP}/CeO₂ and Pt_{NP}/CeZrO₂.

(4) Although the cyclic stability of Pt_{SA}/CeZrO₂ and Pt_{SA}/CeO₂ is similar, the activity of Pt_{SA}/CeO₂ shows a significant decrease compared to the sample before thermal aging at 800 °C. The activities of Pt_{NP}/CeZrO₂ and Pt_{NP}/CeO₂ are comparable, but after thermal aging at 800 °C, there is a noticeable difference in the activities of Pt_{SA}/CeZrO₂ and Pt_{SA}/CeO₂. Therefore, we conclude that Pt_{SA}/CeZrO₂ is more resistant to high-temperature aging at 800 °C and maintains good activity, while Pt_{SA}/CeO₂, with its highly oxidized 6-coordinate Pt-O structure, shows a significant decrease in activity after thermal aging at 800 °C.

Revision: On Supplementary Page 17, We have revised and supplemented the captions and explanations for **Supplementary Figs. 13b** and **10c**, as follow:

Supplementary Fig. 13: (a) Stability performance of catalysts under different temperature; (b, c) Cycling stability performance of catalysts.

As shown in **Supplementary Fig. 13b**, Pt_{NP}/CeZrO₂ exhibits good cyclic stability, while the activity of Pt_{NP}/CeO₂ significantly decreases during the cycling tests. This could be attributed to the highest testing temperature of 800 °C, at which both catalysts may transform into Pt_{SA}. As shown in **Fig. 2a**, after thermal aging at 800 °C, Pt_{SA}/CeO₂ shows poor activity, whereas Pt_{SA}/CeZrO₂ maintains activity similar to that of Pt_{NP}/CeZrO₂. These results further confirm the transformation of Pt_{NP} to Pt_{SA} at 800 °C and highlight the poor high-temperature resistance of Pt_{NP}/CeO₂. As seen in **Supplementary Fig. 13c**, both Pt_{SA} catalysts obtained after 800 °C thermal aging exhibit good cyclic stability, but the activity of Pt_{SA}/CeO₂ is significantly lower than that of Pt_{SA}/CeZrO₂, with a T₅₀ increase of approximately 100 °C, further demonstrating its poor resistance to 800 °C.

Comment 10. Supplementary Table 1, the comparison is not straightforward since the conditions from the literature vary a lot. It is recommended to directly compare the turnover frequencies based on noble metal.

Reply: To address this, we have added the comparison in **Supplementary Table 2** to focus on TOF

values normalized to the noble metal content, ensuring a more consistent and meaningful comparison across the different studies.

Revision: We added **Supplementary Table 2** as follow:

Supplementary Table 2. Comparative TOF of catalysts for the catalytic oxidation of C₃H₈.

Catalysts	Pt loading (wt %) ^a	Reaction rate ($\mu\text{mol}/(\text{g}_{\text{cat}}*\text{s})$) ^b	TOF*10 ³ (s ⁻¹) ^b	Ref
Pt _{SA} /CeZrO ₂	0.04	102.7	501	This work (800 °C-Aged)
Pt _{SA} /CeO ₂	0.04	4.5	22	This work (800 °C-Aged)
Pt/MnCoO _x	0.04	3.5	17	Commercial (800 °C-Aged)
Pt/TiO _{2-x}	0.52	182.1	68	1
Pt/CeO ₂	0.23	112.8	96	3
Pt/LaCoO ₃	0.29	4.5	3	5
Pt/CeO ₂ -HA	0.84	258.9	60	6

^a Measured by ICP.

^b Calculated by the reaction rate at 300 °C.

Supplementary Table 2 shows that Pt_{SA}/CeZrO₂ exhibits the largest TOF for C₃H₈ oxidation compared with other Pt-based catalysts reported in the references, even after thermal aging at 800 °C.

Comment 11. Page 13, line 262. It is still not clear why the Pt_{SA}/CeZrO₂ is able to resist Pt over oxidation under harsh conditions. It is understandable that the Zr doping leads to asymmetrical planar-square coordination of PtO₄ as discussed on page 12, line 246, and the adjacent O thus becomes more active. However, why is this structure Pt_{SA}/CeZrO₂ “stable” and does not go to Pt/CeO₂? Is it because of the energy barrier for this transformation, or is it due to high Zr loading on CeO₂ and therefore no Ce area without Zr is available?

Reply: Thanks for your comments. We first explained the anti-oxidation mechanism of Zr doping and the position of Pt loading. To avoid any misunderstanding, we have also revised the original

text based on your suggestions and supplemented the description of the effects of Zr doping. Please see the Reply (1-2) and Revision (1-2) below.

(1) Because Zr doping can generate more oxygen vacancies (O_v) on the $Ce_{0.8}Zr_{0.2}O_2$ surface (as evidenced by EPR and DFT calculations), under high-temperature conditions, it promotes the substitution of Pt atoms for Ce atoms (located near Zr atoms and O_v). Consequently, a lattice distortion-induced O_v forms around the Pt single atom, creating a four-coordinate structure. The local environment of the Pt single atom (Pt_{SA}) comprises three Pt-O-Ce bonds and one Pt-O-Zr bond. On pure CeO_2 , Pt_{SA} is prone to over-oxidation. Under high-temperature conditions, Pt_{SA} substitute Ce atoms, forming a six-coordinate structure with six Pt-O-Ce bonds and no surrounding oxygen vacancies (O_v). In contrast, Zr doping introduces O_v , which weakens the excessively strong interaction between Pt_{SA} and the CeO_2 support. The localized O_v around Pt_{SA} facilitates the formation of a four-coordinate Pt-O structure during 800 °C thermal treatment, effectively suppressing over-oxidation, as confirmed by EXAFS, Raman spectroscopy, and DFT calculations. *In situ* near-ambient-pressure XPS and AIMD simulations further demonstrate that this low-coordination structure remains stable under reaction conditions. Zr doping regulates the coordination number of Pt_{SA} between 4 and 5 during adsorption/desorption processes dynamically, preventing the formation of over-oxidized six-coordinate configurations.

(2) We have prepared an OM $Ce_{0.8}Zr_{0.2}O_2$ solid solution, where Zr is not loaded onto the surface of CeO_2 . EDS-MAPPING shows that $Ce_{0.8}Zr_{0.2}O_2$ does not undergo phase separation into CeO_2 under reaction temperatures, which is consistent with reported literature (*Environ. Sci. Technol.* **55**, 12607–12618, (2021), *Chem. Eng. J.* **322**, 234–245, (2017)). Although the surface Zr content in $Ce_{0.8}Zr_{0.2}O_2$ is 1/5, the Zr content is much higher than the Pt_{SA} . STEM, EXAFS, and DFT analyses show that Pt_{SA} is more easily captured around Zr, with the local environment of Pt_{SA} containing both Pt-O-Ce and Pt-O-Zr bonds, forming a lower-energy configuration around Zr.

Revision:

(1) We have added and revised the corresponding analysis on Page 13 Lines 251-256 as follows: The most thermodynamically stable configurations of Pt_{SA}/CeO_2 and $Pt_{SA}/CeZrO_2$ are shown in **Supplementary Fig. 20**, where Pt_{SA} bonds with six oxygen atoms on pure CeO_2 , but coordinates with four oxygen atoms in $CeZrO_2$. Furthermore, thermodynamic calculations demonstrate the

lowest formation energy occurs when Pt_{SA} is anchored near Zr atoms and oxygen vacancies (O_v). The STEM results (**Supplementary Fig. 21**) also show an asymmetric atomic distance on both sides of the Pt_{SA} bright spots, which is caused by local lattice distortion induced by the differences in the atomic radii and electronegativity between Ce and Zr. This further indicates that Pt is more likely to be loaded around Zr atoms.

(2) We have added and revised the corresponding analysis on **Page 13 Lines 259** as follows: “In contrast to the symmetric Pt-O peak around 1.71 Å in Pt_{SA}/CeO₂ (**Fig. 3c**), the peak in Pt_{SA}/CeZrO₂ exhibits asymmetry (**Fig. 3d**), due to Pt loaded around Zr atom and the co-existence of Pt-O-Zr and Pt-O-Ce bonds in the Zr-doped sample. (*Angew. Chem. Int. Ed.* **60**, 26054–26062 (2021)).”

Comment 12. Page 19, line 400. The claim “the efficiency of the atomic trapping method for preparing Pt_{SA} catalysts was investigated for the first time” is clearly not true even based on the reference of this work.

Reply: We appreciate the reviewer's insightful comment. Upon re-examining the literature, we clarify that while previous works utilizing atom-trapping technology primarily focused on catalyst synthesis and stability, none quantitatively investigating Pt loss during high-temperature preparation and use. Most studies have only used ICP to characterize the Pt loading of catalysts after aging, without characterizing the Pt loading before aging. Indeed, no one has specifically focused on the Pt capture efficiency during the high-temperature preparation process of atom capture technology (*Science* **353**, 150-154 (2016), *Science* **358**, 1419-1423 (2017), *Nat. Catal.* **3**, 1-10 (2020), *Nat. Commun.* **11**, 1062 (2020), *Angew. Chem. Int. Ed.* **59**, 20691-20696 (2020), *Angew. Chem. Int. Ed.* **60**, 26054–26062 (2021), *Chem. Eng. J.* **426**, 131855 (2021), *Nat. Commun.* **13**, 7070 (2022)). Therefore, this study is the first to investigate Pt loss during high-temperature aging processes and propose strategies to mitigate Pt loss. In the field of robust Pt_{SA} catalysts prepared via atom capture technology, we are the first to propose a method to improve atom capture efficiency by constructing an OM-structured support coating, which suppresses Pt loss. This approach has significant implications for the efficient utilization of precious metals in high-temperature catalysis.

Comment 13. The manuscript should be at least better grammar- and spelling-checked. Mistakes are frequent such as “possesses” (page 2, line 22), “catalysts” (page 4, line 48), “preventing” (page 4, line 68), “industrial” (page 6, line 99), “Macroporous” (Fig. 5 caption).

Reply: We appreciate the reviewer’s careful reading of the manuscript and the helpful feedback regarding grammar and spelling. We have thoroughly proofread the manuscript and corrected the identified mistakes, with the changes marked in **BLUE**. The revised manuscript has been checked for grammar and spelling to ensure accuracy and clarity.

Point-to-point response to the reviewers' comments

Reviewer: 1

General comments: I have re-reviewed the manuscript and find that all the comments have been well-addressed. I have no further comments and think it can be accepted as is.

Reply: Thank you very much for your thoughtful feedback and for taking the time to review the manuscript again. I am pleased to hear that all the comments have been well-addressed. I appreciate your guidance throughout the review process.

Reviewer: 2

General comments: I thank the authors for fully addressing my concerns and carrying out the advices.

Reply: Thank you for your positive feedback. I appreciate your valuable comments and am glad to hear that the revisions meet your expectations. I appreciate your guidance throughout the review process.

Reviewer:3

General comments: The authors have significantly improved the work following reviewers' comments and the overall quality of this work is enhanced. The efforts to respond to reviewers have triggered meaningful and insightful explorations and remarkable new results. There are still several points that remain unclear as follows. I think this paper can be considered for publication after carefully addressing these comments.

Reply: Thank you for your constructive feedback regarding the improvements made to the manuscript. I appreciate your recognition of the efforts to enhance the work. I will carefully address the remaining points, and my detailed responses are as follows.

Comment 1. Fig. 3c and d. I am still having trouble understanding the inlet cartoons showing the Pt structure. At first glance, both Pt structures seem symmetric (although an asymmetric nature is mentioned in the text), and the difference between Pt-O_(Ce) and Pt-O_(Zr) is not clear. Further

clarification is required.

Reply: Thank you for your careful review and insightful questions regarding the Pt structure in **Fig. 3c-d**. First, we have modified the inlet cartoon in **Fig. 3d** (see **Revision 1** for details) by moving the Pt atom outward from the center and labeling the Pt-O-Zr and Pt-O-Ce bonds to better highlight the asymmetry of its coordination structure. Secondly, to better illustrate the differences in Pt coordination, we have now directly compared the EXAFS spectra of aged Pt_{SA}/CeO₂ and Pt_{SA}/CeZrO₂ in **Supplementary Fig. 23**. Additionally, we have added a more detailed explanation of the cause of the asymmetric Pt-O peak in both the main text and the supplementary information, as detailed in the **Revision**.

Revision:

(1) On **Page 12**, we have revised the inlet cartoons in **Fig 3d** as follow:

Fig. 3 (c,d) Extended X-ray Absorption Fine Structure (EXAFS) data fitting for aged Pt_{SA}/CeO₂ (c) and Pt_{SA}/CeZrO₂.

- (2) On **Page S13 Line 258-265**, we have added a more detailed explanation of the cause of the asymmetric Pt-O peak as follow: “In contrast to the symmetric Pt-O_(Ce) peak around 1.71 Å in Pt_{SA}/CeO₂ (**Fig. 3c**), the peak in Pt_{SA}/CeZrO₂ exhibits asymmetry (**Fig. 3d** and **Supplementary Fig. 23**), due to Pt loaded around Zr atom and the co-existence of Pt-O_(Ce) and Pt-O_(Zr) bonds with different but similar bond lengths. (*Angew. Chem. Int. Ed.* **60**, 26054 (2021); *Nat Commun* **10**, 234 (2019)) Meanwhile, an asymmetric square-planar Pt₁O₄ geometry of Pt_{SA}/CeZrO₂ was evidenced by three shorter Pt-O_(Ce) of 1.67 Å and one longer Pt-O_(Zr) of 1.97 Å.”
- (3) On **Page S27**, we compared the EXAFS data for aged Pt_{SA}/CeO₂ and Pt_{SA}/CeZrO₂ in the same figure (**Supplementary Fig. 23**) and provided the corresponding analysis as follow:

Supplementary Fig. 23: EXAFS data for aged Pt_{SA}/CeO₂ and Pt_{SA}/CeZrO₂.

As shown in Fig. 3c-d, Supplementary Table 3, and Supplementary Fig. 23, the first shell of Pt_{SA} on CeO₂ consists of six Pt-O_(Ce) bonds of the same length, producing a symmetric Pt-O peak. In contrast, the first shell of Pt_{SA} on Ce_{0.8}Zr_{0.2}O₂ has three shorter Pt-O_(Ce) bonds at 1.67 Å and one longer Pt-O_(Zr) bond at 1.97 Å. The distortion arises from the different ionic radii of Ce⁴⁺ (~0.92 Å) and Zr⁴⁺ (~0.80 Å), which induce local strain and alter Pt–O bond lengths. These similar but differing bond lengths result in an asymmetric Pt-O peak, further supporting that Pt is loaded near Zr atoms on Pt_{SA}/CeZrO₂.

Comment 2. It is interesting that Pt can still form single atoms with a high Zr doping content in Pt_{SA}/Ce_{0.6}Zr_{0.4}O₂. Then, with a wide range of Zr doping from 0 to 40%, why does Pt/Ce_{0.8}Zr_{0.2}O₂ show the best performance?

Reply: We appreciate the reviewer’s insightful observation and comments. To clarify the optimal performance of Pt_{SA}/Ce_{0.8}Zr_{0.2}O₂, we have added an analysis and corresponding Ref (*Small* **15**, 1903058 (2019)) as detailed in follow **Revision**.

Revision: On **Page S15 Line 143-147**, we have added a more detailed explanation toward the best performance of Pt_{SA}/Ce_{0.8}Zr_{0.2}O₂ as follow: “The superior performance of PtSA/Ce0.8Zr0.2O2 is likely attributed to the preferential loading of PtSA primarily near the Zr sites. In contrast, a low Zr content (< 0.1) may not be sufficient to affect the coordination structure of most PtSA. Additionally, excessive Zr doping (> 0.3) reduces the surface Ce³⁺ species with variable valence, thereby

inhibiting the catalytic oxidation reaction. (*Small* **15**, 1903058 (2019)) Therefore, $\text{Pt}_{\text{SA}}/\text{Ce}_{0.8}\text{Zr}_{0.2}\text{O}_2$ was selected for subsequent research.”

Comment 3. The comparison of $\text{Pt}_{\text{SA}}/\text{CeZrO}_2$ and PtPd/CeZr in Fig. 2e and Fig. 5f is really remarkable. Can you reiterate better why $\text{Pt}_{\text{SA}}/\text{CeZrO}_2$ shows amazingly better performance compared to the commercial sample? I fully understand the exact recipe for the industrial catalyst is confidential. However, there should be reasons behind its complicated composition with the presence of multiple elements Pt, Pd, Ce, Zr, Y, Ba, Al, and Si. Each element may play a role here in enhancing the performance (e.g. Pd assists with low-temperature performance and other metals act as promoters). The most interesting results come from Fig. 2e, in which the catalyst is in powder form without integration of the OM structure based on my understanding. The performance of $\text{Pt}_{\text{SA}}/\text{CeZrO}_2$ is better than the commercial catalyst at all temperatures. Why does $\text{Pt}_{\text{SA}}/\text{CeZrO}_2$ (even without the OM structure) outperform the commercial sample in all aspects with a much simpler composition and structure?

Reply: Thanks for your valuable comment. **(1)** Based on the suggestion, we have provided a more detailed explanation of the performance differences between $\text{Pt}_{\text{SA}}/\text{CeZrO}_2$ and commercial PtPd/CeZr catalysts, mainly attributing the superior performance of $\text{Pt}_{\text{SA}}/\text{CeZrO}_2$ to its excellent sintering resistance and metal loss resistance, as detailed in following **Revision**. **(2)** Although the commercial PtPd/CeZr formulation is quite complex, with multiple elements potentially enhancing the catalyst's activity and stability, it may lack the precise control over the active component's coordination environment and coating structure seen in $\text{Pt}_{\text{SA}}/\text{CeZrO}_2$ catalyst, leading to lower noble metal utilization. **(3)** It seems there may have been a misunderstanding regarding the data presented in Fig. 2e. In fact, the $\text{Pt}_{\text{SA}}/\text{CeZrO}_2$ catalyst throughout the paper have the OM structure. To clarify this point and prevent any confusion, we have highlighted this aspect in the **Abstract** by stating “Specifically, we present an advanced integrated industrial-scale monolithic catalyst, $\text{Pt}_{\text{SA}}/\text{CeZrO}_2$, featuring isolated Pt single atoms (Pt_{SA}) supported on $\text{Ce}_{0.8}\text{Zr}_{0.2}\text{O}_2$ with an ordered macroporous (OM) structure”.

Revision: On **Page 20 Line 437-442**, we have added a more detailed explanation toward the performance differences between OM $\text{Pt}_{\text{SA}}/\text{CeZrO}_2$ and commercial PtPd/CeZr catalysts as follow:

“We further compared the OM Pt_{SA}/CeZrO₂ with the commercial PtPd/CeZr-based catalyst. First, the commercial PtPd/CeZr has noble metal particles around 15 nm (**Supplementary Fig. 15c**), while the OM Pt_{SA}/CeZrO₂ maintains single-atom dispersion even after aging at 800°C, offering better Pt-sintering resistance and noble metal utilization. Moreover, under high-temperature flowing conditions, the OM Pt_{SA}/CeZrO₂ shows better resistance to Pt-loss, with a 37.5% reduction in Pt loss compared to the commercial PtPd/CeZr (**Fig. 5d-e**). Therefore, despite having a lower Pt loading, the OM Pt_{SA}/CeZrO₂ (0.4 g_{Pt}/L) exhibits comparable catalytic combustion activity to the commercial PtPd/CeZr-based catalyst (0.9 g_{Pt}/L+ 0.1 g_{Pd}/L).”

Supplementary Fig. 15: (c) TEM images of Com-PtPd/CeZr.

Fig. 5d, Changes in Pt loading of catalysts after aging at 800 °C and 1100 °C (20% O₂, N₂ balance, flow rate 50 mL/min, 800 °C for 50 hours/1000 °C for 2 hours). **e**, Schematic diagram illustrating the inhibitory effect of the OM structure on Pt loss.

Comment 4. It is interesting to see the activity of Pt_{SA}/CeZrO₂ is even higher in the presence of SO₂, according to Supplementary Fig. 16 (both a and b). Any explanation for this?

Reply: Thanks for your insightful comment. Typically, SO₂ unavoidably deactivates catalysts in most heterogeneous catalytic oxidations. However, for Pt-based catalysts, SO₂ exhibits an

extraordinary boosting effect in C_3H_8 catalytic oxidation, which is not a new phenomenon (*J. Catal.* **184**, 491(1999); *J. Catal.* **201**, 247 (2001); *ACS Catal.* **10**, 13543(2020)). In our previous work, we specifically studied the promoting effect of SO_2 on Pt-based catalysts for C_3H_8 oxidation. We found that SO_2 induces coupling between the S 3p and Pt 5d orbitals, promoting the formation of surface interface sulfite species (Pt–O– SO_3 or (Pt-S-O)-Ti) on the catalyst. For C_3H_8 molecules, a cleaved oxygen atom in Pt–O– SO_3 structure efficiently activates C–H bonds and shifts the oxidation pathway from propionate to a more efficient acrylate pathway, thereby enhancing C_3H_8 oxidation (*Environ. Sci. Technol.* **58**, 3041 (2024); *Environ. Sci. Technol.* **58**, 18020 (2024)). Therefore, in this study, we have added a brief explanation and reference, as detailed in the following **Revision**.

Revision: On **Page S20 Line 201-203**, we have added an explanation about the SO_2 influence as follow: “The introduction of SO_2 would induce a strong coordination interaction between Pt species and SO_4^{2-} , and the two can work synergistically to break the C-C and C-H bonds in C_3H_8 . (*Environ. Sci. Technol.* **58**, 18020 (2024))”

Comment 5. In Fig. 3g, I am having trouble understanding how the trajectory for the Pt coordination number aligns with scattered data points in the AIMD simulation. Could you please clarify this more?

Reply: We appreciate the valuable comments from the reviewers. We first extracted the configurations at specific times from the AIMD results to calculate the Pt_{SA} coordination numbers at those times, as shown by the scatter points in **Fig 3g**. In the previous **Fig 3g**, the curve did not directly correspond to the scatter points, but rather depicted the dynamic fluctuation of the coordination structure, which could cause some confusion. Following your suggestion, we have revised the **Fig 3g** (see **Revision**). The curve now directly matches the coordination numbers represented by the scatter points, offering a clearer illustration of the dynamic fluctuation of the Pt_{SA} coordination number during the high-temperature reaction.

Revision: On **Page 12**, we have revised the **Fig. 3g** as follow:

Fig. 3g, Selected snapshots of the *ab initio* molecular dynamics (AIMD) trajectory for Pt single atom coordination number over $\text{Pt}_{\text{SA}}/\text{CeO}_2$ and $\text{Pt}_{\text{SA}}/\text{CeZrO}_2$ under actual reaction conditions (1 C_3H_8 and 20 O_2 molecules at 400 °C).

Comment 6. Supplementary Fig. 7 shows the particle size of Ce-Zr ranging from 5-30 nm. However, these particles seem to be clearly separated from each other based on the interpretation of panel c. Does it really support the original claim that mesopores stemmed from “the spaces” between Ce-Zr solid solution nanoparticles? Or are Ce-Zr particles with porous structure? It is still not clear here.

Reply: Thank you for your careful observation and review. Indeed, the particles in **Fig. 7c** appear to be separated. It is important to clarify that the OM CeZrO_2 structure is not a monolayer; instead, it is composed of small particles that are stacked together with a rough surface. The particles we observed in the **Fig. 7c** represent the more prominent CeZrO_2 particles on the surface, but they are not actually dispersed. If they were dispersed, the OM structure could not have formed. The broken OM layered structure is clearly visible in **Fig. 7a-b**, which further supports this explanation.

Supplementary Fig.7: TEM images of OM CeZrO_2 at different magnifications.

Comment 7. Despite the valid argument from the authors, the claim still seems an overstatement: “the efficiency of the atomic trapping method for preparing Pt_{SA} catalysts was investigated for the first time”. There should be more descriptive language in this sentence to narrow down and better frame the contributions of this work, and make sure that “first time” holds true here.

Reply: We fully understand your concern and suggestion. We have revised the relevant description and added the condition "under high temperature" for clarification.

Revision: **On Page 18 Line 399-400**, we have revised the sentence as follow: “This work first proposes a scalable strategy for designing ordered macroporous (OM) catalysts to enhance atom trapping efficiency under high temperature.”

Reviewer:4

I co-reviewed this manuscript with one of the reviewers who provided the listed reports. This is part of the *Nature Communications* initiative to facilitate training in peer review and to provide appropriate recognition for Early Career Researchers who co-review manuscripts.

Reply: Thank you very much for taking the time to review the manuscript again. I appreciate your guidance throughout the review process.

This paper comprehensively explored the PtSA/CeZrO₂ catalyst coated on an ordered macroporous structure for diesel oxidation. However, there are major issues (particularly points 1 and 2) detailed below that prevent its publication in its current form.

1. The authors need to justify the claim “a transformative approach for the scalable integration and rational manipulation of robust PtSA catalytic converters”. Both the scalable integration and production of robust PtSA have been reported. This is clear even from the reference list in this manuscript (3, 7, 11-13, 18). Then, how is this work transformative?

2. The two concepts in the manuscript (ordered macropore structure to enhance stability and Zr doping to enhance activity) seem to be separated. Namely, one can probably use an ordered macropore structure to improve the stability of any catalyst, and load PtSA/CeZrO₂ to a conventional industrial catalyst carrier to enhance activity. No clear synergy between the two aspects is discussed. It could be meaningful exploration here for industrial applications, but from an academic perspective, the storyline is thus not clear. It is recommended to separate the discussion of PtSA/CeZrO₂ and catalyst engineering by OM structure, otherwise, the work can be off-focus and distractive.

3. In the title, “ultra-stable” and “dynamic” seem conflicting. How can an “ultra-stable” structure easily change in a dynamic way? Besides, the specific application as a diesel oxidation catalyst is not mentioned in the title, and it is unfairly general unless this catalyst system has been proven effective for a variety of applications.

3. Page 4, line 65, the term “macroporous nanostructure support at the nanoscale” is itself confusing. What exactly is the scale discussed here? In general, macropores refer to the pores in the material with more than 50 nm or even a few microns in diameter.

4. The two statements seem conflicting (page 5, line 87): “Compared with packed-bed reactors, honeycomb monolith-based structured catalytic devices offer higher heat and mass transport rates at a given pressure drop, reduction in transverse temperature gradients, and simpler scalability.” and “However, traditional washcoated monolithic catalysts often reduce material utilization efficiency and catalytic performance due to ineffective exposure of the reactive surface and poor mass transport of reactants.” Why is the mass transport higher but there is still poor mass transport of reactants on the monolith-based structures?

5. Page 7, line 124, why “the increase in mesopores stemmed from the spaces between Ce-Zr solid solution nanoparticles”? What is the particle size discussed here that results in mesopores?

6. Supplementary Fig. 6b. There is no comparison with a baseline sample to support that the catalyst here demonstrates excellent mechanical stability.

7. Page 8, line 164, it is recommended to change “conversion rate” to “conversion” to reduce possible confusion.

8. Supplementary Figs. 10b and c are confusing, and no caption or explanation is available there. Why do you need to test the PtNP/CeZrO₂ catalyst if it will transform into PtSA/CeZrO₂? Why does the activity of PtNP/CeO₂ always become lower but PtNP/CeZrO₂ does not? Also, the stability of PtSA/CeZrO₂ and PtSA/CeO₂ seems to be equally good. Then why is PtSA/CeZrO₂ claimed to show higher stability?

9. Supplementary Table 1, the comparison is not straightforward since the conditions from the literature vary a lot. It is recommended to directly compare the turnover frequencies based on noble metal.

10. Page 13, line 262. It is still not clear why the PtSA/CeZrO₂ is able to resist Pt over-oxidation under harsh conditions. It is understandable that the Zr doping leads to asymmetrical planar-square coordination of PtO₄ as discussed on page 12, line 246, and the adjacent O thus becomes more active. However, why is this structure PtSA/CeZrO₂ “stable” and does not go to Pt/CeO₂? Is it because of the energy barrier for this transformation, or is it due to high Zr loading on CeO₂ and therefore no Ce area without Zr is available?

11. Page 19, line 400. The claim “the efficiency of the atomic trapping method for preparing PtSA catalysts was investigated for the first time” is clearly not true even based on the reference of this work.

12. The manuscript should be at least better grammar- and spelling-checked. Mistakes are frequent such as “possesses” (page 2, line 22), “catalysts” (page 4, line 48), “preventing” (page 4, line 68), “industrial” (page 6, line 99), “Macroporous” (Fig. 5 caption).